



# Water storage and potential hazard of moraine-dammed glacial lake in maritime glaciation region—A case study of Bienong Co

Hongyu Duan[1], Xiaojun Yao[1], Huian Jin[2], Yuan Zhang[1], Qi Wang[3], Zhishui Du[3], Jiayu Hu[1], Qianxun Wang[4]

[1] College of Geography and Environment Science, Northwest Normal University, LanZhou, 730070, China
[2] Gansu Forestry Polytechnic, Tianshui, 741020, China
[3] Northwest Engineering Corporation Limited, Power China, Xi'an 710065, China
[4] Capital Urban Planning and Design Consulting Development Company Limited, Beijing 100038, China

*Correspondence to:* Xiaojun Yao (xj_yao@nwnu.edu.cn)

**Abstract.** The existence of glacial lakes in the Southeastern Tibetan Plateau (SETP) is a potential hazard to the downstream regions, as the failure of some lakes has potential to result in disastrous glacial lake outburst flood (GLOF) events of high-magnitude. In the present study, we conducted a comprehensive investigation for Bienong Co, an end moraine-dammed glacial lake in SETP, including its area evolution analysis, basin morphology simulation, water volume estimation, possible outburst triggers analysis, and one- and two-dimensional hydrodynamic simulation. The results show that the area of Bienong Co was $1.15\pm0.05$ km$^2$ in August 2020, which has remained generally stable over the past four decades. However, it exhibits the high risk of GLOFs due to its larger area, the steep and high moraine dam, the close distance to its mother glacier, and the surrounding steep slopes. The lake basin is relatively flat at bottom and steep on both flanks, and the slope near the glacier (16.5˚) is steeper than that near the moraine dam (11.3˚), with a maximum lake depth of ~181.04 m and a water volume of ~$1.02\times10^8$ m$^3$ in August 2020. Four scenarios of GLOFs based on different breach depths, breach widths and failure times were simulated using the hydrodynamic model of MIKE 11 and MIKE 21 to predict the potential impacts on downstream areas. An extreme-magnitude GLOF would have a catastrophic impact on the downstream region, with most of the settlements, all bridges and majority of Jiazhong Highway along the flow channel being completely submerged. However, in a low-magnitude GLOF, most settlements would be safe or partially inundated. It means that most of the residents in the flow channel of Bienong Co can avoid the damage caused by a low-magnitude GLOF (smaller breach depth of dam). Although three settlements in the downstream area are at risk of being completely submerged in a low-magnitude GLOF event, the flooding arrives late and people have enough time to escape. Finally, the maximum depths of glacial lakes with similar areas were compared for 16 glacial lakes with measured bathymetry data in the Himalayas and Bienong Co in SETP, according to the regional division of maritime and continental zones. The results show that glacial lakes located in maritime regions have larger depths than those in continental regions, and Bienong Co is the deepest glacial lake comparing others in the Himalayas. Therefore, a huge amount of water could be discharged by a potential GLOF event of Bienong Co, creating a serious hazard, which should be taken seriously in the future. Overall, this study of Bienong Co could provide a new understanding of the moraine-dammed glacial lakes in the SETP and a reference of GLOFs to the local government.

## 1 Introduction

Due to global warming, the accelerated retreat and thinning of glaciers has occurred in most regions compared to the last century (Zemp et al., 2019). One result is a rapid increase in the number, area, and volume of glacial lakes worldwide (Shugar et al., 2020; Wang et al., 2020). This is a natural occurrence, where glacier meltwater is confined and stored in certain depressions, and the dam materials can be moraine, ice or bedrock (Vilímek et al., 2013). However, once the dam is damaged and the water is suddenly and catastrophically released, Glacial Lake Outburst Floods (GLOFs), a severe social and geomorphic impacts several dozens of kilometers and more downstream can be caused (Lliboutry, 1977; Richardson and Reynolds, 2000; Osti and Egashira, 2009; Carrivick and Tweed, 2016; Cook et al., 2018; Zheng et al., 2021). Among these,



the moraine-dammed glacial lakes are of particular attention owing to their large volume (Fujita et al., 2013; Veh et al., 2020), weak dam composition, and predisposition to various triggering, such as the ice and/or rock avalanches (Emmer and Cochachin, 2013; Nie et al., 2018) which are the most common sources of GLOFs (Watanbe and Rothacher, 1996; Westoby et al., 2014). The Himalayas and the Southeastern Tibetan Plateau (SETP) region are hotspots for the occurrence of GLOFs caused by moraine-dammed glacial lakes (Wang, 2016). Study shows that the Himalayas, especially the southern region will enter a high

incidence period of GLOFs at the coming decades due to the fluctuated pattern (Veh et al., 2020).

The SETP is a broad mountainous area covering the central and eastern Nyainqêntanglha Ranges, eastern Himalayas and western Hengduan Mountains and having the most complicated terrains (Ke et al., 2014). Owing to a warm and humid climate, a plenty of maritime glaciers have developed here (Yang et al., 2008), featured as the adequate recharge, strong ablation, low snowline distribution, high temperature, fast movement, and strong geological as well as geomorphological effect (Li et al.,

1986; Qin et al., 2007; Liu et al., 2014), which have been observed with the most negative mass balances during the past decades (Kääb et al., 2012; Neckel et al., 2014; Kääb et al., 2015; Brun et al., 2017; Dehecq et al., 2019). Therefore, active glacial processes in conjunction with heavy rainfall during the monsoon season expose the region to the threat of glacial lake related hazards (Wang et al., 2012b). Studies of glacial lakes in the SETP mainly focused on regional-scale assessment of glacial lake changes (Wang et al., 2011a; Song et al., 2016; Wang et al., 2017; Zhang et al., 2020; Zhang et al., 2021),

identification of potentially dangerous glacial lakes (Wang et al., 2011; Liu et al., 2019; Duan et al., 2020; Qi et al., 2020), site-specific analysis of formation mechanism, development trend, risk evolution and management measures of GLOFs (Cui et al., 2003; Cheng et al., 2008, 2009; Sun et al., 2014; Liu et al., 2021; Wang et al., 2021), exploration of geological features of a single glacial lake (Yuan et al., 2012; Liu et al., 2015; Huang et al., 2016). Fewer studies applied hydrodynamic models to simulate outburst flood of glacial lakes in the SETP. Wang et al. (2011b) evaluated the applicability of ASTER GDEM (Global

Digital Elevation Model) and SRTM DEM in the simulation of GLOF process based on HEC RAS hydrodynamic model (Brunner, 2002). Zheng et al. (2021) analyzed and reconstructed a GLOF process chain of Jinwu Co using the published empirical relationships and GIS-based r.avaflow simulation tool (Mergili et al., 2017; Pudasaini and Mergili, 2019; Mergili and Pudasaini, 2020).

As a key factor related to the peak discharge and outburst volume of a GLOF event (Evans, 1987; Huggel et al., 2002),

lake storage capacity is difficult to directly obtain by means of satellite remote sensing approach. Currently, owing to the easy availability of area information from remote sensing images, the volume of glacial lakes is generally estimated using the developed empirical formulas connect glacial lake area and volume based on bathymetric data for a small number of glacial lakes (O'Connor et al., 2001; Huggel et al., 2002; Yao et al., 2014). However, the estimated volume maybe inaccurate because the unique geographical conditions of different glacial lakes (Cook and Quincey, 2015).

GLOFs are frequent in the SETP region (Sun et al., 2014; Zheng et al., 2021) where glacial lakes are formed by strong movement of maritime glaciers. Whereas, there are few publicly available bathymetric data of glacial lakes in the SETP region and previous bathymetric work in the High Mountain Asia region was carried out mainly for those glacial lakes located in the Himalayas (LIGG/WECS/NEA, 1988; Geological survey of India, 1995; Yamada, 1998; Mool et al., 2001; Sakai, 2003; Yamada, 2004; ICIMOD, 2011; Sakai, 2012; Yao et al., 2012; Wang et al., 2015; Haritashya et al., 2018; Sharma et al., 2018;

Li et al., 2021). This is unfavorable to fully understand the morphology and disaster prevention of glacial lakes in the SETP region.

Due to the distribution in harsh environments, the bathymetry measurement of glacial lakes is difficult and risky (Zhang et al., 2020). In recent years, the Unmanned Surface Vessel (USV) have developed rapidly (Liu et al., 2016), which have been widely used in scenarios such as bathymetric map creation, transportation, environmental monitoring, and moraine surveys

(Larrazabal and Peñas, 2016; Yan et al., 2010; Specht et al., 2019a) owing to the favorable personnel safety and security and the high flexibility in complex environments. Despite the wide application in the ocean (Bibuli et al., 2014; Specht et al., 2019b), the USV are rarely employed inland, particularly on glacial lakes (Li et al., 2021). However, high altitude, harsh





conditions, and sophisticated instruments mean that the underwater topography survey of glacial lakes is a potential field of USV application (Li et al., 2021).

In this study, we aim to complete an investigation of the potential GLOF hazard of a typical end moraine-dammed glacial lake, Bienong Co (Co means lake in Tibetan) in SETP based on remote sensing data, field bathymetric data, combining hydrodynamic model. The main tasks include to investigate the evolution of glacial lake' area and parent glacier' elevation based on remote sensing images, model the morphology and estimate the volume of Bienong Co based on field bathymetry data, assess the potential GLOF triggers of Bienong Co, and simulate potential GLOF caused hazards based on field bathymetry

data and DEM data using MIKE 11 and 21 hydrodynamic models (DHI, 2007). In addition, the second purpose is to develop a relationship between the area and volume of moraine dammed glacial lakes based on bathymetric data of Bienong Co and other 16 glacial lakes in the Himalayas.

## 2 Study area

The study objective, Bienong Co glacial lake, is located in the upper area of Yi'ong Zangbo (Zangbo means river in Tibetan)

watershed (30˚05'-31˚03'N, 92˚52'-95˚19'E) in the SETP. The Yi'ong Zangbo originates from the Nyainqêntanglha Mountains, extends about 286 km in length, and drains an area covering 13,533 $km^2$, which is a one-level tributary of the Parlung Zangbo and a one-level tributary of the Yarlung Zangbo (i.e., the Brahmaputra River) (Fig. 1). The terrain is high in the west and low in the east with high mountains and valleys. The climate is warm and humid, featuring the mean annual precipitation of 958 mm and mean annual temperature of 8.8℃ (Ke et al., 2013, 2014). There were 1,907.76 $km^2$ glacier coverage, 105 moraine-

dammed glacial lakes with a total area of 16.87 $km^2$ in 2016 (Duan et al., 2020). Seven glacial lakes, including Bienong Co, were considered to be highly dangerous (Duan et al., 2020), of which, the Jinwu Co collapsed on June 26, 2020 (Zheng et al., 2021). As of 2021 there have been three recorded large GLOF events in the basin, all of which caused very significant damage to the infrastructure settings in the downstream region (Sun et al., 2014; Yao et al., 2014; Zheng et al., 2021) (Fig.1).

Bienong Co is a typical end moraine-dammed lake constrained by the snout of the mother glacier (Mulang Glacier) on

the south and a massive unconsolidated terminal moraine dike on the northwest (Fig. 1). The elevation of water surface is 4745 m with an area of 1.15 ± 0.05 $km^2$ in 2021. Mulang Glacier has an area of 8.29 ± 0.22 $km^2$ and mean surface slope of ~18.28˚. The northeast-southwest oriented frontal moraine dam has the length of ~550 m, mean crest height of ~72 m, mean freeboard of ~10 m, the distal facing slope of ~35˚. A natural outlet with a width of ~50 m in the right of the dam facing downstream. The dam is composed of poorly consolidated, unsorted and uncohesive sediment and the existence of ice core cannot be

determined at present. The flow channel from the Bienong Co along with Xiong Qu (Qu means river in Tibetan) to converging with Song Qu stretches ~52.98 km, with the river longitudinal drop ratio of 14.48‰. There are 19 settlements and 13 bridges densely distributed along the above river channel, as well as a large amount of agricultural land. In addition, the Jiazhong Highway extends closely along the river channel (Fig. 1).

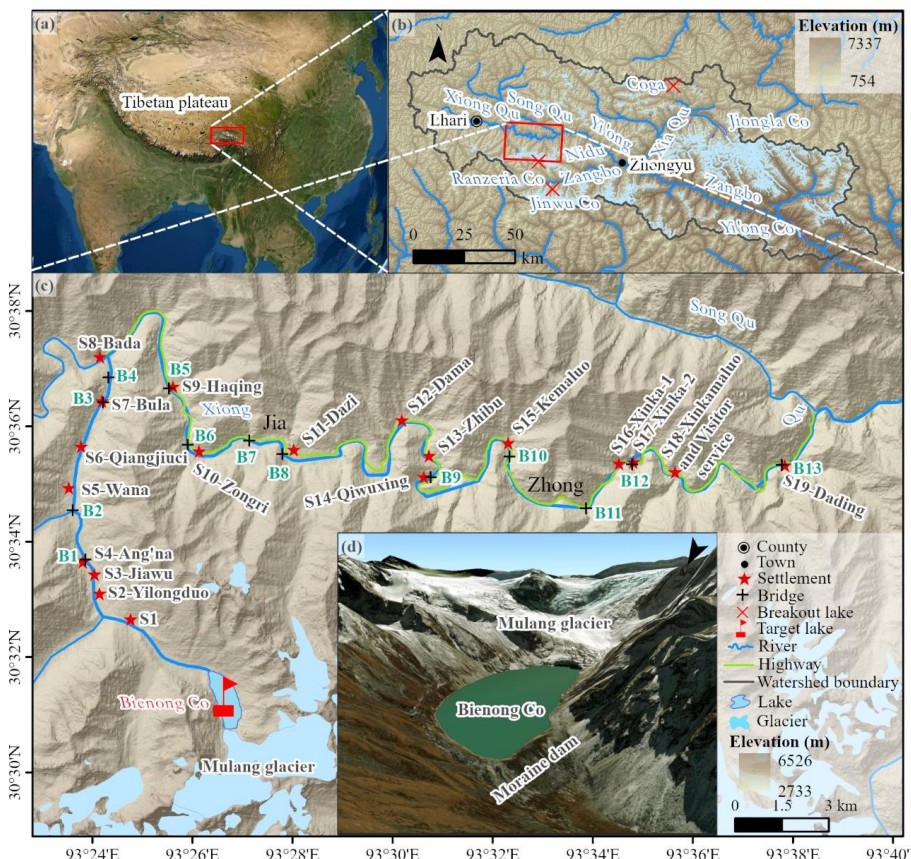


**Figure 1.** The overview of the study area. (a) The location of Yi'ong Zangbo watershed, (b) the location of Bienong Co, (c) distribution of settlements as well as bridges within 52.98 km downstream the Bienong Co, and (d) the close up view of Bienong Co. Bienong Co' location in High Elevation Asia and the proximity picture are from Esri ArcGIS Earth software, topography map is ALOS PALSAR DEM and the locations and the names of settlements and bridges along the flow channel
are obtained from Esri ArcGIS Earth software and Zhongke Tuxin LocaSpace software.

## 3 Materials and methodologies

### 3.1 Delineation of glacier and glacial lake

Due to the high spatial-temporal resolution, satellite images are primary data sources for delineating and monitoring changes in glaciers and glacial lakes (Wang et al., 2013; Zhang et al., 2015; Khadka et al., 2018; Wang et al., 2020). In this study, multi-
sources of remote sensing images were used to investigate the evolution of Bienong Co and Mulang Glacier (Table 1). Since remote sensing images in STEP are heavily affected by clouds and snow, only valuable cloud-free images in late summer and autumn were selected. Images of Landsat MSS on November 10, 1976, Landsat TM on October 9, 1988 and Landsat OLI on September 8, 2021 were used to study the surface area change of Binong Co. Meanwhile, AST14DEM (Maurer et al., 2019) from 2004 to 2018 were used to analyze the surface elevation change of Mulang Glacier.

Studies show that the automatic interpretation based on spectral characteristics of ground objects has advantages in efficiency (Bhardwaj et al., 2015; Zhang et al., 2019), but the manual extraction yields more accurate results (Fujita et al., 2009; Shugar et al., 2020). Considering only one lake and glacier being investigated in this study, the manual visual interpretation method was applied to delineate Bienong Co and Mulang Glacier. While the boundary of the Mulang Glacier





was manually revised based on the Second Chinese Glacier Inventory (SCGI). Then lake and glacier area were calculated

based on the UTM projection and the area error was estimated using the Eq. (1) (Wang et al., 2012b):

$$\varepsilon = \frac{\lambda^2 \cdot p}{2\sqrt{\lambda^2 + \lambda^2}} = \frac{\lambda \cdot p}{2\sqrt{2}}$$

(1)

where $p$ is the perimeter of the glacial lake and $\lambda$ is the spatial resolution of the images used.

**Table 1.** Details of multi-source dataset used in this study.

| Data | Date | Resolution | Application | Source |
|---|---|---|---|---|
| Landsat MSS | 1976-11-10 | 60 m | Glacier and glacial lake mapping | |
| Landsat TM | 1988-10-09 | 30 m | Glacier and glacial lake mapping | 1 |
| Landsat OLI | 2021-09-08 | 30/15 m | Glacier and glacial lake mapping, River channel mapping | |
| World Maxar image | 2017-2022 | 1 m | River channel mapping GLOFs simulation | 2 |
| AST14DEM dataset | 2004-11-05 2005-11-17 2009-10-18 2010-10-30 2013-09-27 2013-10-13 2018-10-27 | 30 m | Glacier elevation measuring | 3 |
| ALOS PALSAR DEM | 2006-2001 | 12.5 m | GLOFs simulation | 3 |
| GLC10 land use and land cover (LULC) product | 2017 | 10 m | GLOFs simulation | 4 |
| Bathymetry data | 2020-08-27 | 5 m | Morphology modeling of glacial lake and GLOFs simulation | 5 |
| Field photos | 2020-08-27 | 4000×6000 dpi | Analysis of the topographic parameters of glacial lake | 5 |
| SCGI | 1970-12 | 1:100000 | Glacier mapping reference | 6 |

1.  USGS (the United States Geological Survey): https://earthexplorer.usgs.gov/

2.  Esri ArcGIS Earth software: http://goto.arcgisonline.com/maps/World_Imagery

3.  NASA (the National Aeronautics and Space Administration) EARTHDATA: https://earthdata.nasa.gov/

4.  GLC10 LULC product: http://data.ess.tsinghua.edu.cn/fromglc10_2017v01.html

5.  Field measurement.

6.  Chinese National Cryosphere Desert Data Centre: http://www.ncdc.ac.cn

**3.2 Bathymetry and modeling**

Lake bathymetric information is one of the most important inputs in the dynamic modeling of GLOFs, which can accurately reflect the topography of the lake basin below the water surface and be used to calculate the potential flood volume released in different breach scenarios (Westoby et al., 2014). In this study, the depth data were obtained by a USV (APACHE 3) system, which consists of four main parts, i.e., the data acquisition module, the data acquisition module, the positioning and navigation

control module, and the power module (Li et al., 2021). The USV system has a draft of 10 cm, which is smaller than the inflatable kayak used in previous studies (Haritashya et al., 2018; Sattar et al., 2019, 2021). The D230 Single-Frequency Depth Sounder mounted on the USV is designed the measure range of 0.15~300 m with the depth resolution of 1 cm and the bathymetry error of ±1 cm+0.1%×h (water depth). The sounder can operate at 200 kHz and water temperature range of -



30°C~60°C, meanwhile a real-time kinematic system enables a precise positioning for the bathymetric position with the horizontal error of ±8 mm, the vertical error of ±15 mm and the directional error of 0.2˚ on the 1 m baseline. Field measurements were carried out on August 27, 2020. We designed four longitudinal routes and 13 transverse routes prior to the survey, along which the USV based measurement was conducted (Fig. 2). The maximum speed of USV can reach 8 m/s, our survey was conducted with a speed of 2 m/s for a total route of 22.58 km in the Bienong Co. Due to absence of any obstructions on the lake, such as ice or small islands, the high performance of the USV, and the real-time monitoring, the survey was accurately completed along the designed route. A total of 16,020 valid sounding points basically covering the entire glacial lake were measured, which well fulfilled the requirement of data density requirement to model the lake basin topography (Fig.2).

Bathymetric map was created within ArcGIS 10.4 software using natural neighbor interpolation algorithm (Thompson et al., 2016; Haritashya et al., 2018; Watson et al., 2018). In addition, Surfer software was used to simulate the 3D morphology of the Bienong Co's lake basin. Lake capacity can be understood as the volume of water storage below a certain water level, which is the volume between a certain spatial curved surface and a certain horizontal surface (Shi et al., 1991). In this study, the volume of Bienong Co was obtained by multiplying the depth data and map resolution (5 m) as Eq. (2):

$$V = \sum_{i=1}^{n} H_i \cdot \lambda \tag{2}$$

where $V$ is the volume (m³) of Bienong Co; $H_i$ is the depth (m) at $i\text{-}th$ pixel; $n$ is the number of the pixels in the lake area; $\lambda$ is the pixels resolution (m²) of the bathymetric map.

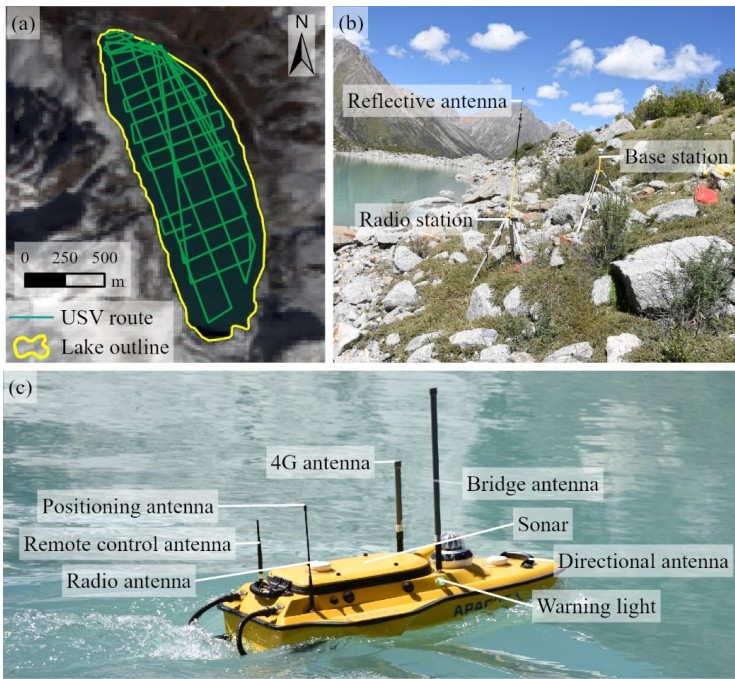

**Figure 2.** The bathymetry of Bienong Co. (a) The sampling path of USV on Bienong Co covering the base map of Landsat OLI pansharpened true color composite image, (b) the USV sampling equipment on land, and (c) in water. Photos taken by Xiaojun Yao on August 27, 2020.

### 3.3 Simulation of the potential GLOFs

### 3.3.1 DEM and LULC



DEM is an important data for glacier and glacial lake related research, for example the extraction of the surface elevation of glaciers and glacial lakes, the elevational parameters of dam, the cross-sections of river channel and so on. The Advanced Land Observing Satellite's (ALOS) mission Phased Array type L-band Synthetic Aperture Radar (PALSAR) yielded detailed
observation of all-weather from day to night and repeat-pass interferometry from 2006 to 2011 (Maskey et al., 2020). The radiometrically terrain-corrected elevation product ALOS PALSAR DEM with a spatial resolution of 12.5 m was released globally in October 2014 by the Alaska Satellite Facility (https://asf.alaska.edu/data-sets/derived-data-sets/alos-palsar-rtc/alos-palsar-radiometric-terrain-correction/) (Scatter et al., 2019) and was adopted in this study, which has been successfully applied in GLOF modeling previously in the Himalayas (Dhote et al., 2019; Sattar et al., 2019; Maskey et al., 2020).

Friction of the river channel to a given flow is determined by the Manning's roughness coefficient (Coon, 1998), which is dependent on the land use and land cover (LULC) of the modeling river channel in the study area. In this study, the GLC10 LULC product (http://data.ess.tsinghua.edu.cn/fromglc10_2017v01.html) with a spatial resolution of 10 m was used to obtain the value of Manning's $N$ in the flow channel.

### 3.3.2 Scenario scheduling

As a dangerous moraine-dammed glacial lake in SETP (Duan et al., 2020), Bienong Co may be struck by ice/snow avalanches, rock fall, landslides, heavy precipitation or earthquake and result in a GLOF event. Considering the impulse waves capable of initiating an overtopping failure of the frontal moraine caused by above factors, we assumed four scenarios based on different breach depths at the dam (Table 2). The worst scenario suffered the dam collapse from the existing outlet to the base of the moraine dam (72 m), other scenarios in which the breach height was reduced by half in turn (36 m, 18 m and 9 m). Based on
the measured bathymetry of Bienong Co, the released water volume ($V_w$) due to different breach height can be easily achieved. And then the average breach width ($W_b$) and the failure time ($T_f$) can be calculated using empirical relationships proposed by Froehlich (1995a) (Eq. (3) and Eq. (4)), which are the mostly used empirical approach for modeling earth-rock dam failures because of the high accuracy and low prediction error (Wahl, 2004).

$$W_b = 0.1803K_0(V_w)^{0.32}(H_b)^{0.19} \qquad (3)$$


$$T_f = 0.00254(V_w)^{0.53}(H_b)^{-0.9} \qquad (4)$$

where, $V_w$ is the released water volume, and $H_b$ is the breach height. All scenarios were modeled as a sine wave progressive breach model where the initial breach forms slowly and accelerates as the outflow velocity and shear stress increase through the breach (Sattar et al., 2021).

By inputting the above breach parameters to the MIKE 11 hydrodynamic model, a hydrography of breach can be obtained.
Froehlich (1995b) proposed an empirical formula to estimate the peak discharge ($Q_{max}$) of output flow (Eq. (5)), which was used to verify results from MIKE 11 model.

$$Q_{max} = 0.607V_w^{0.295}H_b^{1.24} \qquad (5)$$

**Table 2.** Details of breach parameters for different GLOFs scenarios, including the calculated discharge volume (m$^3$) based on bathymetry data and the peak discharges (m$^3$s$^{-1}$) of the breach hydrograph from MIKE 11 model and empirical formula
(Froehlich (1995b)).

| Parameters | Scenario-1 | Scenario-2 | Scenario-3 | Scenario-4 |
|---|---|---|---|---|
| Breach height ($H_b$) (m) | 72 | 36 | 18 | 9 |
| Breach width ($W_b$) (m) | 180 | 131 | 94 | 66 |
| Time of failure ($T_f$) (h) | 0.75 | 1.03 | 1.37 | 1.79 |



| Discharged volume ($V_w$) (×$10^7$ m³) | 6.52 | 3.67 | 1.93 | 0.99 |
| Percentage of discharged volume (%) | 64 | 36 | 19 | 10 |
| Peak discharge (MIKE 11model) (m³ s⁻¹) | 26721 | 11126 | 3716 | 1294 |
| Peak discharge (empirical formula) (m³ s⁻¹) | 24630 | 8801 | 3081 | 1070 |

### 3.3.3 GLOFs modeling

MIKE 11 is a professional engineering software package developed by DHI Water and Environment in 1987, which has powerful capabilities for the numerical simulation of rivers and the replication and calculation of dam breaching processes based on an implicit, finite difference computation of unsteady flows (DHI, 2007). The software has been successfully used for GLOF modeling in Himalayan basins (Jain et al., 2012; Aggarwal et al., 2013; Lohani and Jain, 2016; Thakur et al., 2016). MIKE 11 dam breach model is composed of river channels, reservoirs, dam break structures, etc., in which, the river is represented by cross-sections at regular intervals. In this study, glacial lake was represented as a dam breach structure which includes a reservoir with water level-area relationship, and the breaching parameters of breach depth, breach width, and failure time. Then, the unsteady flow simulations were carried out based on the preset four scenarios and computational interval with dam failure mode of overtopping. Finally, the resulting outflow hydrological curves at breach were used as the upper boundary condition of the two-dimensional model MIKE 21 (DHI, 2007).

MIKE 21 Flow Model is a modeling system for 2D free-surface flows, which is applicable to the simulation of hydraulic and environmental phenomena in lakes, estuaries, bays, coastal areas and seas (DHI, 2007). MIKE 21 model was used to simulate the dynamic routing of the initial breach hydrography along the flow channel from Bienong Co to the convergence with Song Qu (Fig.1). The inputs to the two-dimensional model for an unsteady hydraulic simulation includes terrain data and boundary conditions. The terrain data was represented by a 2D mesh covering the entire flow area, which was obtained from ALOS PALSAR DEM. The unstructured mesh of MIKE 21 is an approximately equilateral triangular mesh with a cell-centered finite volume solution, which simulates the flow field in the area around the river bend and the structure over water excellently. The size of the mesh can be adjusted, namely the focal areas can be encrypted. In this study, the 2D mesh has an individual cell area of 1000 m² in main flow channel and the 10000 m² in other regions. Each cell was defined with a Manning's $N$ value and the topographic elevation. The upstream boundary of the two-dimensional model is the outflow hydrographs at the immediately downstream of moraine dam derived from the MIKE 11 model, and the downstream boundary is the lowest elevation of the terminal of simulated flow channel. The two-dimensional dynamic modeling is solved by the depth-averaged shallow water equations. Furthermore, the significant flood wave parameters like inundation area, discharge, flow depth, flow velocity and arrival time of flood were analyzed to evaluate the potential GLOFs hazard along the flow channel, with the focused attention on the 19 settlements, 13 bridges and Jiazhong Highway.

## 4 Results

### 4.1 Evolution of Bienong Co and the mother glacier

Bienong Co is a stable glacial lake, which only experienced an expansion of about 120 m towards the Mulang Glacier from 1976 to 1988 (maybe earlier), and has remained stable since then (Fig. 3) because it has reached the ice cliff of Mulang Glacier and there is no room for expansion (Fig. 1 and Fig. 3). The same is true for the area of Mulang Glacier, which has remained largely unchanged area over the last 45 years. However, the ice thickness of the whole glacier and the glacier ablation zone has thinned, with the decreasing rate of -0.79 m/a and -6.54 m/a, showing a negative mass balance in the context of climate warming (Fig. 4).






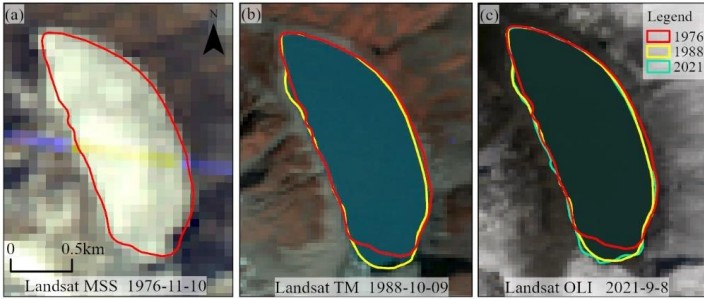

**Figure 3**. The evolution of Bienong Co from 1976 to 2021.

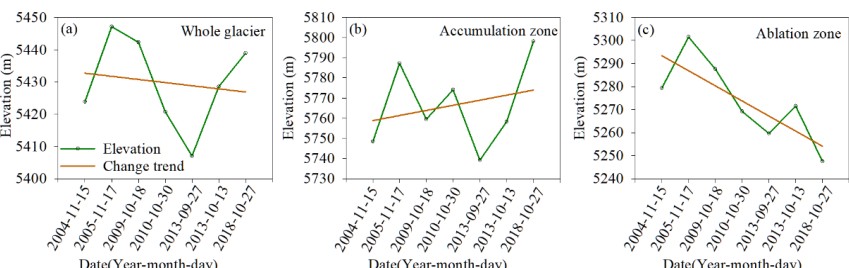

**Figure 4**. The mean elevation changes of Mulang Glacier for (a) the whole glacier, (b) the accumulatio zone, (c) the ablation
zone. The glacier accumulation and ablation zones were divided based on the elevation of the median glacier area (Guo et

al., 2015).

**4.2 Morphology and volume estimation of Bienong Co**

The basin morphology of Bienong Co was modeled based on the TIN grid created by the field depth data (Fig. 2). Apparently,
this lake has a relatively flat basin bottom and both deep flanks (Fig. 5). Similar to most glacial lakes (Yao et al., 2012; Zhou
et al., 2020), the slope of the lake shores near the glacier is steeper than that near the moraine dam. The water depth profile

from moraine dam to mother glacier show that the depth of the lake reaches a maximum of 180 m at about 1000 m from the
moraine dam, corresponding to the slope of 11.3˚. The depth keeps stable at distance of 1000 m to 1500 m from the moraine
dam, and the distance from the mother glacier to the deepest point of the lake is 600 m with the slope of 16.5˚. A depth profile
facing the moraine dam from the left bank to the right bank shows that the left side is steeper than the right side. The glacial
lake reaches its deepest point at 200 m from left shore with the slope of 43.4˚, then maintains flat to 430 m, and the distance

between the bottom and right shore is 273 m with the slope of 32˚. The volume of Bienong Co was calculated using the surface
elevation and the lake bed derived from the TIN grid, which was about $10.2 \times 10^7 \, \text{m}^3$ in 2020, which is a generally accurate
estimate of the magnitude of this moraine-dammed lake.





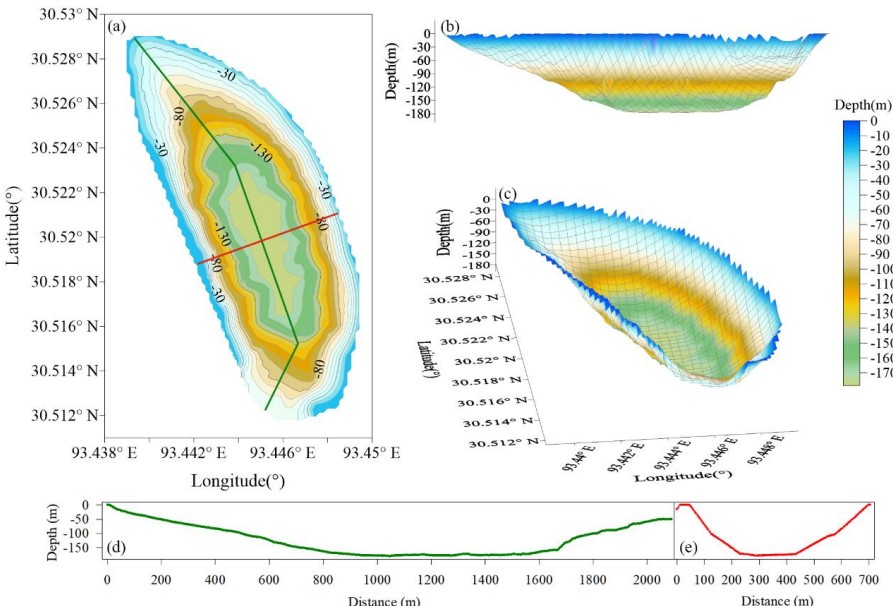

**Figure 5**. Morphology modeling of Bienong Co in 2020 and the equal-scale profiles of distance and depth from the moraine
dam to Mulang Glacier (green line) and from the left shore to right shore (red line).

### 4.3 GLOFs simulations

#### 4.3.1 Moraine-breach and inter-comparison of GLOF discharge

Based on the current area and dam status of Bienong Co, four hypothetical dam failure scenarios were modelled by different
combinations of $H_b$, $W_b$ and $T_f$. Without further consideration of the physics and dynamics of the breach erosion process, each
potential breach event yielded the different $Q_{max}$ based on the given breach parameters (Table 2). In the extreme-magnitude
scenario, the breach was assumed to cut into the base of the dam with the heigh of 72 m (Fig. 6), which is rare in nature and is
largely determined by the condition of the moraine dam and the triggers of the breach, but may occur, for example, under the
action of a strong earthquake. In this case, the Bienong Co can drain about $6.52 \times 10^7$ m$^3$ of $V_w$, accounting for 64% of the total
$V_w$. The breach width and time of failure based on the formulas of Froehlich (1995a) are 180 m and 0.75 hours. Under these
conditions, a sharp rise in the outflow hydrography can be caused, with the empirical formula (Froehlich, 1995b) and MIKE
11 model producing similar $Q_{max}$ of 24,630 m$^3$ s$^{-1}$ and 26,721 m$^3$ s$^{-1}$, respectively, at immediately downstream of the lake (Table
2). In the high- magnitude scenario, the released $V_w$ is $3.67 \times 10^7$ m$^3$, accounting for 36% of the total lake $V_w$ (Table 2). Within
1.03 hours of the breach, the $W_b$ reached a maximum of 131 m and the flooding peaked at 11,126 m$^3$ s$^{-1}$ from MIKE 11 model,
which is 8,801 m$^3$ s$^{-1}$ obtained from the empirical formula and lower 24% than the former (Fig. 6). The moderate-magnitude
scenario is constrained by the condition of $H_b$ in 18 m, $W_b$ in 94 m and $T_f$ in 1.37 hours, resulting in a release of water volume
$1.93 \times 10^7$ m$^3$ (19% of the total volume), and the breach peak of 3,716 m$^3$ s$^{-1}$ from MIKE 11 model and 3,081 m$^3$ s$^{-1}$ from
empirical formula. The low-magnitude scenario ($H_b$: 9 m, $W_b$ :66 m and $T_f$: 1.79 h) drained the $V_w$ of $0.99 \times 10^7$ (10% of the total
$V_w$) with the $Q_{max}$ of 1,294 m$^3$ s$^{-1}$ based on MIKE 11 model and 1,070 m$^3$ s$^{-1}$ from empirical formula. This is the most
conservative situation and the one that is most likely to happen.

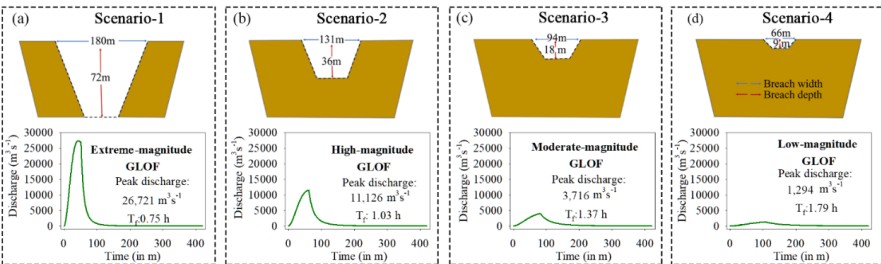


**Figure 6**. Schematic diagram of the breach width and breach depth (from Froehlich, 1995a) for different scenarios of GLOFs and the corresponding outflow hydrographs, (a) the Scenario-1, (b) Scenario-2, (c) Scenario-3 and (d) Scenario-4 reapectively leads to the extreme-magnitude, high-magnitude, moderate-magnitude and low-magnitude GLOFs.

### 4.3.2 Hydraulic characterization of GLOFs along the flow channel

The outflow hydrographs at the immediately downstream of moraine dam for four different scenarios were used as the upper boundary for the two-dimensional MIKE 21 model to simulate the hydraulic behaviors of the GLOFs from the lake to the convergence with Song Qu (Table 3, Fig.6), located at a distance of ~52.98 km downstream. Where the inundation area, $Q_{max}$, maximum flow depth ($D_{max}$) and maximum flow velocity ($V_{max}$) of each individual scenario were evaluated at the 19 settlements along the flow channel (Fig.1). Due to the large magnitude of the GLOF flows, any additional flow added by existing stream

flows in the river channel were considered negligible.

The calculated maximum inundation areas along the given valley from Bienong Co to the ~52.98 km downstream are 13.05 km$^2$, 10.25 km$^2$, 8.32 km$^2$ and 6.64 km$^2$ for the extreme to low magnitude scenarios, with the average $D_{max}$ of 18.31 m, 10.75 m, 6.76 m and 4.41 m, as well as the average $V_{max}$ of 22.44 m s$^{-1}$, 7.63 m s$^{-1}$, 4.78 m s$^{-1}$, 2.92 m s$^{-1}$, respectively (Fig.7, Fig.8 and Fig.9). GLOFs of different magnitudes will pose different potential hazards to each settlement along the flow channel.

In the extreme-magnitude GLOF, the $Q_{max}$ reaches the nearest settlement S1, 3.18 km downstream from Bienong Co, 47 minutes after the breach occurred, which would be almost completely submerged with a $V_{max}$ of 44.49 m s$^{-1}$ and a $D_{max}$ of 10.46 m (Table 3). The $Q_{max}$ at the cross-section passing through S1 can reach up 31,091 m$^3$ s$^{-1}$ (Table 3, Fig.10). The Yilongduo village is located on the right bank of the river channel 4.81 km downstream of Bienong Co, which would be completely inundated by the extreme-magnitude GLOF, with the $Q_{max}$ of up to 38,301 m$^3$ s$^{-1}$, arriving approximately 49 minutes after the event. The $D_{max}$ and $V_{max}$ are 6.89 m and 50.55 m s$^{-1}$, respectively (Table 3, Fig. 7, Fig. 8, Fig. 9 and Fig. 10). The next villages

Jiawu (5.44 km: the distance from Bienong Co), Ang'na (5.98 km), Wa'na (8.52 km), Qiangjiuci (9.81 km), and Bula (11.43 km) all would be fully inundated by the extreme-magnitude GLOF, with the $D_{max}$ severe to over 10 m (Table 3 and Fig. 8). The Buda village (12.99 km) is unique in that it is not located in the flow channel but at left bank of the upper tributary of Xiong Qu before it converging with the flow channel downstream the Bienong Co (Fig.1). The super flood can cross the

highland on the left bank of the flow channel, enter the upstream channel of Xiong Qu and flood Bada village with a $D_{max}$ of 4.81 m, but the region near the mountain can spare from the flooding (Table 3, Fig. 7 and Fig. 8). Haqing village, located at the 17.8 km downstream of Bienong Co, will suffer the most severe damage with a $D_{max}$ of 18.29 m comparing former villages due to its low elevation (Table 3 and Fig. 7). The next Zongri village and Dama village will both be partly flooded, but the Darzi village is fortunate enough to be spared from any flooding (Table 3 and Fig. 7). For the remaining seven villages, Zhibu,

Qiwuxing, Kemaluo, Xinka-1, Xingka-2, Xinkamaluo and Dading, none of them were spared from being completely submerged by the extreme-magnitude GLOF (Table 3 and Fig. 7). However, due to the far distance from the disaster source, they have an advantage of the later arriving of flood, thus have more time to escape. In the extreme-magnitude GLOF event, almost the entire range of the Jiazhong Highway will be disturbed by flood, and all bridges along the route will be inundated and impacted.



The extreme-magnitude GLOF released about 63.75% of $V_w$ in Bienong Co, and the water body proceeds at high speed and energy in the valley, dealing a devastating blow to all man-made elements. However, this scenario rarely occurs unless an extreme earthquake causes the complete collapse of moraine dam. More common is the GLOFs caused by partial collapse of moraine dam. In the simulated high-magnitude GLOF scenario, villages of Jiawu and Zongri will be spared from flooding, and villages of Settlement-1, Yilongduo, Dama and Xinka-1 will be partially flooded. The remaining villages will still be

completely flooded, but the $D_{max}$ will be reduced by about half comparing to the extreme-magnitude GLOF. In the medium-magnitude GLOF, the number of villages safe from GLOF will increase to six, they are Yilongduo, Jiawu, Bada, Zongri, Dazi, and Qiwuxing. The number of villages partially affected by flooding increased to nine, and only four villages will be fully submerged by flooding. They are Kemaluo, Xinka-2, Xinkamaluo and Dading, which all located in the downstream area of the simulated flow channel. In the low-magnitude GLOF, nine villages will avoid damage from flooding, seven villages will

suffer partial influence, and only three villages at the downstream will be fully flooded, but flood will come within 4-7 hours and people can adequately avoid it. In this scenario, half of the Jiazhong Highway will be affected by flooding, but the bridges are not spared because they are located in the middle of the river, and the simulated inundation area basically covers the entire river (Fig.7). Although the simulated low-magnitude GLOF was based only on a moraine dam drop of 9 m and about 9.64% corresponding to $0.99\times10^7$ m³ of the $V_w$ in Bienong Co was released, which equal to the amount of $V_w$ released by the 2020

GLOF at Jinwu Co that caused severe destruction of infrastructure (roads and bridges) and property losses in downstream areas (Zheng et al., 2021). Therefore, the other simulated magnitude-GLOFs in this study represent the more severe scenarios that could happen. In addition, it's worth mentioning that the above GLOFs scenario has not been verified by field measured data, but is only a hypothetical state. The quality of the DEM data and the computational mechanism of the MIKE model have a strong influence on the accuracy of the simulation results. However, this study is still valuable as a reference for potential

GLOFs hazards of this highly dangerous glacial lake.

**Table 3** Flow hydraulics at different sites along the flow channel.

| Village | Distance (km) | Scenario-1 | | | | Scenario-2 | | | | Scenario-3 | | | | Scenario-4 | | | |
|---|---|---|---|---|---|---|---|---|---|---|---|---|---|---|---|---|---|
| | | $Q_{max}$ (m³s⁻¹) | $D_{max}$ (m) | $V_{max}$ (ms⁻¹) | $Q_{max}t$ (h) | $Q_{max}$ (m³s⁻¹) | $D_{max}$ (m) | $V_{max}$ (ms⁻¹) | $Q_{max}t$ (h) | $Q_{max}$ (m³s⁻¹) | $D_{max}$ (m) | $V_{max}$ (ms⁻¹) | $Q_{max}t$ (h) | $Q_{max}$ (m³s⁻¹) | $D_{max}$ (m) | $V_{max}$ (ms⁻¹) | $Q_{max}t$ (h) |
| S1 | 3.18 | 31091 | 10.46* | 44.69 | 0.78 | 13792 | 4.47* | 36.71 | 1.03 | 5081 | 2.27* | 6.75 | 1.32 | 1408 | NR | NR | 1.82 |
| Yilongduo | 4.81 | 38301 | 6.89 | 50.55 | 0.82 | 12033 | 2.52* | 3.48 | 1.13 | 5198 | NR | NR | 1.52 | 1828 | NR | NR | 1.95 |
| Jiawu | 5.44 | 39972 | 11.09 | 65.71 | 0.84 | 11643 | NR | NR | 1.18 | 4331 | NR | NR | 1.50 | 1477 | NR | NR | 1.85 |
| Ang'na | 5.98 | 37701 | 10.28 | 7.07 | 0.85 | 11670 | 4.22 | 2.50 | 1.12 | 4040 | 1.78* | 2.76 | 1.45 | 1351 | NR | NR | 1.98 |
| Wa'na | 8.52 | 30384 | 14.31 | 9.15 | 0.87 | 11111 | 7.21 | 4.71 | 1.22 | 3833 | 5.81* | 2.59 | 1.60 | 1262 | 2.99* | 0.53 | 2.10 |
| Qiangjiuci | 9.81 | 27736 | 11.23 | 27.68 | 0.97 | 11843 | 5.71 | 13.16 | 1.28 | 3811 | 3.71* | 10.02 | 1.67 | 1353 | 1.82* | 6.81 | 2.23 |
| Bula | 11.43 | 28758 | 16.37 | 13.38 | 1.02 | 10948 | 8.77 | 8.54 | 1.28 | 3756 | 5.34* | 6.12 | 1.72 | 1239 | 4.19* | 3.04 | 2.22 |
| Bada | 12.99 | 2260 | 4.81* | 0.26 | 1.03 | 886 | 1.05* | 0.03 | 1.29 | NR | NR | NR | NR | NR | NR | NR | NR |
| Haqing | 17.85 | 23908 | 18.29 | 9.41 | 1.23 | 9947 | 8.48 | 8.37 | 1.55 | 3662 | 5.22* | 7.11 | 2.03 | 1241 | 3.62* | 3.95 | 2.68 |
| Zongri | 20.42 | 22992 | 6.19* | 18.47 | 1.30 | 9751 | NR | NR | 1.63 | 3556 | NR | NR | 2.18 | 1237 | NR | NR | 2.87 |
| Dazi | 24.00 | 22409 | NR | NR | 1.45 | 9266 | NR | NR | 1.82 | 3422 | NR | NR | 2.38 | 1172 | NR | NR | 3.18 |
| Dama | 30.19 | 20463 | 11.63* | 5.15 | 1.68 | 8545 | 7.93* | 3.93 | 2.12 | 3260 | 4.42* | 2.87 | 2.73 | 1119 | 2.69* | 2.14 | 3.65 |
| Zhibu | 32.24 | 19941 | 18.85 | 12.74 | 1.75 | 8510 | 10.31 | 9.14 | 2.27 | 3192 | 6.73* | 6.54 | 2.85 | 1106 | NR | NR | 3.78 |
| Qiwuxing | 33.41 | 18954 | 12.47 | 4.42 | 1.83 | 82210 | 3.64 | 1.52 | 2.27 | 3145 | NR | NR | 2.92 | 1092 | NR | NR | 3.88 |
| Kemaluo | 37.89 | 13849 | 23.36 | 3.49 | 2.20 | 5413 | 14.84 | 2.31 | 2.75 | 216 | 7.05 | 1.84 | 3.52 | 799 | 5.48 | 0.64 | 4.63 |
| Xingka-1 | 43.54 | 11933 | 19.83 | 3.52 | 2.48 | 4802 | 12.06* | 2.40 | 3.28 | 1915 | 5.72* | 2.08 | 4.15 | 748 | 4.30* | 1.26 | 5.42 |
| Xingka-2 | 43.98 | 11685 | 16.43 | 6.01 | 2.73 | 4742 | 10.47 | 3.87 | 3.38 | 1867 | 6.70 | 2.16 | 4.18 | 736 | 3.60* | 0.90 | 5.48 |
| Xinkamaluo | 45.89 | 10863 | 19.01 | 4.66 | 2.72 | 4399 | 11.27 | 2.94 | 3.48 | 1715 | 6.72 | 2.08 | 4.57 | 681 | 4.24 | 1.33 | 5.75 |
| Dading | 50.91 | 7329 | 15.99 | 5.15 | 3.20 | 3100 | 9.76 | 3.11 | 4.12 | 1157 | 5.84 | 1.95 | 5.15 | 455 | 3.56 | 1.20 | 6.35 |

**Note:** NR means that the flood waters did not reach here, * means that only part of the area is flooded.





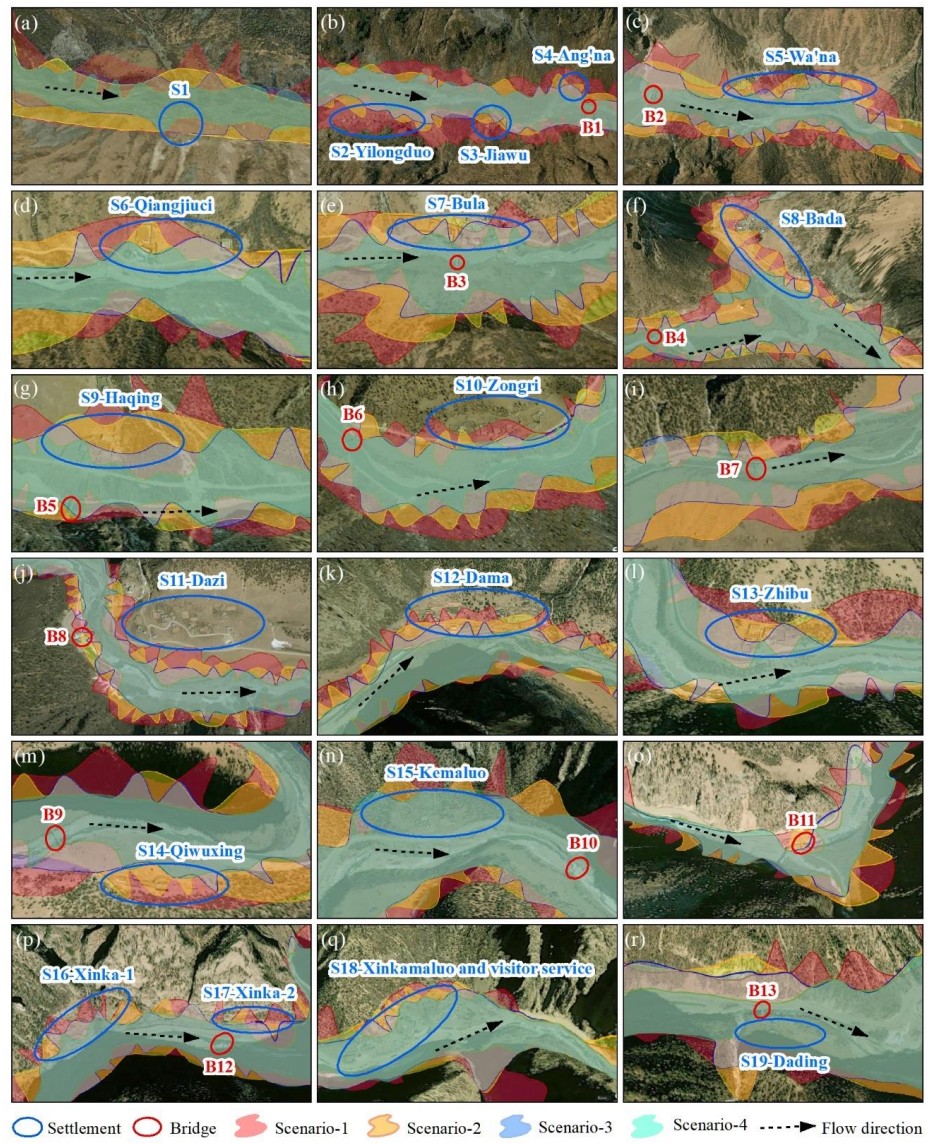

**Figure 7**. High-resolution images (ArcGIS Earth) showing the potential inundation extent at each downstream settlement along

the flow channel (locations can see in Fig.1) for different scenarios caused GLOFs.





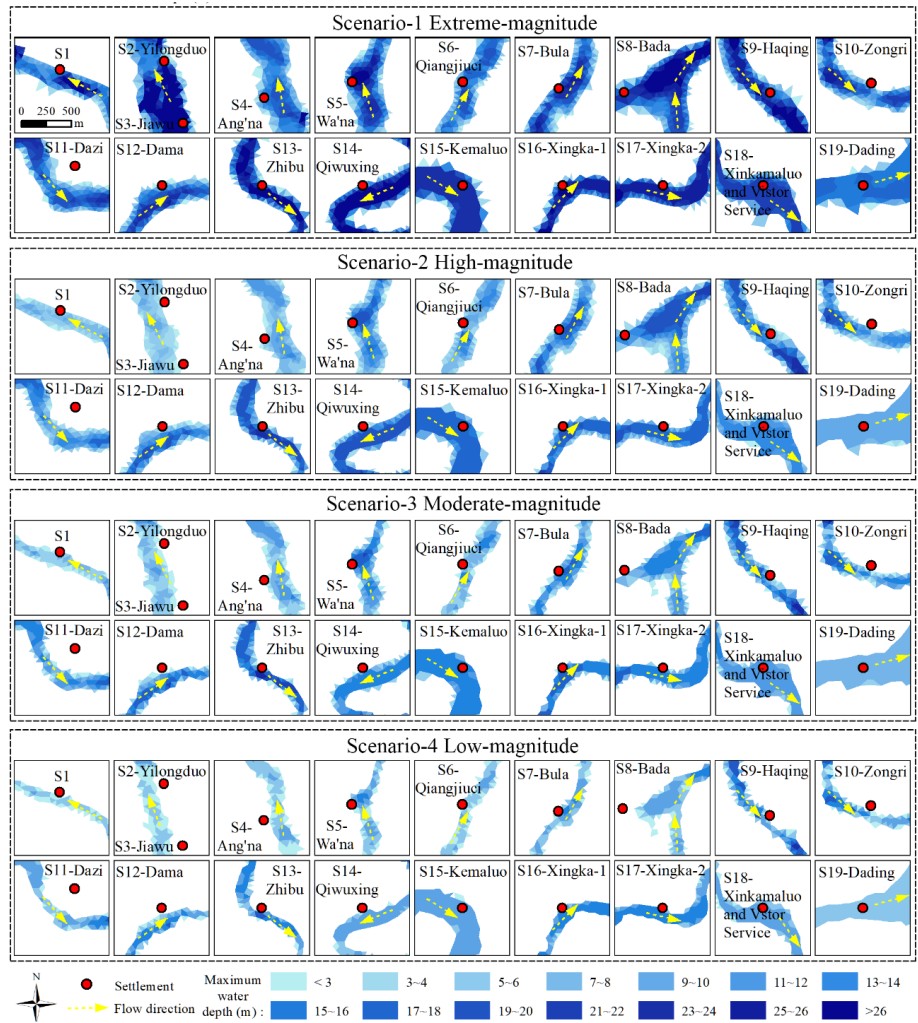

**Figure 8**. Spatially distributed flow depths of scenario-1, scenario-2, scenario-3, and scenario-4 at each settlement along the flow channel (locations can see in Fig.1).



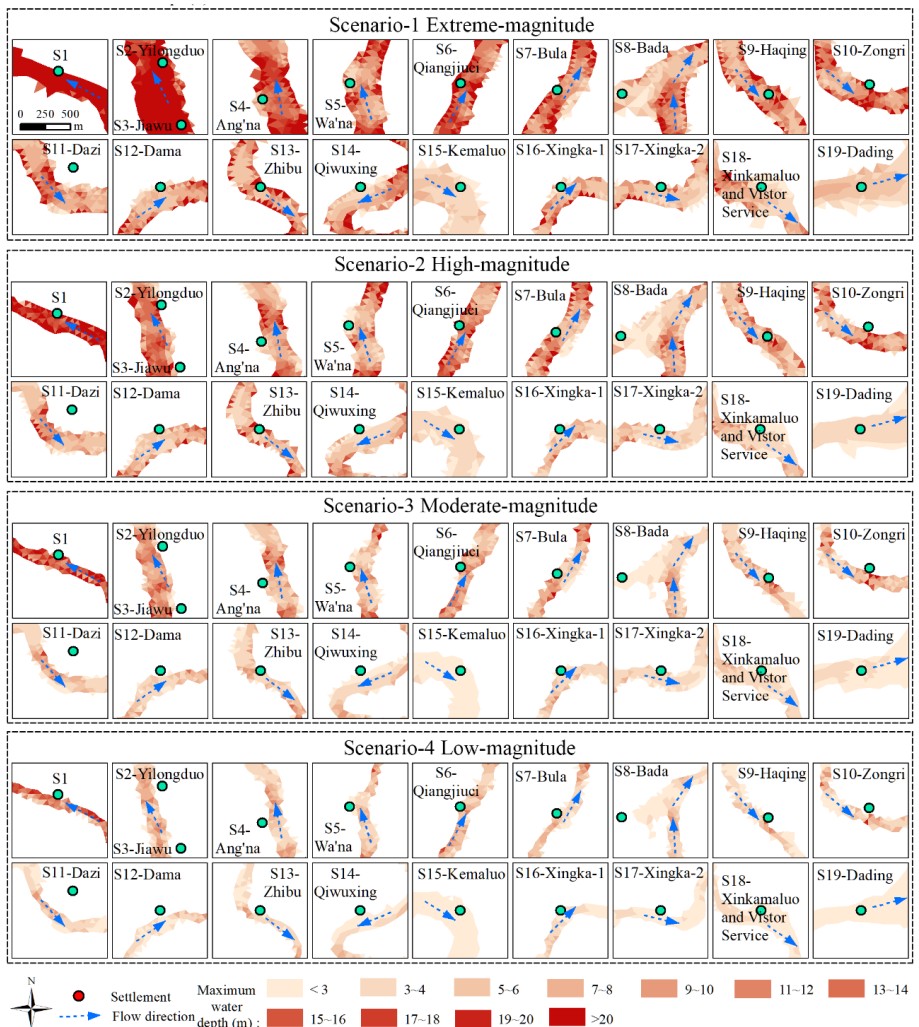

**Figure 9.** Spatially distributed flow velocities of scenario-1, scenario-2, scenario-3, and scenario-4 at each settlement along the flow channel (locations can see in Fig.1).

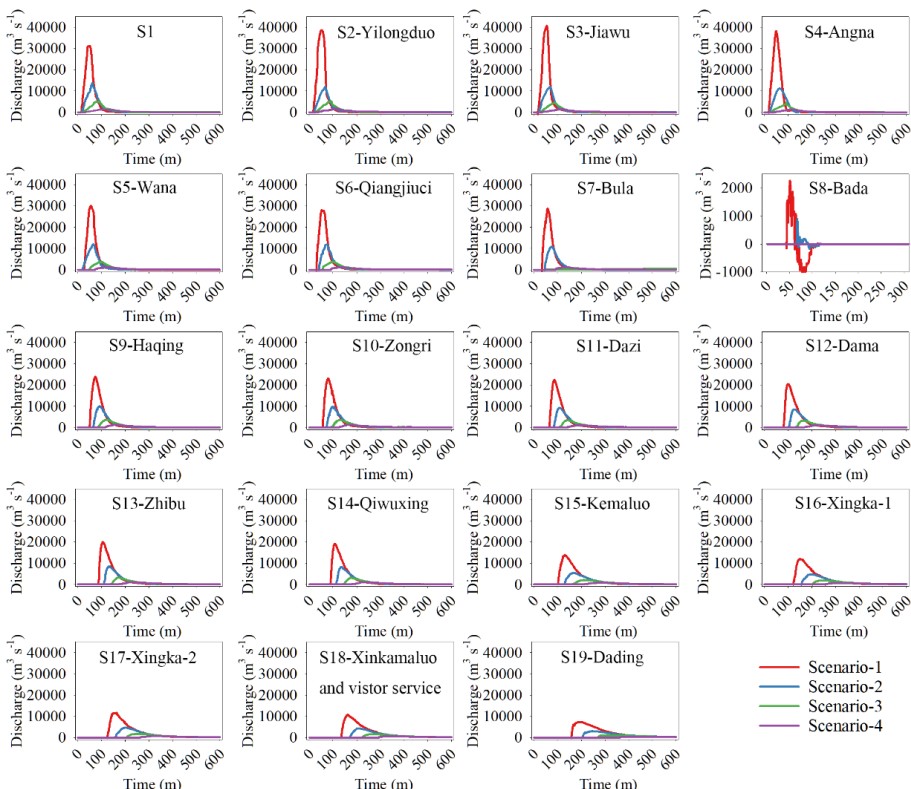

**Figure 10.** Temporal course of discharge at different settlements along the flow channel (locations can see in Fig.1) of four scenarios of GLOFs.





## 5 Discussion

### 5.1 Potential GLOF trigger assessment

Generally speaking, conditions required for glacial lake outburst can be divided into two aspects: the external conditions, that is, the presence of dynamic environments that promote glacial lake outbursts; the inherent conditions that the features of the glacial lake itself are conducive to the occurrence of outburst (Wang, 2016).

### 5.1.1 Inherent conditions

Thickness of mother glacier and the topography beneath it are critical to the further expansion of a glacial lake, and a glacial lake connected to a flat glacier tongue portion of the mother glacier has the potential for continued expansion in the future (Allen et al., 2019; Lala et al., 2018). However, Bienong Co does not have such a condition, because since the 1988 or earlier, it has reached the steep bedrock of its mother glacier, therefore, has the little space to extend due to topographic constraints (Fig. 3 and Fig. 11). Additionally, an outlet on the top of moraine dam allows for continuous drainage of the lake, which contributes to its stability (Fig. 11). And remote sensing images proves that Bienong Co has maintained the current area for a long time (Fig.3), that is to say, the current moraine dam can withstand the pressure generated by the water volume. However, the hazard of Bienong Co cannot be ignored due to its area ($1.15 \text{ km}^2$) that much larger than the three breached glacial lakes (i.e., Coga: $0.42 \text{ km}^2$ (Yao et al., 2014), Ranzeria Co: $0.58 \text{ km}^2$ (Sun et al., 2014) and Jinwu Co: $0.58 \text{ km}^2$ (Zheng et al., 2021)) in this region and other outburst triggers.

Dam characteristics, such as dam geometry (dam width to height ratio, width of crest, dam distal face slope, freeboard), dam material properties, ice-cored moraine conditions govern the stability of the dam (Huggel et al., 2004; Prakash and Wang et al., 2011a; Nagarajan, 2017). Bienong Co is constrained by the end moraine, composed of loose sand and gravel (Table 4 and Fig. 11), which has poor coagulability and stability. However, no ice body was observed from the exposed outlet section (Table 4 and Fig. 11), indicating the impossibility of a disaster caused by the collapse of the dam due to ice melting. The moraine dam is 550 m wide and the height is variable with an average height of 72 m and the width-height ratio of 7.64. According to the thresholds favoring GLOFs of dam width smaller than 60 m proposed by Lv et al. (1999), width-height ratio smaller than 0.2 proposed by Huggel et al. (2004), the moraine dam of Bienong Co is generally stable (Table 4 and Fig. 11). However, mean freeboard of 10 m and the distal facing slope of 35° are the conditions conducive to GLOFs based on the favoring thresholds of smaller than 25 m (Mergili et al., 2011) and larger than 20° (Lv et al., 1999) (Table 4 and Fig. 11).

**Table 4** The morphometric status of Bienong Co

|  | Lake characteristics | Morphology |  | Lake characteristics | Morphology |
|---|---|---|---|---|---|
| **Glacial lake** | Type | Proglacial lake | **Mulang Glacier** | Area | 8.29 km² |
|  | Area | 1.15±0.05 km² |  | Average surface slope | 18.28° |
|  | Length | 2 km |  | Height of glacier cliff | 122-186 m |
|  | Water surface elevation | 4745 m | **Moraine dam** | Type | Moraine |
|  | Facing direction | Northwest |  | Width of crest | 550 m |
|  | Maximum depth | 181.04 m |  | Mean height of crest | 72 m |
|  | Average depth | 85.40 m |  | Width-height ratio | 7.64 |
|  | Snow/avalanche site |  |  | Mean freeboard | 10 m |
|  | Outlet condition | Free flow |  | Distal facing slope | 35° |
|  | Contact with mother glacier | Yes |  | Ice core | No |

**Note:** The area of Bienong Co and Mulang Glacier are derived from a scene of Landsat OLI image on September 18, 2021, and the elevation and slope are measured based on ALOS PALSAR DEM.



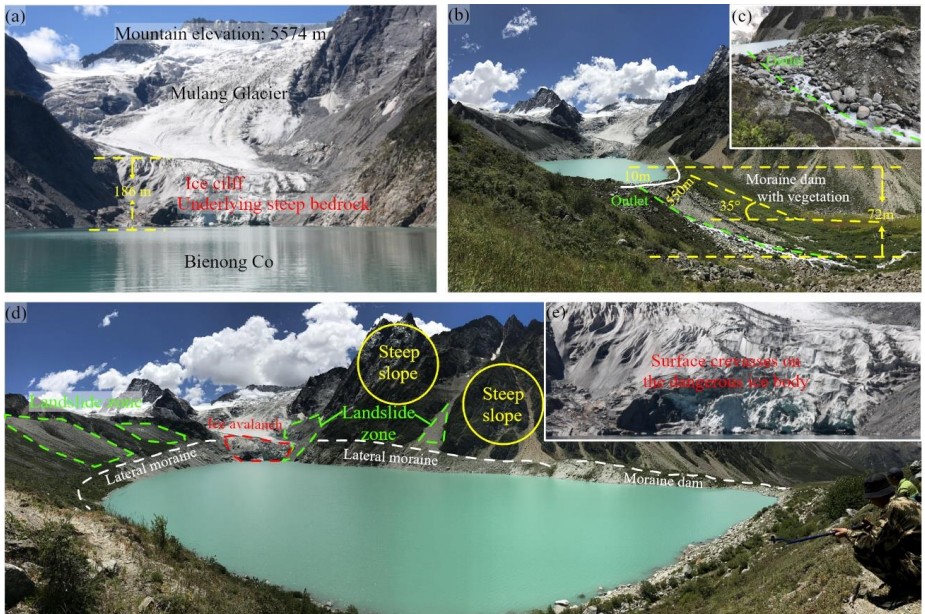

**Figure 11**. The hazards assessment of Bienong Co. (a) The connection condition of Mulang Glacier and Bienong Co, (b) and (c) the moraine dam condition of Bienong Co, (d) and (e) the external conditions of Bienong Co. Photos taken by Xiaojun Yao on August 27, 2020.

### 5.1.2 External conditions

Studies show that GLOFs can be triggered by several factors, mass movement such as landslides, ice avalanches and rock
slides are considered to be the most important triggers in the GLOF hazard chain (Worni et al., 2014), which can cause overtopping (Risio et al., 2011). Field investigation shows that there are many granular sandy landslides around Bienong Co (Fig. 11), which is a potential trigger for disaster. The GLOF of Jinwu Co, a moraine dammed glacial lake located near the Bienong Co, was caused by an initial landslide on the left side (Zheng et al., 2021). Additionally, the steep lateral moraine and slopes that lie 45°-60° are deemed as to conducive landslides and snow avalanching (Rounce et al., 2016; Sattar et al., 2021),
which exist in at least two locations around the Bienong Co. One more current concern is that the Mulang Glacier has been directly connected to the Bienong Co from the time of its area remained stable (Fig. 3), instead of separating from the mother glacier as other glacial lakes, such as Jialong Co (Li et al., 2021). The tongue of Mulang Glacier is currently steep and dangerous due to the extensive surface crevasses (Fig. 11), which makes it highly susceptible to outburst floods because of the highly exposure of the lake to ice avalanching and glacier calving. And in the context of global warming, the glacial melting
water will lubricate the glacier itself, so the hanging ice is prone to slide into the lake (Wang et al., 2015), which may induce surge waves capable of overtopping terminal moraine structures. Plenty of studies documented that most of the moraine dam failure on the Tibetan Plateau were triggered by overtopping waves generated from ice avalanches (Wang et al., 2015). The elevation of the top of the hazardous ice body from the lake surface was measured to be about 122-186 m, with an area of about 0.16 km$^2$ using a boundary where the crevasse terminates and the slope changes significantly (Fig. 11). If some or all of
the ice body fall into the glacial lake under extreme weather or seismic conditions, the pressure caused by its stirring waves will produce a fatal blow to the moraine dam, reducing the stability of the glacial lake dam and most likely causing the collapse of dam and the occurrence of GLOFs.

In addition to the impact of material movement on the dam, the continuous rise of water level caused by constant summer



meltwater or concentrated precipitation will also have an impact on the dynamics at the outlet of the glacial lake, increasing
the scouring of the breach in extreme cases, leading to an increase in the size of the breach and triggering the occurrence of
glacial lake outburst floods (Clague and Evans, 2000; Worni et al., 2012; Allen et al., 2015). The elevation difference between
the glacial lake surface and the outside of the moraine dam is 80 m (Fig. 11), which creates a steep spillway that is prone to
erosion of the outlet.

The potential triggers mentioned above are speculative conclusions based on already disrupted glacial lakes and current
experience, and are hypothetical and probabilistic statements that will not necessarily occur. And with the current drastic
changes in the external environment such as earthquakes and climate change, the above various characteristics of Bienong Co
will be affected. In the future, the ground ice conditions, dam parameters and moraine dynamics of the glacial lakes should be
further investigated to make more detailed simulations and early warnings of glacial lake outburst floods. Furthermore,
earthquake is a non-ignorable factor, although is of the small occurrence probability, the consequences can be catastrophic
once it happens. The extreme scenario of the simulated GLOF with breach down to dam bottom in this study is motivated by
this assumption.

Although factors that trigger a glacial lake to produce an outburst are divided into external and internal conditions, external
conditions often cause an imbalance state of the moraine-dammed glacial lake, and many GLOFs are often the result of one
trigger that stimulates changes in other factors, or a combination of factors (Zheng et al., 2021). For example, the Tam Pokhari
Glacial Lake in Nepal on September 3, 1998 was damaged by the resultant internal and physical properties of the dike under
the combined effect of external forces such as heavy precipitation in the lake basin, earthquake, and snow/ice avalanche, and
the dike was eventually breached due to a decrease in water resistance stress (Dwivedi et al., 2000; Osti et al., 2011). Whereas
the Jinwu Co that occurred within this watershed in 2020 was a pre-emptive weakening of the dam due to increased flows
from landslides, which may have facilitated erosion of the lake by subsequent rainfall, snowmelt, and/or smaller secondary
landslide or avalanche events (Liu et al., 2021; Zheng et al., 2021).

## 5.2 Comparison of morphology characteristics for glacial lakes in continental and maritime glaciation regions

Accurate information of lake-bottom topography and dimensions of the associated moraine is essential to evaluate the actual
mechanism of a glacial lake outburst, thereby defining the actual total volume of water that will potentially be released in a
failure event, which is also one of the important inputs in the dynamic model of GLOFs (Westoby et al., 2014). However, there
are very few glacial lakes with field bathymetric data. Based on the published data, bathymetric data of a total of 16 glacial
lakes distributed in Himalayas including Tibet of China, India, Nepal and Bhutan were collected in this study (Table 5).
Glaciers can be divided into continental type and maritime type according to the difference in physical properties (Xie and Liu,
2010), and the latter has a stronger geological and geomorphological effect than the former (Qin et al., 2007; Liu et al., 2014).
In the Himalayas, glaciers on the northern aspect of the central section are of continental type, whereas those on the eastern
section and the southern aspect of the central section are of maritime type (Qin et al., 2007; Liu et al., 2014). Glacial lakes are
produced by glacial action, and the great difference in physical properties between continental and maritime glaciers inevitably
has an impact on the lake basin's morphology. Theoretically, glacial lakes produced by maritime glaciers could have deeper
basins duo to the faster movement and the induced strong geological effect (Xie and Liu, 2010). For verification, we compared
the depth differences between glacial lakes of similar size in the continental and maritime glaciation zones. Longbasaba Lake
is the largest glacial lake with field bathymetric data in the continental glaciation zone, having an area of ~1.22 km$^2$ and a
largest depth of ~102 m in 2009 (Yao et al., 2012). In the maritime glaciation zone, Bienong Co in 2020 and Lugge Lake in
2002 had an area of ~1.15 km$^2$ and ~1.17 km$^2$, respectively, which were 5.74% and 4.10% smaller than Longbasaba Lake,
however, their largest depths were 77.45% and 23.52% larger than the Longbasaba Lake. Similarly, South Lhonak Lake in
2016 and Imja Lake in 2014, with an area of 1.31 km$^2$ and 1.30 km$^2$ respectively, were 7.38% and 6.56% larger than
Longbasaba Lake. And its maximum depth was 131 m and 150 m respectively, which were 28.43% and 47.06% larger than





Longbasaba respectively. In addition, Abmachimai Co in the continental glaciation zone had an area of 0.56 km² in 1987, and Lower Barun Lake in 1993 and Imja lake in 1992 in the maritime glaciation zone both had an area of 0.60 km², which was 7.14% larger than that of Abmachimai Co. Whereas the largest depth of Abmachimai Co was smaller than Lower Barun Lake's 51.29% and Imja Lake's 37.5%. As well as, the area of Qangzonk Co in the continental glaciation zone in 1987 and Thulagi

lake in the maritime glaciation zone in 1995 was both 0.76 km², but the maximum depth of the former was 19.12% smaller than that of the latter. However, it's worth noting that Jialong Co and Cirenma Co, located in the continental glaciation zone, have larger depths than the glacial lakes in the continental glaciation zone of similar size, such as Lower Barun lake in 1993 and Tam Pokhari Lake (unknown date). That may because they located in the Zhangzangbo valley, where the climate is dominated by the Indian monsoon (Wang et al., 2015, Li et al., 2020) and therefore the warm and humid air currents have a

greater impact on glaciers. As well as the classification of continental and maritime glacier zones is only a general range, without considering the topographic and climatic peculiarities of small areas. Overall, the comparison shows that the glacial lakes of same or similar area are deeper in the maritime glaciation zone. Notably, the subject of this study, Beinong Co, located in the SETP, has the largest average depth compared with glacial lakes in the Himalayas. The deeper glacier lake will store more water in a same area, and more volume of water will be released by a GLOF event, resulting in a more severe disaster to

downstream area.

**Table 5** Parameters of moraine-dammed glacial lakes having field bathymetric data.

| No. | Name | Location (Lat, Lon) | Region | Survey date | Area (km²) | Mean depth (m) | Largest Depth (m) | Volume ($10^8$ m³) | Mother glacier Type | Source |
|-----|------|---------------------|--------|-------------|------------|----------------|-------------------|--------------------|---------------------|--------|
| 1 | Bienong Co | 30.52, 93.45 | China | Aug, 2020 | 1.15 ± 0.05 | 85.40 | 181.04 | 1.02 | Maritime | This study |
| 2 | Jialong Co | 28.21, 85.85 | China | Aug, 2020 | 0.59± 0.02 | 63.11 | 133.43 | 0.38 | Continental | Li et al.,2021 |
| 3 | Cirenma Co | 28.06, 86.05 | China | Sep, 2012 | 0.39 ± 0.4 | 55 ± 2 | 115 ± 2 | 0.18 | Continental | Wang et al., 2015 |
| 4 | Longbasaba | 27.95, 88.08 | China | Nov, 2009 | 1.22 ± 0.023 | 48 ± 2 | 102 ± 2 | 0.64 | Continental | Yao et al., 2012 |
| 5 | Abmachimai Co | 28.09, 87.64 | China | Apr, 1987 | 0.56 | 33.93 | 72 | 0.19 | Continental | LIGG/WECS/NEA, 1988 |
| 6 | Qangzonk Co | 27.93, 87.88 | China | Apr, 1987 | 0.76 | 28.16 | 68 | 0.21 | Continental | LIGG/WECS/NEA, 1988 |
| 7 | Poqu Co | 28.30, 86.16 | China | Apr, 1987 | 0.31 | 19.35 | 33.93 | 0.06 | Continental | LIGG/WECS/NEA, 1988 |
| 8 | Gelhalpu Co | 27.96, 87.81 | China | - | 0.55 | 46.44 | - | 0.26 | Continental | Sakai, 2012 |
| 9 | South Lhonak | 27.91, 88.20 | India | Aug, 2014 & 2016 | 1.31 ± 0.001 | 50.24 | 131± 2.5 | 0.66 | Maritime | Sharma et al., 2018 |
| 10 | Thulagi | 28.50, 84.48 | Nepal | Oct, 2017 | 0.9 | 40.11 | 76 | 0.36 | Maritime | Haritashya et al., 2018 |
|  |  |  |  | Jul, 2009 | 0.94 | - | 80 | 0.35 |  | ICIMOD, 2011 |
|  |  |  |  | Mar, 1995 | 0.76 | - | 81 | 0.22 |  | Yamada, 1998 |
| 11 | Lower Barun | 27.80, 87.10 | Nepal | Oct, 2015 | 1.8 | 62.39 | 205 | 1.12 | Maritime | Haritashya et al., 2018 |
|  |  |  |  | May, 1993 | 0.60 | - | 109 | 0.28 |  | Yamada, 1998 |
| 12 | Imja | 27.90, 86.92 | Nepal | Oct, 2014 | 1.3 | 60.31 | 150 | 0.78 | Maritime | Haritashya et al., 2018 |
|  |  |  |  | May, 2009 | 1.01 | - | 97 | 0.36 |  | ICIMOD, 2011 |
|  |  |  |  | Apr, 2002 | 0.90 | - | 91 | 0.355 |  | Sakai et al., 2003 |
|  |  |  |  | Apr, 1992 | 0.60 | - | 99 | 0.28 |  | Yamada, 1998 |
| 13 | Tsho Rolpa | 27.87, 86.47 | Nepal | Aug-Sep, 2009 | 1.54 | 55.81 | 134 | 0.86 | Maritime | ICIMOD, 2011 |
|  |  |  |  | Feb, 1993 & 1994 | 1.39 | - | 131 | 0.77 |  | Yamada, 1998 |





| 14 | Dig Tsho | 27.87, 86.59 | Nepal | - | 0.5 | 20 | - | 0.11 | Maritime | Mool et al., 2001 |
| 15 | Tam Pokhari | 27.74, 86.84 | Nepal | - | 0.47 | 45.21 | - | 0.21 | Maritime | Mool et al., 2001 |
| 16 | Lugge | 28.09, 90.30 | Buhtan | Sep-Oct, 2002 | 1.17 | 49.83 | 126 | 0.58 | Maritime | Yamada, 2004 |
| 17 | Raphsthren | 28.10, 90.25 | Buhtan | 1984 & 1986 | 1.38 | 48.43 | 88 | 0.67 | Maritime | Geological survey of India, 1995 |

### 5.3 Relationship between area and volume

On the basis of the bathymetric map of USV results, the empirical relationships with significant correlations for area-volume, area-depth and depth-volume of Bienong Co were established (Fig. 12), and the valuable information is pinned on the hope that could provide a data reference for future studies of Bienong Co and other glacial lakes in the region. Due to the scarcity of glacial lake bathymetry data and its importance for GLOF hazard, several scholars have proposed the relationship between volume and area of glacial lake through available data (O'Connor et al., 2001; Huggel et al., 2002; Sakai, 2012; Wang et al., 2012a; Yao er al., 2012, Cook and Quincey, 2015). In this study, we fitted the relationship between area and volume based on a total of 24 bathymetric data for 16 glacial lakes (some lakes were measured multiple times) in the Himalayan region and Bienong Co in SETP. The results show that there is a significant correlation between area and volume with the correlation coefficient of 0.8391 at the level of significance less than 0.0001 (Fig. 13). But Bienong Co is obviously an outlier, and the correlation coefficient is lower than that by Wang et al., (2012) of 0.919. Therefore, we refitted the relationship without Bienong Co, resulting in a significant correlation with the correlation coefficient of 0.9426 higher than that by Wang et al., (2012) (Fig. 13).

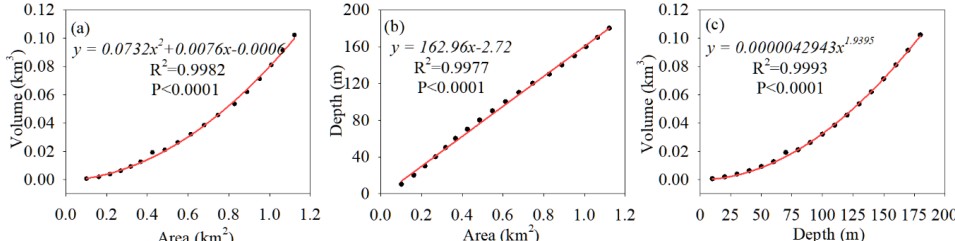

**Figure 12.** Nonlinear fitting of (a) area and volume, (b) area and depth and (c) depth and volume of Bienong Co.

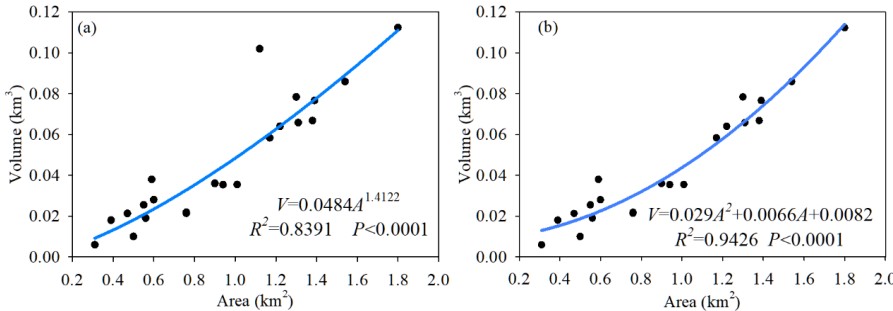

**Figure 13.** Relationship between area and volume of moraine-dammed glacial lakes in the Himalayas with and without Bienong Co in SETP based on available bathymetric data (in Table 5).





To compare the accuracy of our proposed area-volume relationship based on glacial lakes in the Himalayas and Bienong
Co in SETP, we summarized the published area-volume relationships for glacial lakes in the global and Himalayan regions
(Table 6). There are three equations extracted from just one glacial lake, three equations based on glacial lakes in the Himalaya,
one equation based on glacial lakes in the Himalaya and SETP, and four equations based on other regions worldwide including
the Himalaya (Table 6). Table 7 shows the comparison of measured and calculated lake volumes for moraine-dammed glacial
lakes in the Himalayas and SETP (lakes in Table 5) using the relationships between area and volume in table 6, noting that
only relationships based on multiple glacial lakes were selected. Overall, the calculated results based on the relationships of
O'Connor et al., (2001) as well as Cook and Quincey (2015) show very exaggerated overestimation errors for most of the
glacial lakes (Table 7 and Fig.14), indicating that the glacial lakes on which these equations are proposed are much deeper
than those in the Himalayan region. However, Huggel et al.,'s (2002) formula, although based on glacial lakes in non-
Himalayan regions, yields most of the underestimated results with small errors than that from the relationships of O'Connor et
al., (2001) as well as Cook and Quincey (2015). The formulas based on all or most of the glacial lakes located in the Himalaya
show relatively high calculation accuracy for the lakes in this region. In which the overall calculation accuracy of the formula
proposed in this study based on the glacial lake including Bienong Co is less than that of Sakai's (2012) formula, but the
formula developed without Bienong Co, namely based on glacial lakes that entirely in the Himalaya produces the highest
overall calculation accuracy (Table 7 and Fig.14), showing that our formula is a reliable reference for estimating the water
volume of the glacial lake in the Himalayas.

**Table 6** Summary of relationships between area and volume of glacial lakes based on measured bathymetry data.

| Type | Glacial lake in region | Formula ($R^2$:value) (A in $m^2$, V in $m^3$) | Source |
|---|---|---|---|
| Single glacial lake | Bienong Co in SETP | $V=0.0801A^{1.9146}$ ($R^2$: 0.9982) | This study |
| | Longbasaba in Himalaya | $V=0.0493A^{0.9304}$ ($R^2$: 0.9903) | Yao et al., 2012 |
| | Jialong Co in Himalaya | $V=616146.71793-20.76527A+2.6828\times10^{-4}A^2+2.17215\times10^{-10}A^3$ ($R^2$: 0.9999) | Li et al., 2021 |
| Multiple glacial lakes | 17 moraine-dammed glacial lakes in Himalayas and SETP | $V=0.0484A^{1.4122}$ ($R^2$: 0.8391) | This study (a) |
| | 16 moraine-dammed glacial lakes in Himalayas | $V=0.029A^2+0.0066A+0.0082$ ($R^2$: 0.9426) | This study (b) |
| | Two thermokarst and 14 moraine-dammed glacial lakes in the Himalayas | $V=0.4324A^{1.5307}$ ($R^2$: *) | Sakai, 2012 |
| | 20 moraine-dammed glacial lakes in the Himalayas | $V=0.0354A^{1.3724}$ ($R^2$: 0.919) | Wang et al., 2012a |
| | Seven moraine-dammed glacial lakes North America | $V=3.114A+0.0001685A^2$ ($R^2$: *) | O'Connor et al., 2001 |
| | Eight ice-dammed, one thermokarst and six moraine-dammed glacial lakes in North America, South America, Iceland and Alps Mountains | $V=0.104A^{1.42}$ ($R^2$: 0.91) | Huggel et al., 2002 |
| | (a)  Same as above | $V=0.1217A^{1.4129}$ ($R^2$: 0.95) | Cook and Quincey, 2015 |
| | (b)  45 glacial lakes including supraglacial, ice-dammed and moraine-dammed glacial lakes worldwide. | $V=0.1607A^{1.3778}$ ($R^2$: 0.74) | Cook and Quincey, 2015 |

Note: $R^2$: * refers that the original study does not specify the value.

**Table 7** Comparison of measured and calculated lake volumes for moraine-dammed glacial lakes in the Himalayas and SETP
(lakes in Table 5) using the relationships between area and volume in table 6.





| Lake Year | Area/km² | Measured volume/km³ | This study (a) Volume/km³ | Error /% | This study (b) Volume/km³ | Error /% | Sakai, 2012 Volume/km³ | Error /% | Wang et al., 2012a Volume/km³ | Error /% | O'Connor et al., 2001 Volume/km³ | Error /% | Huggel et al., 2002 Volume/km³ | Error /% | Cook and Quincey, 2015 (a) Volume/km³ | Error /% | Cook and Quincey, 2015 (b) Volume/km³ | Error /% |
|---|---|---|---|---|---|---|---|---|---|---|---|---|---|---|---|---|---|---|
| Bienong Co 2020 | 1.15 | 0.102 | 0.054 | -46.92 | 0.059 | -42.20 | 0.054 | -47.50 | 0.043 | -57.96 | 0.226 | 121.98 | 0.042 | -58.83 | 0.148 | 45.36 | 0.195 | 91.01 |
| Jialong Co 2020 | 0.59 | 0.038 | 0.022 | -41.61 | 0.023 | -39.54 | 0.019 | -49.26 | 0.017 | -54.84 | 0.060 | 59.19 | 0.016 | -57.16 | 0.058 | 51.97 | 0.078 | 104.41 |
| Cirenma Co 2012 | 0.39 | 0.018 | 0.015 | -15.64 | 0.013 | -28.87 | 0.010 | -43.16 | 0.010 | -45.99 | 0.027 | 49.13 | 0.009 | -49.76 | 0.032 | 78.74 | 0.044 | 143.96 |
| Longbasaba 2009 | 1.22 | 0.064 | 0.059 | -7.16 | 0.064 | 0.14 | 0.059 | -8.40 | 0.047 | -27.33 | 0.255 | 297.80 | 0.046 | -28.64 | 0.161 | 151.84 | 0.211 | 230.23 |
| Abmachimai Co 1987 | 0.56 | 0.019 | 0.021 | 10.48 | 0.021 | 12.33 | 0.018 | -6.31 | 0.016 | -15.93 | 0.055 | 187.29 | 0.015 | -20.44 | 0.054 | 182.33 | 0.072 | 280.47 |
| Qangzonk Co 1987 | 0.76 | 0.021 | 0.030 | 42.70 | 0.033 | 56.43 | 0.028 | 35.28 | 0.024 | 15.67 | 0.100 | 374.72 | 0.023 | 11.06 | 0.083 | 293.25 | 0.110 | 424.30 |
| Poqu Co 1987 | 0.31 | 0.006 | 0.013 | 117.22 | 0.009 | 54.31 | 0.007 | 19.99 | 0.007 | 18.25 | 0.017 | 185.97 | 0.007 | 8.80 | 0.023 | 287.69 | 0.032 | 433.41 |
| Gelhalpu Co* | 0.55 | 0.026 | 0.021 | -20.76 | 0.021 | -19.98 | 0.017 | -33.40 | 0.016 | -40.06 | 0.053 | 102.63 | 0.015 | -43.33 | 0.052 | 101.13 | 0.071 | 171.22 |
| South Lhonak 2016 | 1.31 | 0.066 | 0.067 | 0.93 | 0.071 | 7.38 | 0.065 | -0.95 | 0.051 | -22.30 | 0.293 | 344.31 | 0.051 | -23.44 | 0.178 | 170.05 | 0.233 | 253.22 |
| Thulagi 2017 | 0.9 | 0.036 | 0.038 | 4.53 | 0.042 | 15.86 | 0.037 | 2.22 | 0.031 | -14.91 | 0.139 | 286.91 | 0.030 | -17.63 | 0.105 | 191.30 | 0.139 | 286.07 |
| Thulagi 2009 | 0.94 | 0.35 | 0.040 | -88.56 | 0.044 | -87.33 | 0.039 | -88.76 | 0.033 | -90.71 | 0.152 | -56.62 | 0.032 | -90.99 | 0.112 | -68.14 | 0.148 | -57.84 |
| Thulagi 1995 | 0.76 | 0.22 | 0.030 | -86.38 | 0.033 | -85.07 | 0.028 | -87.09 | 0.024 | -88.96 | 0.100 | -54.69 | 0.023 | -89.40 | 0.083 | -62.46 | 0.110 | -49.95 |
| Lower Barun 2015 | 1.8 | 0.112 | 0.114 | 1.82 | 0.111 | -0.89 | 0.106 | -5.07 | 0.079 | -29.19 | 0.552 | 392.45 | 0.079 | -29.16 | 0.279 | 149.31 | 0.361 | 222.49 |
| Lower Barun 1993 | 0.60 | 0.28 | 0.023 | -91.93 | 0.024 | -91.60 | 0.020 | -92.93 | 0.018 | -93.73 | 0.063 | -77.67 | 0.017 | -94.05 | 0.059 | -78.88 | 0.079 | -71.61 |
| Imja 2014 | 1.3 | 0.078 | 0.066 | -15.65 | 0.070 | -10.12 | 0.065 | -17.17 | 0.051 | -34.94 | 0.289 | 270.27 | 0.050 | -35.92 | 0.176 | 126.04 | 0.231 | 195.74 |
| Imja 2009 | 1.01 | 0.36 | 0.044 | -87.65 | 0.049 | -86.37 | 0.044 | -87.80 | 0.036 | -90.03 | 0.175 | -51.38 | 0.035 | -90.30 | 0.123 | -65.72 | 0.163 | -54.74 |
| Imja 2002 | 0.90 | 0.355 | 0.038 | -89.4 | 0.042 | -88.25 | 0.037 | -89.63 | 0.031 | -91.37 | 0.139 | -60.76 | 0.030 | -91.65 | 0.105 | -70.46 | 0.139 | -60.85 |
| Imja 1992 | 0.60 | 0.28 | 0.023 | -91.93 | 0.024 | -91.60 | 0.020 | -92.93 | 0.018 | -93.73 | 0.063 | -77.67 | 0.017 | -94.05 | 0.059 | -78.88 | 0.079 | -71.61 |
| Tsho Rolpa 2009 | 1.54 | 0.086 | 0.087 | 1.33 | 0.089 | 3.55 | 0.084 | -2.63 | 0.064 | -25.55 | 0.404 | 370.24 | 0.064 | -26.07 | 0.224 | 160.46 | 0.291 | 238.75 |
| Tsho Rolpa 1994 | 1.39 | 0.77 | 0.073 | -90.47 | 0.077 | -89.99 | 0.072 | -90.70 | 0.056 | -92.78 | 0.330 | -57.16 | 0.055 | -92.86 | 0.194 | -74.83 | 0.253 | -67.15 |
| Dig Tsho* | 0.5 | 0.011 | 0.019 | 70.45 | 0.018 | 65.32 | 0.015 | 36.05 | 0.014 | 24.30 | 0.044 | 297.11 | 0.013 | 17.00 | 0.046 | 315.50 | 0.062 | 462.17 |
| Tam Pokhari* | 0.47 | 0.021 | 0.018 | -15.68 | 0.017 | -20.65 | 0.014 | -35.17 | 0.013 | -40.19 | 0.039 | 84.22 | 0.012 | -43.87 | 0.042 | 99.42 | 0.057 | 170.40 |
| Lugge 2002 | 1.17 | 0.058 | 0.056 | -4.10 | 0.060 | 4.16 | 0.055 | -5.20 | 0.044 | -24.29 | 0.234 | 303.97 | 0.043 | -25.80 | 0.152 | 161.94 | 0.200 | 243.98 |
| Raphsthren 1986 | 1.3 | 0.067 | 0.066 | -1.81 | 0.070 | 4.64 | 0.065 | -3.57 | 0.051 | -24.26 | 0.289 | 331.06 | 0.050 | -25.40 | 0.176 | 163.15 | 0.231 | 244.30 |

Note: Error = (Volume of empirical formulas − Bathymetrically derived volume) / Bathymetrically derived volume × 100%.

"*" of lake/year means that the year is unclear. Formula-(a) and Formula-(b) are the relationships between area and volume of glacial lakes in table 5 with and without Bienong Co, respectively.



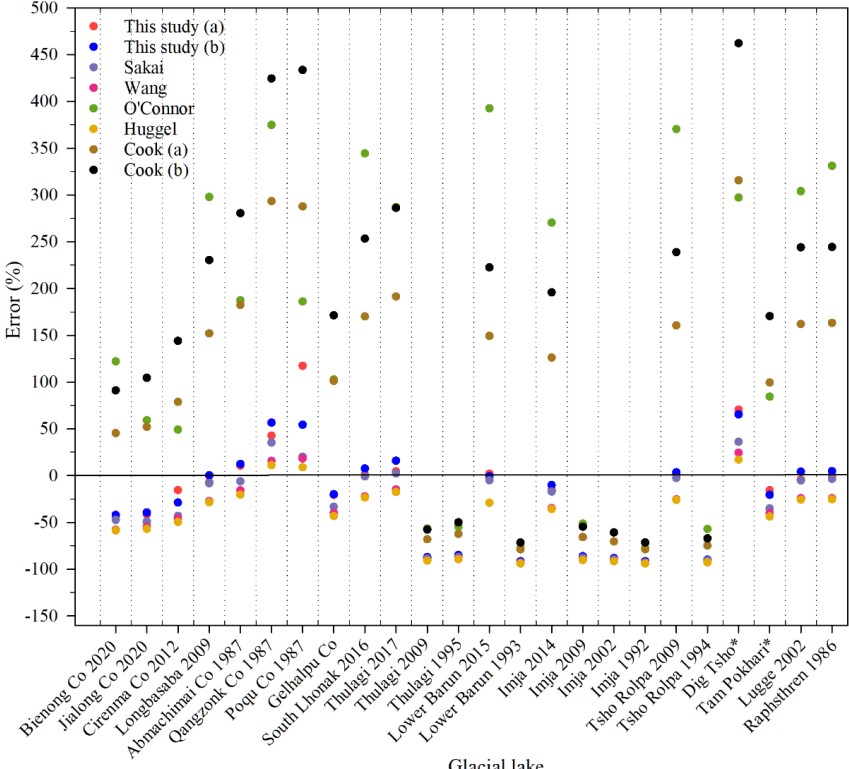

**Figure 14.** Comparison of errors from the calculated volumes for moraine-dammed glacial lakes in the Himalayas and SETP (glacial lakes in Table 5) using the relationships between area and volume in table 6.

## 6 Conclusion

As a moraine-dammed glacial lake located in the maritime glaciation region, Bienong Co has been highly regarded by local government due to its larger area and high hazard of GLOF. Based on bathymetric data obtained from field investigation, combined with remote sensing images and DEM data, we analyzed the potential hazard of Bienong Co, accurately modeled its basin morphology and estimated its water volume. Furthermore, we simulated the hydraulic behavior at breach and in the downstream flow channel based on four different scenarios of GLOFs to assess the hazard of different magnitudes. The following main conclusions were drawn:

(1) Bienong Co is a very stable glacial lake that has remained essentially constant in size over the past four decades. Also maintaining its area is the mother glacier, but it has undergone significant thinning, especially in the ablation zone, which is supposed to be a response to climate warming. According to the field bathymetric data, the lake basin morphology of Bienong Co features a relatively flat basin bottom and the steep flanks, with the slope near the glacier (16.5°) is steeper than that near the moraine dam (11.3°). In August 2020, the maximum depth of Bienong Co was ~181.04 m, with the water storage capacity of ~10.2×10$^7$ m$^3$.

(2) Four scenarios of extreme-, high-, medium- and low-magnitude of GLOFs based on different combinations of breach depth (72 m, 36 m, 18 m and 9 m), breach width (180 m, 131 m, 94 m and 66 m), and failure time (0.75 h, 1.03 h, 1.37 h and 1.79 h) produced the peak discharges of 26,721 m$^3$ s$^{-1}$, 11,126 m$^3$ s$^{-1}$, 3,716 m$^3$ s$^{-1}$ and 1,294 m$^3$ s$^{-1}$ by MIKE 11 model as well as 24,630 m$^3$ s$^{-1}$, 8,801 m$^3$ s$^{-1}$, 3,081 m$^3$ s$^{-1}$ and 1,070 m$^3$ s$^{-1}$ by empirical relationship. Extreme-magnitude GLOF will have a catastrophic impact on downstream, affecting almost all man-made facilities including settlements, bridges



and roads along the river channel. However, it has a small probability of occurrence, and the simulation in this paper only represents a potential possibility. In contrast, the low-magnitude GLOF will produce a relatively mild influence on the downstream flow channel, with only three villages at the downstream being fully flooded. Nonetheless, the impact should not be despised, because the amount of water released in low-magnitude scenario is comparable to that in the Jinwu Co' GLOF event in 2020 which caused a great damage to the downstream region.

(3) Glacial lakes in the maritime glaciation region are generally deeper than those in the continental glaciation region, therefore store more water, resulting in a more severe disaster to downstream area in the event of a GLOF. Bienong Co, a moraine-dammed glacial lake located in the SETP, has the largest relative depth comparing with those located in the Himalayas. Therefore, glacial lakes in maritime glaciation region, such as in the SETP should be given more attention in the future.

**Author contributions.** HD contributed the conceptualization, methodology, software, formal analysis, visualization and writing of the original draft; XY contributed the conceptualization, supervision, funding acquisition, investigation of the glacial lake, as well as review and editing of the manuscript; HJ, YZ and QW contributed the investigation of the glacial lake; ZD and QW contributed the model progress; JH contributed the setting up the experimental equipment and obtaining data.

**Competing interests.** The authors declare that they have no conflict of interest.

**Financial support.** This research has been supported by the National Key Research Program of China (no. 2019YFE0127700), National Natural Science Foundation of China (grant nos. 41861013 and 42071089), "Innovation Star" of Outstanding Graduate Student Program in Gansu Province (no. 2021-CXZX-215) and Northwest Normal University's 2020 Graduate Research Grant Program (no. 2020KYZZ001012).

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
