# Peer review of "Lake volume and potential hazards of moraine-dammed glacial lakes in temperate glaciation regions—A case study of Bienong Co"

_The Cryosphere, 2022_

## Author Comment (AC1)

**Review#1**

This study aims at estimating lake volume and modelling potential GLOF from the Lake Bienong, SE Tibet Plateau. In general, I'm convinced that such studies are needed and are logical step following region-wide GLOF susceptibility assessments. The outcomes might be of interest for research community as well as DRR practitioners. The authors of this study employ broadly used data (Landsat images, ALSO PALSAR DEM, …) and methods (bathymetric surveying, empirical equations for deriving breach parameters, MIKE11 modelling tool) in new geographical context. As such, this study brings only limited novelty in terms of methodological development and to some extent only replicates the workflows of previous GLOF modelling studies of different lakes. I would maybe expect more novelty in leading journals such as the Cryosphere. Below I provide my comments to individual parts of this study:

The section about lake evolution actually presents no evolution and can be omitted or summarized in one sentence in the Introduction or Study area section in my opinion.

**Reply:** We present the evolution of the glacial lake and its mother glacier in Results section to make the relative complete study and presentation of the research objective, but the fact is that the glacial lake has not evolved too drastically. Based on your suggestion, we present this part in the Study area section.

The authors defined 4 moraine dam breach (GLOF) scenarios which are then modelled in the MIKE11 and MIKE21 software. However, the moraine dam breach is not the beginning of the process chain, but a consequence of certain triggering event. Considering that this is detailed case study of only one lake, I would expect the authors to analyze the whole process chain in as much detail as possible, i.e. starting with detailed quantitative analysis of potential GLOF triggers which would help to define and justify dam breach scenarios.

However, this is not met in the current version of the manuscript. My major concern is that the breach scenarios are defined subjectively and are not linked to possible GLOF triggers identified in Section 5.1. What is described in Section 5.1. gives mainly qualitative overview of potential GLOF triggers. This is perhaps true for many lakes in the region, but there is no link to considered breach depth scenarios. And this is the major drawback of this study in my opinion.

**Reply:** In this study, with the objective of assessing the possible risk to downstream areas from potential GLOFs in different magnitudes, we assumed four scenarios based on different break heights of the moraine dam. We appreciate your valuable and constructive comments on this study. According to your suggestion, we analyzed the whole process chain of GLOFs starting with detailed quantitative analysis of potential GLOFs triggers, following with the generation and propagation of waves in the lake caused by potential triggers, the erosion of the dam by overtopping floods, and the resulting GLOFs in the downstream. We analyze the potential GLOF triggers in Section 5.1, i. e. ice avalanches from the tongue of its parent glacier and steep lateral moraine landslides on both sides. Considering that the ice avalanches are the most common GLOFs trigger in the Tibet, China, which induced the outburst of Rangze Glacial Lake, just 10 km away from Bienong Co. And the lateral moraine landslide is the GLOFs trigger of Jinwu Co, which is only 24 km away from Bienong Co, so in this revision, we took the two potential triggers for analysis. Wang et al., (2012) defined the volume of dangerous glacier (VDG) as the glacier volume from the location of abrupt change in the slope to the glacier terminus or the volume of terminal glacier where ice cracks are well developed. We adopted the latter, i.e., the glacier volume with a surface area of 0.19 $km^2$

for crevasse development, as the potentially dangerous ice body of the parent glacier of Bienong Co. To simulate the subsequent effects triggered by ice avalanche material in different magnitude, we divided the potentially hazardous ice body into three parts based on the surface elevation range, representing partial or total collapse of the hazardous ice body, the dangerous ice body 01, the dangerous ice body 02, and the dangerous ice body 03, simulating three scenarios of low-, medium- and high- magnitude (Figure1). In scenario-1, ice body at elevation below 4844 m (the dangerous ice body 01) yields a release area of 0.05 km$^2$ with the mean height difference of 76 m from the lake surface, which is the low-magnitude trigger. In scenario-2, ice body at elevation below 4889 m (the dangerous ice body 02) yields a release area of 0.11 km$^2$ with the mean height difference of 103 m from the lake surface, which is the moderate-magnitude trigger. In scenario-3, the total ice body of crevasse with an area of 0.19 km$^2$ (the dangerous ice body 03) that is below the elevation range of 4896-4900 m and has the mean height difference of 131 m from the lake surface, which is the extreme-magnitude trigger. For the release depth we still adopted a hypothetical approach because there is no reliable reference data available. The release depth of the ice avalanche in each scenario is assumed to be its mean height difference from the lake surface, i.e., all of the ice body above the lake surface falls off. This is a very extreme case because the subglacial topography is unknown and the lower part of the ice tongue may be in the shape of a dome that would overestimate the avalanche volume of the collapsed ice body. But in the Rapid Mass Movement Simulation (RAMMS) Avalanche module (Bartelt et al., 2013) simulation, the real ice body falling into the glacial lake is reduced, and the model will make a judgment based on the topography instead of dumping the entire ice body we set into the lake.

[Figure]

Figure 1. Illustration of setting triggers for potential GLOFs in Bienong Co

In addition, we selected lateral moraine landslides as another GLOF triggers for Bienong Co, which is not common on the Tibetan Plateau, but the 2020 GLOF of Jinwu Co was caused by a lateral moraine landslide (Liu et al., 2021; Zheng et al., 2021). The slope of the lateral moraine where the landslides occurred in Jinwu Co is

between 30˚-45˚, and there are several lateral moraines around the Bienong Co that fit into this slope range. We selected two release areas of lateral moraine landslide, one is located on the right bank of Bienong Co facing the mother glacier, near the moraine dam with an area of 0.015 km$^2$ (lateral moraine landslide zone 01), another is located on the left bank of Bienong Co facing the mother glacier, near the mother glacier with an area of 0.024 km$^2$ (lateral moraine landslide zone 02). Two sites are at different distances from the moraine dam and further, we set three different release depths of 2 m (scenario-1, low-magnitude), 5 m (scenario-2, moderate-magnitude), and 10 m (scenario-3, extreme-magnitude) for each zone. The above settings fully consider the impact of triggers on Bienong Co under different scenarios, and the results are used as the input for the subsequent disaster chain.

Reference:

1. *Wang X, Liu S, Ding Y, et al. An approach for estimating the breach probabilities of moraine-dammed lakes in the Chinese Himalayas using remote-sensing data. Natural Hazards and Earth System Sciences, 2012, 12(10): 3109-3122.*

2. *Bartelt, P., Buehler, Y., Christen, M., Deubelbeiss, Y., Graf, C., McArdell, B., Sals, M., and Schneider, M.: RAMMS: Rapid Mass Movement Simulation: A numerical model for debris flows in research and practice, User Manual v1.5 – Debris Flow, Swiss Institute for Snow and Avalanche Research SLF, Birmensdorf, 2013.*

3. *Liu, J. K., Zhou, L. X., Zhang, J. J., and Zhao, W. Y.: Characteristics of Jiwencuo GLOF, Lhari county, Tibet. Geological Review, 67: 17–18. https://doi.org/10.16509/j.georeview.2021.s1.007, 2021.*

4. *Zheng, G. X., Mergili, M., Emmer, A., Allen, S., and Stoffel, M.: The 2020 glacial lake outburst flood at Jinwuco, Tibet: causes, impacts, and implications for hazard and risk assessment, The Cryosphere, 15, 3159–3180, https://doi.org/10.5194/tc-2020-379, 2021.*

For instance, I wonder what would need to be the magnitude of triggering slope movement to initiate 72 m deep breach? Is there any evidence that such slope movement could occur in the lake's surrounding? Landslide zones identified in Figure 11 don't seem to be releasing large mass volumes into the lake. Ideally, the initial slope movement, displacement wave propagation in the lake as well as dam breaching would be modelled, not only the GLOF. Critical questions regarding potential GLOF triggers are: Is there any evidence of mass movements entering the lake in the past? Have you observed any evidence from your analysis of remote sensing images and DEMs? Is there any evidence of potential future mass movements (displaced blocks, surface ruptures, opening crevasses, etc.)? Any evidence from your field work? Did any strong earthquake hit the region in the past? Did the lake experience any precipitation / temperature extremes? Do you expect them to change in the future? Considering the scope of the study (case study of one lake), individual triggers should be identified, quantified and treated more in depth in my opinion, feeding the definition of dam breach (GLOF) scenarios.

**Reply:** The Landsat image series shows a ~120 m retreat at the Bienong Co's ice tongue from 1976-1988 (maybe earlier), after which small changes were not identified because of the coarse spatial resolution of the available images. Comparing the image in ArcGIS Earth softwaer (earlier than 2020, but the exact date is not known) (Figure2-left) with the photos taken during the 2020 field survey (Figure2-right), it is clear that the glacier tongue was in a significantly different state and have collapsed in 2020 compared to the earlier period. It is clear from the above evidence that the glacier tongue is in a changing state, although we did not directly witness one such change process during our field survey. Meanwhile, a large number of cracks appear on the mother glacier's tongue of in

Bienong Co, clearly visible in both remotely sensed images and real photographs (Figure 2), which is the basis of a potential ice avalanche into the lake. In addition, Lv et al., (1999) proposed that a slope of the mother glacier' tongue greater than 8° is conducive to the occurrence of ice avalanches, meaning the potentiality of the GLOFs. In terms of Bienong Co, slopes of the dangerous ice body01, dangerous ice body02, dangerous ice body03 and ablation zone are 20°, 22°, 26° and 32°, respectively, all of which exceed the above threshold, indicating the potential hazard.

[Figure]

Figure 2. Comparison at the tongue of the mother glacier of the glacial lake in different periods (left: the high-resolution image from ArcGIS Earth software, the date is earlier than 2020, but the exact date is not known; right: taken in August 2020)

The scenario for the extreme-magnitude of GLOFs that downcutting to the base of the moraine dam (72 m) is informed by the hypothesis of Sattar et al. (2021) for Lower Barun Glacial Lake, for which they set the extreme scenario of the outburst incision reaching the bottom of the moraine (104 m) due to the initial erosion of the overtopping floods. We assume that this event may be caused by one or a combination of factors such as ice/avalanches, rock falls, landslides, heavy precipitation, or earthquakes. According to your comments, this assumption may be relatively subjective because of our lack of investigation of strong earthquakes and extreme precipitation and temperature events in the region and the small probability of GLOFs due to earthquakes in the Tibet, China. The probability of glacial meltwater and heavy precipitation and its related combination induced GLOFs event can account for one-fifth, such as the GLOF of Jinwu Co in 2020 was caused by precipitation-induced lateral moraine landslides to make the glacial lake destabilization, and finally GLOF event. In Tibet, China, the ice avalanche/ice landslide and its related combination of factors induced by the GLOFs event accounts for more than seventy percent. However, quantitative simulations of dam breach erosion due to such a trigger mechanism were not performed in our study.

Therefore, in this revision, we first quantified the potential GLOF triggers based on your comments, in which two types of ice avalanches and lateral moraine landslides were set, and the details of the quantification are described in the previous reply. Then, we simulated the initial slope motion, the propagation of displacement waves in the lake, and the dam failure in the ideal case you mentioned. The initial slope motion is modeled by the Rapid Mass Movement Simulation (RAMMS) Avalanche module (Bartelt et al., 2013), and the subsequent propagation of displacement waves in the lake, climbing, moraine dam erosion, dam failure and downstream flood propagation are simulated by the Basic Simulation Environment for Computation of Environmental Flow and Natural Hazard

Simulation (BASEMENT) model ((Vetsch et al., 2017).   the Rapid Mass Movement Simulation (RAMMS) Debris Flow module (Bartelt et al., 2013).

Reference:

1.  Lv, R. R., Tang, X. B., and Li, D. J.: Glacial lake outburst mudslide in Tibet, Chengdu University of Science and Technology Press, Chengdu, 69–105, 1999.

2.  Sattar, A., Haritashya, U. K., Kargel, J. S., Leonard, G. J., and Chase, D. V.: Modeling Lake Outburst and Downstream Hazard Assessment of the Lower Barun Glacial Lake, Nepal Himalaya, J. Hydrol., 598, 126208, https://doi.org/10.1016/j.jhydrol.2021.126208, 2021.

3.  Bartelt, P., Buehler, Y., Christen, M., Deubelbeiss, Y., Graf, C., McArdell, B., Sals, M., and Schneider, M.: RAMMS: Rapid Mass Movement Simulation: A numerical model for debris flows in research and practice, User Manual v1.5 – Debris Flow, Swiss Institute for Snow and Avalanche Research SLF, Birmensdorf, 2013.

4.  Vetsch, D., Siviglia, A., Ehrbar, D., Facchini, M., Kammerer, S., Koch, A., Peter, S., Vonwiller, L., Gerber, M., Volz, C., Farshi, D., Mueller, R., Rousselot, P., Veprek, R., and Faeh, R.: System Manuals ofBASEMENT, Version 2.7. Laboratory of Hydraulics, Glaciology and Hydrology (VAW), ETH Zurich, available at: http://www.basement.ethz.ch, last access: 3 November 2017.

For the modelling part (Section 4.3, Table 3, Figures 7 to 10), flow velocities and peak discharge drop in Bada to 0.26 m/s (2,260 $m^3$/s) in the most extreme scenario, after which it again speeds up to 18.47 m/s (22,992 $m^3$/s) in Zongri is contra-intuitive and should be discussed / explained. In Figure 10, you even have negative peak discharge in Bada (?). Also the flow velocities of Scenario-1 (S1 to Jiawu; 44 to 65 m/s) seem unrealistically high considering it is supposed to originate from moraine dam breach.

**Reply:** The Bada village should not have been the subject of analysis because it is not located directly in the downstream channel of Bienong Co, but in another channel that intersects with the downstream channel of Bienong Co. However, in the extreme-magnitude scenario originally simulated, the Bada village was also affected by the huge flood. Therefore, the impact of flooding on it was analyzed. Because the Bada village is relatively far from the main channel of the flood, the flow and velocity here are significantly reduced. The flood flows into the cross-section representing this region and then flows out in the opposite direction, i.e., exited the region, which is the reason for the negative value of the flow in the later period. But this situation will be improved in this modeling. Again, thank you very much for your valuable comments.

Finally, the authors invested a lot of effort in comparing lake volumes (Section 5.2) and lake volume estimates (Section 5.3), but the implications of these comparisons are nebulous to me and the statistical treatment is incorrect. Considering the bathymetry done by the authors, it is clear that: (i) they have the best possible lake volume estimate for their study; (ii) their one bathymetry can hardly be used to validate or evaluate existing area-volume relationships. Strikingly, bathymetric data used to develop the new area-volume relationship (Table 5, Figure 12) are then used for the performance assessment (Tabel 7, Figure 14), which is statistically not correct. Moreover, some of the existing equations (e.g. Fujita's equation developed specifically for Himalayan lakes) are not considered in this comparison. Further, the number of lakes listed in Table 5 is too low to generate any meaningful conclusions about a difference between lakes associated with continental and maritime glaciers. In addition, it seems that simply larger lakes are deeper as you only have one lake > 1 $km^2$ associated with continental glacier in

your dataset while all lakes associated with maritime glaciers are $> 1km^2$. This makes the Discussion section overall weak.

**Reply:** Thank you for your comment, we fully understand your opinions and for the discussion part, we will work on the glacial lake hazard quantification, evaluation of the existing glacial lake volume estimation relationships and uncertainty study of GLOFs process chain simulation.

Table 4: if the lake has a surface outflow, dam freeboard = 0m.

**Reply:** Thank you for your tips.

Mean breach (Eq. 3) is used as max. breach in Figure 6; please check

**Reply:** In this revision, the dam-break process will be re-simulated with the BASEMENT model, and Equation 3, which calculates the average breach depth, and Figure 6, which represents the shape of the breach, are discarded, so we will reconsider this problem. We appreciate your comment.

I think that especially GLOF triggers need to be addressed in more detailed and quantitative way first, resulting in re-definition and justification of dam breach scenarios. Also the Discussion section should be re-worked substantially in my opinion. To sum up, I recommend major revisions of this manuscript.

**Reply:** In this revision, we have simulated the GLOFs processes, including the initial slope movement, generation and propagation of wave in the lake, dam breaching and the floods behaviors in the downstream channel starting from quantifying the triggers of according to your suggestion. The above is the first aspect of the major revisions. Second, for the discussion section, in this revision, we will not attempt to update the existing volume estimation formulae using bathymetry data of Bienong Co because sample size is too small. The discussion will focus on a deeper investigation of the danger of the glacial lake, the risk of glacial lake outburst floods to the downstream region, the evaluation of existing glacial lake volume estimation relationships, the evaluation of GLOFs simulations, etc.

We attached reviewer 2's comments on this paper, as well as our response.

**Review#2**

This is an interesting paper that investigates a glacial lake and conducts analyses aiming to assess its evolution, basin morphology, estimate its water volume, analyses some possible outburst triggers analysis, and conducts simulations of likely inundation under GLOF scenarios. In general, the paper does a good job of developing these themes and forms a comprehensive case study that could be published given some reasonably substantial changes. I list these here and also some more specific issues with the paper.

1 The paper is often written in a rather vague and imprecise way. In addition, there is often an incorrect use of English. I sympathise with the authors in this; it is difficult for non-English speakers to write precisely and accurately in English but this paper would benefit enormously from careful editing and rewriting by a native English speaker.

The literature review is generally comprehensive although there are some papers that should have been referenced and I highlight some of these later. The rationale for the study is clear and appropriate.

**Reply:** We appreciate your valuable comments on this manuscript and your full understanding of our language

issues. Combining Reviewer 1's and your comments, we will make significant changes to the manuscript. Reviewer 1 suggests that we analyze the whole process chain in as much detail as possible, i.e. starting with detailed quantitative analysis of potential GLOF triggers, we did the above in this revision. And, as you suggested, we will provide a more detailed explanation and analysis of the typicality and dangers of the glacial lakes we have studied in this revision.

3 The sampling strategy and methodology is not clearly discussed. Explain why this lake was chosen. Is this lake representative of others in the region? If so, why and how do you know this? If the lake is not representative, then the authors need to explain its significance.

**Reply:** In fact, Bienong Co Glacial Lake is representative of the region (the Yi'ong Zangbo river basin in the southeastern Tibetan Plateau) in which it is located, and the characteristics of the Yi'ong Zangbo river basin and Bienong Co are described in our Study area section. But the way it is written does not seem to be clear enough to make you question it. We will improve this in this revision.

The southeastern Tibetan Plateau, with high mountains and deep valleys and under the influence of the Indian monsoon, has developed numerous maritime glaciers and moraine-dammed and other types of glacial lakes. In this study, the Yi'ong Zangbo river basin in the southeastern Tibetan Plateau was used as the study area, and according to Duan et al. (2020), there are 105 moraine-dammed glacial lakes (the main lake type for GLOFs occurring in the Himalaya and southeastern Tibetan Plateau) within the basin with a total area of 16.87 $km^2$, in which, 67 lakes have an area larger than 0.02 $km^2$. Based on the characteristics of the breached glacial lakes in the southeastern Tibetan Plateau, Duan et al. (2020) selected five indicators of (1) the area of the mother glacier, (2) distance between the lake and glacier terminus, (3) slope between the lake and glacier, (4) mean slope of the moraine dam, and (5) snout steepness of the mother glacier to evaluate the GLOFs potentiality of the 67 glacial lakes, and GLOFs susceptibility was classified as very high, high, medium and low. The results show that there were 10 glacial lakes with a very GLOFs potentiality, of which seven lakes had an area larger than 0.1 $km^2$, five had an area larger than 0.2 $km^2$, and only two lakes had the area larger than 0.5 $km^2$, one is Jinwu Co, which has already breached, with an area of 0.53 $km^2$ before the failure (2016), and the other is the subject of this study, Bienong Co, whose area in 2016 is about 1.12 $km^2$. GLOFS of smaller glacial lakes cause limited damage in sparsely populated regions like Tibet, so larger lakes tend to be valued more. The area of Bienong Co is nearly twice as large as the area of the already collapsed Jinwu Co, and given the huge damage caused by the outburst flood of Jinwu Co to the downstream region, the potential GLOFs hazard of Bienong Co cannot be ignored.

Secondly, three GLOFs have occurred in the Yi'ong Zangbo river basin, the Jinwu Co that breached in June 2020 is 24 km away from Bienong Co in a straight line, and the Ranze Glacial Lake that breached in July 2013 is only 10 km away Bienong Co in a straight line. Considering the similar geological and climatic environment due to the proximity, and Bienong Co was also assessed as a very high dangerous glacial lake in previous study, we believe that it is necessary and meaningful to select Bienong Co for GLOFs process chain simulation.

Reference:

1. *Duan H, Yao X, Zhang D, et al. Glacial lake changes and identification of potentially dangerous glacial lakes in the Yi'ong Zangbo River Basin. Water, 2020, 12(2): 538.*

4 The methodologies used are explained and justified well, and you have used an appropriate range of techniques

to explore the geomorphology, characteristics and evolution of the lake and its future behaviour.

**Reply:** Thank you very much for your affirmation of the method adopted in this study. However, according to the comments of reviewer 1, the simulation of this study still has some shortcomings. We will adopt more detailed models and methods to simulate the process chain of glacial lake outburst flood.

5 I am interested in why Bienong Co is regarded as a dangerous lake (lines 100 and 239)? The paper demonstrates that the lake has remained stable for some time, and that it cannot expand further. It also argues that the moraine dam does not contain an ice core. So the description of the lake as 'dangerous' requires much more discussion and evidence. This is important because there is always the temptation to describe any moraine - dammed lake as being 'dangerous' even when the evidence for this is lacking. I know of one well-known reviewer of similar papers who regularly rejects all papers who make this assertion without clear evidence!

**Reply:** There are many indicators that can be used to assess the hazard of glacial lakes. Zhang et al. (2022) systematically summarized the indicators for assessing glacial lake's hazard in the Himalayan, including six main categories: (1) ice avalanches, glacier collapse, (2) rock fall, landslide, or other solid mass movement, (3) dam instability, (4) heavy precipitation, various liquid inflows, (5) characteristic of lake, (6) influence to downstream area. Many specific indicators are included in each category for a total of 57. Zhang et al. (2022) selected the best combination of five factors for assessing the potential hazard of a glacial lake from 57 factors by designing experiments: (1) the average slope of the mother glacier, (2) the likelihood of material entering the lake, (3) the average slope of the moraine dam, (4) the watershed area, and (5) lake perimeter, GLOF triggers corresponding to glacial collapse, rockfall and landslides, dam instability, heavy precipitation or other fluid inflow, and lake characteristics. Considering that ice avalanches and landslides trigger almost more than half of the GLOF (Emmer and Cochachin, 2013); the average slope of the mother glacier and the likelihood of material entering the lake were given higher weights in assessing the GLOF potential of glacial lakes in the Himalayan using the fuzzy consistency matrix method.

The hazards of a glacial lake are influenced by multi factors, such as the area and its growth rate, the properties of the dam and the slope of the surrounding environment. In terms of Bienong Co, its area has remained essentially stable over the last 30 years, which is the main reason why it is considered to be of low risk. However, its huge size is a danger in itself. Whether its moraine dam contains ice cores is currently unknown. But, the present of ice core is only one of the factors representing the stability of moraine dams, dam height, dam texture (consolidated or unconsolidated, bedrock, or other) dam freeboard, mean slope, the present of ice core, the present of leakage, et al. are also the important characteristics. The moraine dams of Bienong Co are high and steep, consisting of loose materials. With a freeboard of 0 m, any mass entering the lake will create an overtopping flow that may erode the moraine dam and make the breach larger, thus releasing a large amount of water from the lake. Sattar et al. (2021) even hypothesized the situation where the moraine dam collapses to the bottom (104 m) due to the overtopping flow. Therefore, from the viewpoint of moraine dam, Bienong Co is dangerous, because the moraine dams of the two nearby glacial lakes that have broken both have such characteristics.

Furthermore, based on the genesis of breached glacial lakes on the Tibetan Plateau, ice avalanches and landslides appear to trigger more than half of the GLOFs, and it is reasonable to assign a higher weight to the possibility of ice avalanches and landslides entering glacial lakes in studies related to glacial lake hazard assessment.

Lv et al., (1999) proposed that a slope of the mother glacier' tongue greater than 8° is conducive to the occurrence of ice avalanches, meaning the potentiality of the GLOFs. In terms of Bienong Co, slopes of the ablation zone are 20°, which exceed the above threshold, indicating the potential hazard. And, the moraine slope destabilization conditions that lead to the failure of the Jinwu Co also exist around the Bienong Co, so it may also become a potential trigger.

Finally, studies show that (Sun et al., 2014; Liu et al., 2021 and Zheng et al., 2021) the Ji Weng wrong and Ran Zerzhong A wrong collapse although the avalanche and ice avalanche into the lake and side moraine landslide into the lake is the most important factor, but in the early sustained precipitation and the rapid recovery of temperature still on the ice / avalanche and side moraine landslide has a certain role.

Finally, studies show that although avalanche and ice avalanche into the lake and lateral moraine landslide into the lake were the dominant factors in the GLOFs of Jinwu Co and Ranze Lake, the preceding precipitation and rapid warming still played a role in the occurrence of ice/avalanche and lateral moraine landslide. Bienong Co is in such close proximity to these two glacial lakes that the same climatic conditions are likely to act on Bienong Co and cause GLOFs to occur.

In summary is the explanation of the potential hazards of Bienong Co. We believe that it is reasonable to evaluate Bienong Co as a glacial lake with the GLOFs possibility, and although the exact trigger and timing of its break cannot be determined, our study at least provides a prediction and assessment of the potential hazards of this glacial lake.

Reference:

1   Zhang T, Wang W, Gao T, et al. An integrative method for identifying potentially dangerous glacial lakes in the Himalayas[J]. Science of The Total Environment, 2022, 806:150442.

2.   Sattar, A., Haritashya, U. K., Kargel, J. S., Leonard, G. J., and Chase, D. V.: Modeling Lake Outburst and Downstream Hazard Assessment of the Lower Barun Glacial Lake, Nepal Himalaya, J. Hydrol., 598, 126208, https://doi.org/10.1016/j.jhydrol.2021.126208, 2021.

3.   Lv, R. R., Tang, X. B., and Li, D. J.: Glacial lake outburst mudslide in Tibet, Chengdu University of Science and Technology Press, Chengdu, 69–105, 1999.

4.   Sun, M. P., Liu, S. Y., Yao, X. J., and Li, L.: The cause and potential hazard of glacial lake outburst flood occurred on July 5, 2013 in Jiali County, Tibet, Journal of Glaciology and Geocryology, 36, 158–165, https://doi.org/158-165,10.7522/j.issn.1000-0240.2014.0020, 2014.

5.   Liu, J. K., Zhou, L. X., Zhang, J. J., and Zhao, W. Y.: Characteristics of Jiwencuo GLOF, Lhari county, Tibet. Geological Review, 67: 17–18. https://doi.org/10.16509/j.georeview.2021.s1.007, 2021.

6.   Zheng, G. X., Mergili, M., Emmer, A., Allen, S., and Stoffel, M.: The 2020 glacial lake outburst flood at Jinwuco, Tibet: causes, impacts, and implications for hazard and risk assessment, The Cryosphere, 15, 3159–3180, https://doi.org/10.5194/tc-2020-379, 2021.

6 The paper forms a detailed and comprehensive assessment of the site and models some plausible GLOF inundation scenarios.   But this is essentially a paper about one lake and therefore could be criticised as being a bit parochial.   What does this say about moraine-dammed lakes more generally? Why is this paper significant enough to be published in a mainstream journal like The Cryosphere?   It therefore needs much more wider context,

and some sense of why the techniques you used are an advance on other similar work, or why you have provided new insights. Otherwise you have just provided an interesting case study.

**Reply:** At present, most of the studies on glacial lakes in the maritime glaciation zone of the southeastern Tibetan Plateau focus on the analysis of glacial lake changes and identification of potentially dangerous glacial lakes, etc. There are few or no published studies on glacial lake depth measurements, lake basin simulations, water volume estimation and the outburst flood simulations. Therefore, the work done in this study is of great importance for understanding the basin morphology and glacial lake depth, as well as estimating the water storage of the terminal moraine-dammed lake in the maritime glaciation zone. In addition, the simulation of GLOFs from ice avalanche and lateral moraine landslide to wave generation and propagation in the lake, dam erosion and breach, and downstream flood behavior based on the numerical models RAMMS and BASEMENT is carried out in this revision, and this whole process chain is implemented with a view to providing a research paradigm and reference for the simulation of other glacial lake breaches in the region in the future.

7 The title mentions water storage, but little is made of this in the paper. How does it compare with other lakes in the region?

**Reply:** Bathymetry data accurately reflects the basin morphology of the glacial lake, based on which the water storage capacity of the lake can be accurately calculated. In the first part of the results, we simulated the morphology of the glacial lake and estimated its water volume. However, the description of the title does not apply "water storage" but "volume", perhaps the former would be better. Comparisons with other lakes in the region are not covered because few actual bathymetric data for other glacial lakes in the region are currently publicly available. In addition, the section 5.2 Comparison of morphological characteristics for glacial lakes in continental and maritime glaciation regions in the Discussion is to explain that glacial lakes in the maritime glaciation zone may be deeper and therefore store more water by comparing with lakes with similar area in the continental glaciation zone.

8 Under the modelling scenarios presented here, some villages downstream will be completely inundated by the largest GLOF. What are the ethical issues that derive from such an analysis. I agree that we should prepare such assessments but the local inhabitants will be rightly concerned. I'm interested in their views of such analyses.

**Reply:** The two GLOFs that have occurred in this basin have both caused a serious disaster in the downstream region. For example, the outburst of Ranze Lake in July 2013 resulted in missing persons, destroyed houses, and serious damage to bridges, roads and other infrastructure, with direct economic losses of up to 270 million yuan. The outburst of Jinwu Co in June 2020 caused no casualties, but did cause significant damage to infrastructure (such as roads and bridges) and property damage in downstream areas. In addition, Bienong Co is a combination of glacier and glacial lake, the surrounding scenery is beautiful, and in the hearts of the local residents have some kind of religious beliefs representative. GLOFs disaster, therefore, is always an extreme concern for the local government, and during our survey, there were local government staff involved. And for the results of this study, it is a reference for the local government with a view to provide them with some theoretical help. For the local population, because of the presence of settlements, infrastructure and agricultural fields, based on the potential hazards generated by Bienong Co's GLOFS, they are most likely to use engineering measures such as lowering the water level to mitigate the potential threat.

Fig 8.    Difficult to differentiate colours in depth assessments.

**Reply:** We changed this expression in this revision.

Specific issues.

Line 9.    Omit 'the' after 'hazard to'

**Reply:** We are sorry for this grammar error and thank you for the tip.

Line 10 Insert 'the' before 'potential'.

**Reply:** We are sorry for this grammar error and thank you for the tip.

Line 11    Explain the typology of 'maritime'. These glaciers aren't maritime (meaning close to the sea).

**Reply:** Glaciers are a product of climate, and China has the largest amount of mountain glaciers in the world at low-and middle-latitude regions. Based on the development conditions and physical properties, glaciers in China are divided into continental glaciers and maritime glaciers. Continental glaciers are distributed from the Altai Mountains in the north, to the Northern Slope of the middle Himalayas in the south, from the Pamirs in the west, and to the Lenglongling and Anyemaqen Mountains in the east. They are characterized by low recharge, weak ablation, high snow line, low temperature, slow movement speed, and weak geological and geomorphological effects. According to the Second Chines Glacier Inventory, the glacier number (37,770) and area (46,200 km$^2$) account for 81. 4% and 77.8% of the total number and area in China, respectively. Maritime glaciers are mainly located in the eastern and southern slopes of the Himalayas, the eastern and middle sections of the Nyingchi Tanggula Range, and the Hengduan Mountains on the southeastern Tibetan Plateau. They are characterized by abundant recharge, strong ablation, low snowline distribution, high temperature, fast movement, and strong geological and geomorphological effects. The number of glaciers (8,607) and area (13,203 km$^2$) account for 18.6% and 22.2% of the number and area of glaciers in China respectively. Maritime glaciers are not distributed on the seashore, but their development and changes are controlled by the Indian monsoon. Table 1 shows the main features of continental glacier and maritime glacier in China. In this study, the Yi'ong Zangbo River Basin is located in the southeastern Tibetan Plateau and therefore is a maritime glaciation region.

Table 1 The main features of continental glacier and temperate glacier in China (Shi et al., 1964; Yang 1991; Su et al., 2000; Ding 2012)

| Glacier type | Climate | Average annual precipitation/ mm | Average annual temperature/ °C | Average summer temperature/ °C | Ice-forming effect | Ice temper-ature | Motion speed/ (m a$^{-1}$) |
|---|---|---|---|---|---|---|---|
| maritime glaciers | Indian Monsoonal circulation climate | 1000~3000 | >-6 | 1~5 | Warm soaking-Recrystallization | -1~0 | >100 |
| continental glaciers | Highland monsoon and continental climate | 200~1000 | <-6 | <3 | Soaking-Freezing | <-1 | 10~100 |

References:

1    Ding Yihui. *Environmental Characteristic of West China and Its Evolution. Beijing: Science Press. 2002: 166-173.*

2    Shi Yafeng, Xie Zichu. *The characteris- tics of existing glaciers in China. Acta Geographica Sinica. 1964. 30(3): 183-208*.

3    *Su Zhen. Shi Yafeng. Response of monsoonal temperate glaciers in China to global warming since the little ice age. Journal of Glaciolgy and Geocryology. 2000. 22(3): 223-229.*

4    *Yang Zhenniang. Glacier Water Resources in China. Lanzhou: Gansu Science and Technology Press, 1991.*

Line 12    Cite: Harrison, S., Kargel, J.S., Huggel, C., Reynolds, J., Shugar, D.H., Betts, R.A., Emmer, A., Glasser, N., Haritashya, U.K., Klimeš, J. and Reinhardt, L., 2018. Climate change and the global pattern of moraine-dammed glacial lake outburst floods. The Cryosphere, 12(4), pp.1195-1209.

**Reply:** Thank you for recommending this paper to us and we will make citations where appropriate.

Line 13     'such as the ice and/or rock avalanches'. There is no need for the definite article (ie 'the) when the noun is plural. This rule applies throughout this paper.

**Reply:** Thanks a lot for your guidance on our grammar. We will correct this situation throughout the text.

Line 14    Sentence starting 'Study shows…'. This sentence requires rewriting.    It is ambiguous and vague.    This is also an hypothesis and presupposes that we understand the link between climate change, glacier recession and GLOF incidence.

**Reply:** Unfortunately, our description of this sentence was so succinct that it confused the reader, and we will be rewriting the sentence.

Line 48 Re 'maritime glaciers'. Reword this sentence. Maritime is the wrong description of these glaciers. This means close to the sea.

**Reply:** Literally, maritime means close to the sea, and the glacier in the study is not near the sea. However, this writing style was adopted in published papers (such as the following two papers), so we used this writing style.

[Figure]

Line 51 Sentence starting 'Therefore'. I completely understand what you are trying to say here....but the use of language is incorrect.

**Reply:** Unfortunately, our description of this sentence was so succinct that it confused the reader, and we will be rewriting the sentence.

Line 71 Delete 'Whereas'.

**Reply:** Thank you for your tips, we have corrected it.

Line 82    I don't agree with this. USV are used quite a lot.    Lots of other examples.    Cite: Wilson, R., Harrison, S., Reynolds, J., Hubbard, A., Glasser, N.F., Wündrich, O., Anacona, P.I., Mao, L. and Shannon, S., 2019. The 2015 Chileno Valley glacial lake outburst flood, Patagonia. Geomorphology, 332, pp.51-65.    Also papers therein.

**Reply:** This is a reference from other people's research, and if you disagree with this statement, we can remove this statement.

Line 149  the data acquisition module, the data acquisition module. This is repeated.

**Reply:** Sorry for our carelessness, here is a small mistake that it should be the data acquisition module, the data transmission module.

Line 239  Why is the lake described as 'dangerous' if it is stable?

**Reply:** Here we may not have described it clearly enough. The stability of a glacial lake means that its area remains unchanged, which is an indication of its stability. But the surrounding factors of the glacial lake, such as the mother glacier, the surrounding moraine, and the climate change are unstable, and they may act on the glacial lake to cause the original stability of the glacial lake to be destroyed, thus inducing glacial lake outburst flood. To avoid ambiguity, it might be better to write *constant area* here.

Line 273  Where is the evidence that strong earthquakes can produce a full-depth incision in a terminal moraine?

**Reply:** A full-depth incision in a terminal moraine is informed by the hypothesis of Sattar et al. (2021) for Lower Barun Glacial Lake, for which they set the extreme scenario of the outburst incision reaching the bottom of the moraine (104 m) due to the initial erosion of the overtopping floods. We assume that this event may be caused by one or a combination of factors such as ice/avalanches, rock falls, landslides, heavy precipitation, or earthquakes.

Our assumption is designed to simulate the impact of an extreme-magnitude scenario glacial lake outburst flooding on the downstream region, but the possibility of such scenarios occurring is poorly considered. Reviewer I also questioned this issue. Therefore, we have paid enough attention to this situation and improved it. According to review 1's suggestion, we analyzed the whole process chain of GLOFs starting with detailed quantitative analysis of potential GLOFs triggers, following with the generation and propagation of waves in the lake caused by potential triggers, the erosion of the dam by overtopping floods, and the resulting GLOFs in the downstream.

1  *Sattar, A., Haritashya, U. K., Kargel, J. S., Leonard, G. J., and Chase, D. V.: Modeling Lake Outburst and Downstream Hazard Assessment of the Lower Barun Glacial Lake, Nepal Himalaya, J. Hydrol., 598, 126208, https://doi.org/10.1016/j.jhydrol.2021.126208, 2021.*

Line 296  the ~52.98 km downstream. Explain this.

**Reply:** We only study the impact of flooding in the downstream channel at a distance of about 53 km from Bienong Co.

---

## Author Response (AR1)

Dear Editor,

Based on the comments of two reviewers, we have made significant modifications to this study, mainly to extend the original simulation of GLOF propagation downstream based on the hypothetical dam failure to a process chain simulation of GLOFs triggered by material movements. Three model were used to simulated the GLOFs process chain, RAMMS model was used for simulation of potential mass movement (Christen et al., 2010), BASEMENT model was used to simulate the displacement wave in the lake, Heller-Hager model was used as a calibration for BASEMENT's results, and BASEMENT model was also adopted to simulate the dynamic breaching process of moraine dam, the propagation of flood wave and the inundation in downstream. We described the details of the GLOF process chain's simulation in the methodological section of the manuscript. Since this manuscript has changed very dramatically from the previous version, the version we uploaded in revision mode using Word software shows our changes in detail, so we do not explain them in detail in this note. Thank you for your guidance and help!

Best wishes!

---

## Author Response (AR2)

**Editor**

Dear editor,

Based on the comments of two reviewers, we have made significant modifications to this study, mainly to extend the original simulation of GLOF propagation downstream based on the hypothetical dam failure to a process chain simulation of GLOFs triggered by material movements. Three model were used to simulated the GLOFs process chain, RAMMS model was used for simulation of potential mass movement, BASEMENT model was used to simulate the displacement wave in the lake, Heller-Hager model was used as a calibration for BASEMENT's results, and BASEMENT model was also adopted to simulate the dynamic breaching process of moraine dam, the propagation of flood wave and the inundation in downstream. We described the details of the GLOF process chain's simulation in the methodological section of the manuscript. Based on your hints, we have responded to both reviewers' comments in point-by-point detail. Thank you for your tips and help, again.

Best wishes!

**Review#1**

Dear reviewer,

We sincerely thank you for your valuable comments on this study. After three months of work, we completed the first major revision of this manuscript according to your comments. The revised version is quite different from the initial version due to the addition of the simulation for glacial lake outburst floods (GLOFs) process chain. We first simulated the movement of triggers using the RAMMS model, GLOFs begin with ice avalanches or landslides. Secondly, we simulated and visualized the wave's propagation in the lake based on the output of the RAMMS model using the BASEMENT model. And then, the overtopping flow and erosion of the moraine dam was simulated using the BASEMENT model. Finally, the hydraulic behavior of GLOFs in the downstream region was also simulated using the BASENMENT model.

  In addition, we made some modifications in the **Discussion** section according to your and other reviewer's comments. To batter illustrate the potential danger and the typicality of Bienong Co, we moved the description of outburst potential characteristics of Bienong Co from the original **Discussion** section to the **Study area** section in this revision. Furthermore, we removed the content to developing an updated area-volume relationship for glacial lakes based on published bathymetry data due to the scarcity of samples. However, the peculiarity of Bienong Co was still clarified by comparing with glacial lakes with in-site measurement. Lastly, we made some clarifications of the shortcomings and uncertainties in the simulation of GLOFs.

Note:In this response, the black text is the reviewer's comments, and the blue text is our response where name of section is in boldface and the content in our revised manuscript is in italic font.

This study aims at estimating lake volume and modelling potential GLOF from the Lake Bienong, SE Tibet Plateau. In general, I'm convinced that such studies are needed and are logical step following region-wide GLOF susceptibility assessments. The outcomes might be of interest for research community as well as DRR practitioners.

The authors of this study employ broadly used data (Landsat images, ALSO PALSAR DEM, …) and methods (bathymetric surveying, empirical equations for deriving breach parameters, MIKE11 modelling tool) in new geographical context. As such, this study brings only limited novelty in terms of methodological development and to some extent only replicates the workflows of previous GLOF modelling studies of different lakes. I would maybe expect more novelty in leading journals such as the Cryosphere. Below I provide my comments to individual parts of this study:

The section about lake evolution actually presents no evolution and can be omitted or summarized in one sentence in the Introduction or Study area section in my opinion.

**Reply:** According to your suggestion, we present this part in **Study area** section. The description is "*The elevation of water surface in 2021 was 4745 m covering an area of 1.15 ± 0.05 km² that has experienced less significant changes. Mulang Glacier had an area of 8.29 ± 0.22 km² and mean surface slope of ~18.28˚, which has also remained largely unchanged area over the last 45 years. However, the glacier ablation zone experienced a thinning process of 6.5 m/a.*".

The authors defined 4 moraine dam breach (GLOF) scenarios which are then modelled in the MIKE11 and MIKE21 software. However, the moraine dam breach is not the beginning of the process chain, but a consequence of certain triggering event. Considering that this is detailed case study of only one lake, I would expect the authors to analyze the whole process chain in as much detail as possible, i.e. starting with detailed quantitative analysis of potential GLOF trigger which would help to define and justify dam breach scenarios.

However, this is not met in the current version of the manuscript. My major concern is that the breach scenarios are defined subjectively and are not linked to possible GLOF trigger identified in Section 5.1. What is described in Section 5.1. gives mainly qualitative overview of potential GLOF trigger. This is perhaps true for many lakes in the region, but there is no link to considered breach depth scenarios. And this is the major drawback of this study in my opinion.

**Reply:** By referring to the triggers of the collapsed glacial lakes on the Tibetan Plateau, we assumed two different types of triggers for potential GLOFs of Bienong Co, i.e., ice avalanches and landslides. Ice avalanches come from the tongue of the Mulang Glacier (the mother glacier of Bienong Co), and we assumed three different magnitudes. Landslides come from moraines on both sides of Bienong Co, and we assumed landslide movements at two different locations with different release depths. In this study, ice avalanches and landslides differ in the impulse to the glacial lake and the propagation of displacement waves in the lake caused by the different locations. The description in the manuscript is "*Based on a survey of the environment surrounding Bienong Co, ice avalanches from Mulang Glacier and two locations of lateral moraine landslide were selected as potential triggers for GLOFs. Wang et al., (2012) defined the volume of dangerous glacier as the volume from the location of abrupt changing slope to the glacier termini or the volume of glacier snout where ice cracks are well developed. We adopted the latter, i.e., the crevasse-developed ice body of Mulang Glacier shown in MapWorld image with a surface area of 0.19 km² was selected as the potentially ice avalanche source of Bienong Co (Fig. 1a). For the convenience of subsequent description, we name it Scenario A. The elevation difference between the top of the dangerous ice body and the lake surface was measured to be about 155 - 208 m based on ALOS PALSAR DEM. We divided the dangerous ice body into three parts according to elevation range to simulate subsequence processes from ice*

[revised manuscript text omitted]

For instance, I wonder what would need to be the magnitude of triggering slope movement to initiate 72 m deep breach? Is there any evidence that such slope movement could occur in the lake's surrounding? Landslide zones identified in Figure 11 don't seem to be releasing large mass volumes into the lake. Ideally, the initial slope movement, displacement wave propagation in the lake as well as dam breaching would be modelled, not only the GLOF. Critical questions regarding potential GLOF trigger are: Is there any evidence of mass movements entering the lake in the past? Have you observed any evidence from your analysis of remote sensing images and DEMs? Is there any evidence of potential future mass movements (displaced blocks, surface ruptures, opening crevasses, etc.)? Any evidence from your field work? Did any strong earthquake hit the region in the past? Did the lake experience any precipitation / temperature extremes? Do you expect them to change in the future? Considering the scope of the study (case study of one lake), individual trigger should be identified, quantified and treated more in depth in my opinion, feeding the definition of dam breach (GLOF) scenarios.

**Reply:** In the initial version, the GLOFs was based on the assumed break depth of moraine dam. We assumed an initiate 72 m deep breach of the moraine dam referring the study of Sattar et al. (2021) for the Lower Barun Glacial Lake, in which, the extreme-magnitude GLOFs of Lower Barun Glacial Lake was assumed that breach incision occurs until it reaches the base of the moraine (i.e., 104 m). Based on your suggestions, we identify and quantify the triggers of GLOFs from a more in-depth perspective in this revision. The Mulang Glacier is directly connected to Bienong Co, and the slope of the glacier tongue is steep and crevices are developed. Although, no ice avalanche

was observed in our field investigation, we found that collapse occurred at the glacier tongue by comparing the MapWorld image (earlier than 2020, but the exact date is not clear) (Fig 2-left) with the photos taken during the 2020 field survey (Fig 2-right). Earthquake-triggered glacial lake outburst floods are not common in Tibet, China, therefore we ignore this causative factor in this revision. Furthermore, due to the lack of reliable meteorological observations around the lake, there is no way to know whether the glacial lake has experienced extreme precipitation/temperature conditions. However, studies have shown that Jinwu Co, 24 km away from Bienong Co, experienced several days of continuous rainfall associated with the South Asian monsoon immediately prior to the outburst, which contributed to the formation of lateral moraine landslides on the one hand, and increased runoff at the spillway on the other hand. Based on your suggestions, we have simulated the initial slope movement, the propagation of displacement waves, the dam failure, and the downstream flooding behavior in this revision. The initial slope movement originated with the terminus of Mulang Glacier and the landslides of the moraines on both sides. The specific definition of scenarios for ice avalanche and landslide was described in the previous reply.

[Figure]

**Figure 2**. Comparison at the tongue of the mother glacier of Bienong Co in different periods (left: the MapWorld image early than 2020, but the exact date is unknown; right: taken in Aug 27 2020)

For the modelling part (Section 4.3, Table 3, Figures 7 to 10), flow velocities and peak discharge drop in Bada to 0.26 m/s (2,260 m$^3$/s) in the most extreme scenario, after which it again speeds up to 18.47 m/s (22,992 m$^3$/s) in Zongri is contra-intuitive and should be discussed / explained. In Figure 10, you even have negative peak discharge in Bada (?). Also the flow velocities of Scenario-1 (S1 to Jiawu; 44 to 65 m/s) seem unrealistically high considering it is supposed to originate from moraine dam breach.

**Reply:** The Bada village is the object of analysis because it is not located in the downstream channel of Bienong Co, but in another channel that intersects with the downstream channel of Bienong Co. However, in the extreme-magnitude scenario originally simulated, the Bada village was also affected by the huge flood. Therefore, the impact of flooding on it was analyzed. Because the Bada village is relatively far from the main channel of the flood, the flow and velocity are significantly reduced. The flood flows into the cross-section representing Bada village and then flows out in the opposite direction, i.e., exited the region, which is the reason for the negative value of the flow in the later period. In this revision, we fixed this problem by changing the cross section used to measure floods. Finally, the authors invested a lot of effort in comparing lake volumes (Section 5.2) and lake volume estimates (Section 5.3), but the implications of these comparisons are nebulous to me and the statistical treatment is incorrect. Considering the bathymetry done by the authors, it is clear that: (i) they have the best possible lake volume estimate for their study; (ii) their one bathymetry can hardly be used to validate or evaluate existing area-volume

relationships. Strikingly, bathymetric data used to develop the new area-volume relationship (Table 5, Figure 12) are then used for the performance assessment (Tabel 7, Figure 14), which is statistically not correct. Moreover, some of the existing equations (e.g. Fujita's equation developed specifically for Himalayan lakes) are not considered in this comparison. Further, the number of lakes listed in Table 5 is too low to generate any meaningful conclusions about a difference between lakes associated with continental and maritime glaciers. In addition, it seems that simply larger lakes are deeper as you only have one lake > 1 km$^2$ associated with continental glacier in your dataset while all lakes associated with maritime glaciers are > 1 km$^2$. This makes the Discussion section overall weak.

**Reply:** Thank you for the comment. In this revision, we removed the content about the development of area-volume relationship for glacial lakes based on published bathymetry data due to the scarcity of samples. However, information about the volume of water stored in this glacial lake is still stated. Firstly, *"Considering the rarity of bathymetric data but the frequent occurrence of GLOFs in the region, we try to explore more information about glacial lakes in the region by using bathymetry and water storage of Bienong Co. First, relationships with significant correlations for area-volume (Fig .3a), area-depth (Fig .3b) and depth-volume (Fig .3c) of Bienong Co were established (Fig. 3), and the valuable information is pinned on the hope that could provide a data reference for future studies of Bienong Co and other glacial lakes in the region."* Then, *"We compared the depth and water storage information of Bienong Co with other glacial lakes that have been measured."*. Lastly, *"We estimate the water storage of Bienong Co using published equations based on glacial lakes on the Tibetan Plateau, and the results show that the eight published volume-area/width-length relationships all underestimate the volume of Bienong Co to varying degrees (Tab.1)."* The results show that Bienong Co is the relative deepest glacial lake among these on the Tibetan Plateau that currently have been measured.

[Figure]

**Figure 3.** Fitting relationship of (a) area and volume, (b) area and depth and (c) depth and volume of Bienong Co.

**Table 1** Calculated volumes of Bienong Co based on published volume-area relationships for glacial lake in Tibetan Plateau.

| No | Source | | Relationships | Calculated Volume | Error (%) |
|---|---|---|---|---|---|
| 1 | Qi et al., 2022 | (1) | $V=0.04066A^{1.184}-0.003207w_{mx}/l_{mx}$ | $46.9 \times 10^6$ m$^3$ | -54% |
| 2 | | (2) | $V=0.0126A^2+0.0056A+0.0132$ | $36.3 \times 10^6$ m$^3$ | -65% |
| 3 | Wang et al., 2012 | | $V=0.0354A^{1.3724}$ | $42.9 \times 10^6$ m$^3$ | -58% |
| 4 | Sakai, 2012 | | $V=0.04324A^{1.5307}$ | $53.6 \times 10^6$ m$^3$ | -48% |
| 5 | Yao et al., 2012 | | $V=0.0493A^{0.9304}$ | $56.1 \times 10^6$ m$^3$ | -45% |
| 6 | Fujita et al., 2013 | | $V = 0.055A^{1.25}$ | $65.5 \times 10^6$ m$^3$ | -36% |
| 7 | Khanal et al., 2015 | | $V = 0.0578A^{1.5}$ | $71.3 \times 10^6$ m$^3$ | -30% |
| 8 | Zhou et al., 2020 | | $V=0.0717 w_{mx}^2 l_{mx}$ | $70.3 \times 10^6$ m$^3$ | -31% |

Note: Error = (Volume of empirical formulas − Bathymetrically derived volume) / Bathymetrically derived volume × 100%.

In additional, we illustrated some shortcomings and uncertainties in the simulation of GLOF. Firstly, "*Trigger is the beginning of the simulated GLOFs process chain in this study and we only consider ice avalanche and landslide scenarios, instead of other factors, such as increased glacial meltwater and heavy precipitation. The magnitude, location and probability of ice avalanche and landslide are the largest sources of uncertainty in this study. Ice avalanche is the trigger for over 70% of GLOFs on the Tibetan Plateau, but there is no reliable reference of the magnitude including release area and depth of previous ice avalanche events. Ice avalanche in this study come from the mother glacier tongue where the slope is relative steep and fissures are well-developed. We simulated three different-magnitude ice avalanches, each scenario assumes that the ice body breaks off in the vertical direction until to the lake surface, which is unrealistic and may overestimate the volume of ice avalanche. In addition, we also consider landslide as a trigger given the failure of Jinwu Co in 2020. Two release areas were selected by referring the slope and location of Jinwu Co's landslide, however, the release depth has no quantified reference data and we assumed three release depths of 2 m, 5 m and 10 m for each release area to simulate the consequences resulting from as many scenarios as possible.*" Secondly, "*the grain size distribution of moraine dam of Bienong Co was not obtained in this study, and the simulation was performed by referencing an inventory of glacial lakes in the Indian Himalaya. Although the data used have been validated to be reliably general, the grain size distribution of the moraine dam of Bienong Co itself would be more useful for an accurate simulation of moraine dam's erosion.*". Finally, "*DEM data are the most important basic data affecting the downstream propagation of GLOFs in this study. ALOS PALSAR DEM data with a spatial resolution of 12.5 m have been widely used in studies related to cryospheric changes and disasters. In this study, the DEM was pre-processed to fill sinks, but there was still the phenomenon of flood water being piled up in some deep puddle during the simulation, i.e., there were errors in the DEM data, especially in the relatively narrow valley. We have manually smoothed several large bumps according to the elevation of the upstream and downstream. However, there are still some smaller bumps that converge the flow to a section of the flow channel, mainly in the downstream area.*".

Table 4: if the lake has a surface outflow, dam freeboard = 0m.

**Reply:** Thank you for your tips, we used a freeboard of 0 m as the parameter for the simulation in the revised study.

Mean breach (Eq. 3) is used as max. breach in Figure 6; please check

**Reply:** In this revision, the dam-break process was re-simulated with the BASEMENT model, and Equation 3 and Figure 6 are discarded. We appreciate your comment.

I think that especially GLOF trigger need to be addressed in more detailed and quantitative way first, resulting in re-definition and justification of dam breach scenarios. Also the Discussion section should be re-worked substantially in my opinion. To sum up, I recommend major revisions of this manuscript.

**Reply:** In this revision, we have simulated the GLOFs process chain, including the initial slope movement, the generation and propagation of wave in the lake, dam breaching and the floods behaviors in the downstream channel starting from quantifying the trigger according to your suggestion. And, for the **Discussion** section, we focused on the water storage of Bienong Co and uncertainty analysis of GLOFs process chain simulation.

Review#2

Dear reviewer,

We sincerely thank you for your valuable comments on this study. After three months of work, we completed the first major revision of this manuscript according to your comments. The revised version is quite different from the initial version due to the addition of the simulation for glacial lake outburst floods (GLOFs) process chain. We first simulated the movement of triggers using the RAMMS model, GLOFs begin with ice avalanches or landslides. Secondly, we simulated and visualized the wave's propagation in the lake based on the output of the RAMMS model using the BASEMENT model. And then, the overtopping flow and erosion of the moraine dam was simulated using the BASEMENT model. Finally, the hydraulic behavior of GLOFs in the downstream region was also simulated using the BASENMENT model.

In addition, we made some modifications in the **Discussion** section according to your and other reviewer's comments. To batter illustrate the potential danger and the typicality of Bienong Co, we moved the description of outburst potential characteristics of Bienong Co from the original **Discussion** section to the **Study area** section in this revision. Furthermore, we removed the content to developing an updated area-volume relationship for glacial lakes based on published bathymetry data due to the scarcity of samples. However, the peculiarity of Bienong Co was still clarified by comparing with glacial lakes with in-site measurement. Lastly, we made some clarifications of the shortcomings and uncertainties in the simulation of GLOFs.

Note:In this response, the black text is the reviewer's comments, and the blue text is our response where name of section is in boldface and the content in our revised manuscript is in italic font.

This is an interesting paper that investigates a glacial lake and conducts analyses aiming to assess its evolution, basin morphology, estimate its water volume, analyses some possible outburst trigger analysis, and conducts simulations of likely inundation under GLOF scenarios. In general, the paper does a good job of developing these themes and forms a comprehensive case study that could be published given some reasonably substantial changes. I list these here and also some more specific issues with the paper.

Reply: We greatly appreciate your approval of this study and we have made some improvements in this revised version.1 The paper is often written in a rather vague and imprecise way.  In addition, there is often an incorrect use of English. I sympathise with the authors in this; it is difficult for non-English speakers to write precisely and accurately in English but this paper would benefit enormously from careful editing and rewriting by a native English speaker.

The literature review is generally comprehensive although there are some papers that should have been referenced and I highlight some of these later. The rationale for the study is clear and appropriate.

Reply: Thanks for your approval and your full understanding of our language issues. Combining Reviewer 1's and your comments, we made significant modifications to the manuscript.

3 The sampling strategy and methodology is not clearly discussed. Explain why this lake was chosen. Is this lake representative of others in the region? If so, why and how do you know this? If the lake is not representative, then the authors need to explain its significance.

Reply: Maritime glaciers and moraine-dammed glacial lakes are widely developed in the Southeastern Tibetan Plateau where several GLOFs occurred in recent years. Three GLOFs have occurred in the Yi'ong Zangbo Basin, and there are several glacial lakes assessed as having highly potential of failure, including Bienong Co. Bienong

Co has a relatively high potential of failure, mainly because it is directly connected to its mother glacier, which has a crevasse-developed and steeply sloping terminus. Secondly, both sides of Bienong Co widely distributed loose moraine material with slope greater than 30°, which could slide into the lake under continuous precipitation, thereby undermining the stability of the glacial lake. Jinwu Co, a glacial lake is only 24 km away from Bienong Co, and broke in 2020 due to an initial lateral moraine landslide. The above two features indicate that Bienong Co has the most common triggers that cause most of the GLOFs in Tibet, China, i.e., ice avalanches and landslides. Thirdly, Bienong Co has an area of above 1.1 $km^2$, which is a relatively large size for most glacial lakes. And the bathymetry data shows that it has a larger depth, indicating that a relatively large amount of water could be released. Fourthly, the moraine dam of Bienong Co is high and steep and consists of loose moraine material. Therefore, Bienong Co is considered to have a high potential of failure as well as be representative of the region. Meanwhile, Bienong Co has been paid more attentions by local government for many settlements in the downstream region, as well as the only road for this region.

4 The methodologies used are explained and justified well, and you have used an appropriate range of techniques to explore the geomorphology, characteristics and evolution of the lake and its future behaviour.

**Reply:** Thank you very much for your affirmation of the method adopted in this study. However, according to the comments of Reviewer 1, the simulation of this study still has some shortcomings. We adopted more comprehensive models and methods to simulate the process chain of GLOFs.

5 I am interested in why Bienong Co is regarded as a dangerous lake (lines 100 and 239)? The paper demonstrates that the lake has remained stable for some time, and that it cannot expand further. It also argues that the moraine dam does not contain an ice core. So the description of the lake as 'dangerous' requires much more discussion and evidence. This is important because there is always the temptation to describe any moraine - dammed lake as being 'dangerous' even when the evidence for this is lacking. I know of one well-known reviewer of similar papers who regularly rejects all papers who make this assertion without clear evidence!

**Reply:** Thank you for your comments. The outburst potential is controlled by multi factors, and Bienong Co has GLOF potential under the comprehensive evaluation in several literatures. The small areal variation and the absence of ice cores in the moraine dam are two features that are not conducive to breach. While, other factors such as the connection with the mother glacier, the steep slope and developed fissure at the mother glacier tongue, the loose moraine on both sides, the relatively large area and depth of the lake, and the high and steep moraine dam are factors that are conducive to the failure of Bienong Co. Therefore, we believe that Bienong Co is potential of failure when considered all factors together.

[revised manuscript text omitted]

6 The paper forms a detailed and comprehensive assessment of the site and models some plausible GLOF inundation scenarios. But this is essentially a paper about one lake and therefore could be criticised as being a bit parochial. What does this say about moraine-dammed lakes more generally? Why is this paper significant enough to be published in a mainstream journal like The Cryosphere?   It therefore needs much more wider context, and some sense of why the techniques you used are an advance on other similar work, or why you have provided new insights. Otherwise, you have just provided an interesting case study.

**Reply:** At present, most of the studies of glacial lakes in the maritime glaciation zone of the Southeastern Tibetan Plateau focus on the analysis of glacial lake changes and identification of potentially dangerous glacial lakes, etc. There are few or no published studies on glacial lake depth measurements, lake basin simulations, water volume calculation and the GLOFs simulation. Therefore, the work of this study is of great significance for understanding the basin morphology and glacial lake depth, as well as the water storage of the terminal moraine-dammed lake in the maritime glaciation zone. In addition, the simulation of GLOFs from ice avalanche and lateral moraine landslide to wave generation and propagation in lake, dam erosion and breach, and downstream flood behavior are carried out using the RAMMS Avalanche module and the BASEMENT model in this revision. The simulation of GLOFs process chain provides research paradigm and reference for the simulation of other glacial lake breaches in the region in the future.

7 The title mentions water storage, but little is made of this in the paper. How does it compare with other lakes in the region?

**Reply:** Bathymetry data accurately reflects the basin morphology of the glacial lake, based on which the water storage of a lake can be calculated. In **Results** section, we simulated the morphology of Bienong Co and estimated its water volume. However, the description of the title does not use "water storage" but "volume", perhaps the former would be better. Comparisons with other lakes in this region are not covered because few bathymetric data for other glacial lakes in the region are currently available. However, in this revision, we compared the water storage of Bienong Co with other glacial lakes having bathymetry data in the **Discussion** section and calculated the water storage of Bienong Co using the currently available area-volume relationship for glacial lakes in the Tibetan Plateau, and the results show that these formulas all underestimate the water storage of Bienong Co in different degrees, which reflects the larger relative depth of Bienong Co and the unique characteristics of glacial lakes in maritime glacierized zone.

8 Under the modelling scenarios presented here, some villages downstream will be completely inundated by the largest GLOF. What are the ethical issues that derive from such an analysis. I agree that we should prepare such assessments but the local inhabitants will be rightly concerned. I'm interested in their views of such analyses.

**Reply:** The two GLOFs that occurred in the Yi'ong Zangbo Basin have caused serious disasters in the downstream region. The outburst of Ranzeria Co in July 2013 resulted in missing persons, destroyed houses, and serious damage to bridges, roads, and other infrastructure, with direct economic losses of up to 270 million yuan. The outburst of Jinwu Co in June 2020 caused no casualties, but did cause significant damage to infrastructure (such as roads and bridges) and property damage in downstream areas. In addition, Bienong Co is a combination of glacier and glacial lake with beautiful scenery and is a representative of the local Tibetan religious beliefs. GLOFs disaster, therefore, is always an extreme concern for the local government, and during our survey, there was local government staff involved. This study could provide some theoretical help to local governments in understanding this glacial lake. For residents, due to the presence of settlements, infrastructure and farmland, engineering measures such as lowering water levels are most likely to be used to mitigate potential threats based on the potential hazards generated by Bienong Co's GLOF.

Fig 8.    Difficult to differentiate colours in depth assessments.

**Reply:** We changed this expression in this revision.

[Figure]

**Figure 12.** The potential threat of GLOFs to settlements and roads in the downstream under different ice avalanches and landslide scenarios (the background is MapWorld image).

Specific issues.

Line 9.    Omit 'the' after 'hazard to'

**Reply:** We are sorry for this grammar error and thank you for the tip.

Line 10 Insert 'the' before 'potential'.

**Reply:** We are sorry for this grammar error and thank you for the tip.

Line 11    Explain the typology of 'maritime'. These glaciers aren't maritime (meaning close to the sea).

**Reply:** China has the largest amount of mountain glaciers at low-and middle-latitude regions in the world. Based on the development conditions and physical properties, glaciers in China are divided into continental glaciers and maritime glaciers. Continental glaciers are distributed from the Altai Mountains in the north, to the Northern Slope of the middle Himalayas in the south, from the Pamirs in the west, and to the Lenglongling and Anyemaqen Mountains in the east. They are characterized by low recharge, weak ablation, high snow line, low temperature, slow movement speed, and weak geological and geomorphological effects. According to the Second Chinese Glacier Inventory, the glacier number (37,770) and area (46,200 km$^2$) account for 81. 4% and 77.8% of the total

number and area in China, respectively. Maritime glaciers are mainly located in the eastern and southern parts of the Himalayas, the eastern and middle sections of the Nyingchi Tanggula Range, and the Hengduan Mountains on the Southeastern Tibetan Plateau. They are characterized by abundant recharge, strong ablation, low snowline distribution, high temperature, fast movement, and strong geological and geomorphological effects. The number of glaciers (8,607) and area (13,203 km$^2$) account for 18.6% and 22.2% of the number and area of glaciers in China respectively. Maritime glaciers are not distributed on the seashore, but their development and changes are controlled by the Indian monsoon. Table 1 shows the main features of continental glacier and maritime glacier in China. In this study, the Yi'ong Zangbo Basin is located in the Southeastern Tibetan Plateau and therefore is a maritime glaciation zone.

Table 1 The main features of continental glacier and temperate glacier in China (Shi et al., 1964; Yang 1991; Su et al., 2000; Ding 2012)

| Glacier type | Climate | Average annual precipitation/ mm | Average annual temperature/ °C | Average summer temperature/ °C | Ice-forming effect | Ice temper-ature | Motion speed/ (m a$^{-1}$) |
|---|---|---|---|---|---|---|---|
| maritime glaciers | Indian Monsoonal circulation climate | 1000~3000 | >-6 | 1~5 | Warm soaking-Recrystallization | -1~0 | >100 |
| continental glaciers | Highland monsoon and continental climate | 200~1000 | <-6 | <3 | Soaking-Freezing | <-1 | 10~100 |

Reply: Thank you for recommending this paper to us and we will make citations where appropriate.

Line 13    'such as the ice and/or rock avalanches'. There is no need for the definite article (ie 'the) when the noun is plural. This rule applies throughout this paper.

Reply: Thanks a lot for your guidance on our grammar. We correct this situation throughout the text.

Line 14    Sentence starting 'Study shows…'. This sentence requires rewriting.    It is ambiguous and vague.    This is also an hypothesis and presupposes that we understand the link between climate change, glacier recession and GLOF incidence.

Reply: Unfortunately, our description of this sentence confused the reader. What we were trying to convey here is

that the area of Bienong Co is relatively large for most smaller glacial lakes. However, the area of the glacial lake has not undergone a dramatic expansion in the last 40 years. In this revision, we weakened the glacial lake evolution and emphasized the simulation of GLOFs process chain, so this sentence was removed from the abstract.

Line 48 Re 'maritime glaciers'. Reword this sentence. Maritime is the wrong description of these glaciers. This means close to the sea.

**Reply:** Literally, maritime means close to the sea, and the glacier in the study is not near the sea. However, this writing style was adopted in published papers (such as the following two papers), so we used this writing style.

[Figure]

Line 51 Sentence starting 'Therefore'. I completely understand what you are trying to say here....but the use of language is incorrect.

**Reply:** Unfortunately, our description of this sentence was so succinct that it confused the reader, and we rewrote the sentence.

Line 71 Delete 'Whereas'.

**Reply:** Thank you for your tips, we have corrected it.

Line 82 I don't agree with this. USV are used quite a lot. Lots of other examples. Cite: Wilson, R., Harrison, S., Reynolds, J., Hubbard, A., Glasser, N.F., Wündrich, O., Anacona, P.I., Mao, L. and Shannon, S., 2019. The 2015 Chileno Valley glacial lake outburst flood, Patagonia. Geomorphology, 332, pp.51-65. Also papers therein.

**Reply:** Bathymetric data of glacial lakes in China on the Tibetan Plateau are extremely scarce. In recent years, the unmanned vessels have been used for glacial lake bathymetry, but the extremely harsh natural environment and poor traffic conditions make this application still rare.

Line 149 the data acquisition module, the data acquisition module. This is repeated.

**Reply:** Sorry for our carelessness, here is a mistake that it should be the data acquisition module, the data transmission module.

Line 239 Why is the lake described as 'dangerous' if it is stable?

**Reply:** Here we may not have described it clearly enough. The stability of Bienong Co refers that its area remains unchanged. But the surrounding environment, such as the mother glacier, the lateral moraine, and climate are unstable, and they may act on the glacial lake to destroy the original stability, thus may inducing GLOFs. In this revision, we describe it as "*The area of Bienong Co has remained basically stable in the past 40 years, but it's area of 1.15 ± 0.05 km² in 2021 is almost twice the size of the two nearby failure glacial lakes, one is Jinwu Co*

[revised manuscript text omitted]

Line 273    Where is the evidence that strong earthquakes can produce a full-depth incision in a terminal moraine?

**Reply:** A full-depth incision in moraine dam was informed by the hypothesis of Sattar et al. (2021) for Lower Barun Glacial Lake, for which they set the extreme scenario of the outburst incision reaching the bottom of the moraine (104 m) due to the initial erosion of the overtopping floods. Our assumption is to simulate the impact of an extreme-magnitude GLOF on the downstream region, but the occurring possibility of the scenario is poorly considered. Reviewer 1 also questioned this assumption. Therefore, we paid enough attention to this situation and improved it. According to Reviewer 1's suggestion, we simulated the GLOF process chain starting with detailed quantitative analysis of potential trigger, followed by the generation and propagation of waves in the lake, the erosion of the dam by overtopping floods, and the resulting flood in the downstream region.

Line 296    the ~52.98 km downstream. Explain this.

**Reply:** We only study the impact of GLOFs in the downstream channel within the distance of ~53 km from Bienong Co.

---

## Author Response (AR3)

**Reply to editor**

Dear editor,

We are very pleased that this revision has been approved by you and the two reviewers. At the same time, we thank you for your comments on the revision of this manuscript. We value the comments made by you and the two reviewers and have responded to them carefully. Details are as follows:

1) All points raised by Reviewer #1 seem very pertinent and should be addressed. Two comments seem particularly important: (i) The uncertainties in Section 5.2 have indeed to be discussed. Whenever possible, this discussion should be quantitative, although I understand that a full error propagation might be difficult to implement. In such a case, even more emphasis should be set on the qualitative discussion. (ii) The comment for Lines 499-500, pointing at a questionable generalization, is important too. If a speculation is strictly necessary in your view (personally, I don't think this is the case), it should be clearly flagged as such.

**Reply:** Thank you for the comments given on this study. We sincerely accept the two constructive comments given by reviewer 1 and have made the best effort to revise them. Uncertainty quantification is indeed a very difficult task. In the process of doing literature research in this study, it is found that there are relatively few studies on the quantification of errors in each link of the GLOF process chain, and most of the studies focus on the establishment of the simulation process and the elaboration of the results. Therefore, this part was weakened in our pre-simulation process. However, this part is very important for scientific research. In this revision, we combined your comments and those of reviewer 1, we described and highlighted potential uncertainties in the simulations in the different links of the GLOF process chain modelling by reviewing a large amount of literature, but it is only qualitatively. In addition, for the problem in Lines 499-500, we agree with the reviewers' comments, and we removed the last sentence of illustrating a difference of glacial lakes between Himalaya and SETP.

2) Reviewer #2 explicitly asks for the language to be improved. This advice should be followed, possibly with the help of external support. In terms of specific wording, the reviewer had also commented on the use of the term "maritime", pointing out that the wording seems inadequate for the addressed study side. I agree with this appreciation, in the sense that the location seems a prime example for what one would call

"continental", and not "maritime". Please change this wording throughout the manuscript in your next submission.

**Reply:** We sincerely accept the comments of reviewer 2, who raised this issue in the first revision. But due to the huge workload of the GLOF process chain simulation in the major revision, we did not have more time to make improvements to the language. In this revision, we invited Mr. Jake Carpenter from USA to improve the language of this paper. We hope that this revision will make up for the shortcomings of our language. In addition, we replaced the word "maritime" with "temperate" which is also the wording commonly used in the literature to describe glaciers in our study region.

3) I noticed that your manuscript did not contain a "Code and data availability" statement so far. Please include such a statement in your revision, as this is required by TC's guidelines (https://www.the-cryosphere.net/submission.html). When preparing the statement, please make sure to respect TC's Data Policy (https://www.the-cryosphere.net/policies/data_policy.html).

**Reply:** We are sorry for not noticing this issue and we addressed it in this revision. "***Code and data availability***: *The Landsat MSS/TM/OLI image are available from the United States Geological Survey (https://earthexplorer.usgs.gov/). The AST14DEM dataset and ALOS PALSAR DEM used in this study can be obtained from the National Aeronautics and Space Administration (NASA) EARTHDATA website (https://earthdata.nasa.gov/). The MapWorld image is provided by the National Platform for Common Geospatial Information Services (https://www.tianditu.gov.cn/). The GLC10 LULC product is available from http://data.ess.tsinghua.edu.cn/fromglc10_2017v01.html.*"

**Reply to Referee #1**

Dear reviewer,

We are very happy to see your approval of the revised manuscript. Thank you for your guidance over the past months. We valued the excellent comments you have made on this study and made corresponding modifications.

1. Section 4.2.4 – it would be nice to how many man-made structures (buildings, bridges, km of roads, etc.) are exposed to individual GLOF scenarios.

**Reply:** Following your suggestion, we summarized the count of man-made structures (buildings, bridges, km of roads, etc.) are exposed to individual GLOF scenarios. "*The*

*total areas of houses, courtyards, and farmlands (around settlement) affected by Scenarios A1, A2, A3, B3, and C3 were estimated to be 23,984 m², 32,076 m², 41,038 m2, 3,820 m² and 3,918 m², respectively. In Scenarios A1, A2, A3, all 13 bridges and road with a length of approximately 35 km are within the flood zones, and in Scenarios B3 and C3, there are four bridges as well as with a length of approximately 3.6 km and 6.7 km within the flood zones. This study only assesses the potential impact of floods on these man-made structures, but the magnitude of the impact is beyond the scope of this study.*"

2. Section 5.2 – the uncertainties are not really addressed here; I suggest you to discuss the uncertainties of individual components (models) in your model chain and how they propagate. The (dis)advantages compared to the single model approach (e.g. r.avaflow) should be discussed.

**Reply:** We sincerely accept your constructive comment and have made the best effort to revise them. Uncertainty quantification is indeed a very difficult task. In the process of doing literature research in this study, it is found that there are relatively few studies on the quantification of errors in each link of the GLOF process chain, and most of the studies focus on the establishment of the simulation process and the elaboration of the results. Therefore, this part was weakened in our pre-simulation process. However, this part is very important for scientific research. In this revision, we described and highlighted potential uncertainties in the simulations in the different links of the GLOF process chain modelling by reviewing a large amount of literature, but it is only qualitatively. We know that you are a well-known expert in the field of glacial lake outburst floods, and we hope you can give some advice if we have not written well enough in this section. In any case, thank you very much for your great help in this study.

3. Terminology–you use 'eruption', 'breakout' and 'outburst' throughout the text – I suggest to unify and use only one of those (preferably 'outburst'); similarly, I suggest to use 'lake volume' instead of 'water storage'

**Reply:** Thank you for your advice on the writing guidelines, we standardized the terminology in the text as you suggested.

4. L140-141: please check the correctness of the calculation of the width-to-height ratio (I think you used width perpendicular to the streamline while width parallel to the streamline should be used in the calculation)

**Reply:** Thank you for your guidance. As you would think, we calculated the aspect

ratio using the width perpendicular to the flow line. Based on your suggestion and with reference to Prakash and Nagarajan (2017) (Figure 1), we re-measured the dam width of Bienong Co (Figure 2). The dam height of Bienong Co is smaller than its maximum water depth, we measured the distance from the inside to the outside of the dam parallel to the flow direction at a lake depth of 72 m as the dam width (Figure 2).

[Figure]

**Figure 5**. Factors for outburst susceptibility: (1) lake area, (2) increase in lake area, (3) lake–glacier proximity (horizontal distance), (4) slope between lake and glacier, (5) moraine width-to-height ratio, (6) width of moraine crest, (7) slope of distal face of moraine, (8) lake freeboard, (9) mass movement impact on lake, (10) Cloudburst or extreme rainfall event (11) seismicity

Figure 1 Illustration of moraine dam width in study of Prakash and Nagarajan (2017).

[Figure]

Figure 2 Dam with of Bienong Co.

5. L499-500: this shows that the two lakes are different, you don't have enough data to generalize about a difference between Himalaya and SETP

**Reply:** In view of the suggestion you raised, we removed the last sentence of illustrating a difference of glacial lakes between Himalaya and SETP.

**Reply to Referee #2**

Dear reviewer,

We are very grateful for your review and comments on this paper. You raised the issue

of the readability of this manuscript in English in your first review. Unfortunately, we did not manage to improve the language due to the significant workload in model simulation and the time constraints in the last overhaul. In this revision, we invited a native English-speaking expert to revise and improve the language of this manuscript. Thank you again for your recognition and help. I was fortunate to have been given the opportunity to visit the University of South Wales in the UK with official support from China. However, I am now experiencing problems with the visa and I am working to resolve this issue. If I ever get the chance to come to the UK I hope to come and visit you and ask for your knowledge of glacial lakes. Thank you again. Best wishes to you.

1. This paper is now fine. Previously I made some changes to the written English. There are still some small problems here and there. Please do not start sentences with 'And' (line 20; decide whether to use the present or past tense in your sentences (e.g. line 69). Line 152: 'it's' means 'it is' and is different from the possessive 'its'.

**Reply:** We are very sorry for our grammatical problems. In this revision, we invited a native English-speaking expert to check our language problems in the hope of making substantial improvements to the English grammar of this manuscript.

2. You should use the definite article (The) before nouns. For example, 'Bathymetric' in line 251 should have 'The' before it. There are numerous examples throughout the paper.

**Reply:** I am sorry that I have a poor understanding of the use of "the" in English expressions. In this revision, I will try to avoid such problems as far as possible.

3. Please cite: Harrison, S., Kargel, J.S., Huggel, C., Reynolds, J., Shugar, D.H., Betts, R.A., Emmer, A., Glasser, N., Haritashya, U.K., Klimeš, J. and Reinhardt, L., 2018. Climate change and the global pattern of moraine-dammed glacial lake outburst floods. The Cryosphere, 12(4), pp.1195-1209. This should be around line 59.

**Reply:** Thank you for recommending this article to us, it is very interesting and we gained a lot of knowledge from it. I have cited it in this article.

---

## Author Response (AR4)

**Dear editor,**

First, we apologize for the inappropriate wording related to the language enhancement of the manuscript in the last reply. We have valued every comment you and the two reviewers made. But we are sorry that the wording in our last reply may be inappropriate, giving you the impression that we did not take the language issue seriously. In our previous revision, we did invite a native English scholar to assist us in revising the language of this article. Unfortunately, there are still some problems in the revised version. We appreciate your detailed revision tips for our abstract, according to which we edited the abstract carefully. Furthermore, we carefully examined the language of this article and modified many inappropriate wordings and sentences. We explained every revised sentence in this version in the point-by-point reply. We hope that the revised version is more reasonable and better.

In addition, I encountered difficulty in this revision. I did not complete the work timely and efficiently because I was sick. China ended her management and control policy for COVID-19 in early December 2022. Many people were quickly infected with the virus, including me. I was treated in the hospital for many days because my symptoms were severe. Therefore, the revision of the article was delayed for some days. I sincerely apologize for the inconvenience caused to TC's editorial team by not submitting the revised article on time and not explaining my situation to the editorial team to request sufficient time for revision. We sincerely hope that the peer-review process for this article will continue. Thank you again. Best wishes to you. Happy New Year.

**Abstract**

**Line 1:** Please change the title into "Lake volume and potential hazards of moraine-dammed glacier lakes — A case study of Bienong Co, southeastern Tibetan Plateau" (The point is that both the words "glacial / glaciation" and "temperate" seem to be used in a wrong way here)

**Reply:** We appreciate and accept your suggestion to change the title. However, "glacial lake" is a more common usage than "glacier lake", and is used in most studies (Figure 1). Therefore, we used "glacial lake" in this study and the title is "Lake volume and potential hazards of moraine-dammed glacial lakes—A case study of Bienong Co, Southeastern Tibetan Plateau". Thank you again.

[Figure]

**Figure 1.** Two studies of glacial lakes.

**Line 11:** Replace "some" with "such", change "flood" to "floods", delete "events of high-magnitude".

**Reply:** Thank you for your tips. We made changes accordingly.

**Line 12:** Change "end moraine-dammed glacial lakes" to "moraine-dammed glacial lakes".

**Reply:** Thank you for your tip. We have made the change accordingly.

**Line 13:** Add "the" before "SETP".

**Reply:** Thank you for your tip. We made the change accordingly.

**Line 13:** Change "simulated" to "determined".

**Reply:** Thank you for your tip. We made the change accordingly.

**Line 14-15:** This sentence does not make sense. From what I understood, this sentence should be reworded into "Then we analysed scenarios for possible lake outbursts. These scenarios included the possibility of GLOFs being triggered by ice avalanches entering the lake (Scenarios A1-3) or by landslides from the lateral moraines (Scenarios B1-3 and C1-3)."

**Reply:** Thank you for your tip. Based on your suggestion and our consideration, this sentence has been revised to "These scenarios included the possibility of GLOFs being triggered by ice avalanches (Scenarios A1-3) from the mother glacier or by landslides from the lateral moraines (Scenarios B1-3 and C1-3)."

**Line 16:** "The volume" of what? Do you mean "The GLOF volume"? Or what volume is meant here? It can't be the volume of the lake, obviously.

**Reply:** Thank you for your tip. It is the avalanche volume. We made the change accordingly. This sentence has been revised to "Avalanche volumes of the nine trigger scenarios were obtained from the RAMMS modeling results.".

**Line 16:** Add "the" before "nine trigger scenarios".

**Reply:** Thank you for your tip. We made the change accordingly.

**Line 16:** well, to my knowledge, RAMMS provides simulations of mass movement trajectories, not "volumes of trigger scenarios". This sentence needs to be amended.

**Reply:** Thank you for your tip. We changed the sentence to "Avalanche volumes of the nine trigger scenarios were obtained from the RAMMS modeling results.".

**Line 17:** Please split the sentence into two (or more) separate messages: with four "and", the sentence is impossible to follow.

**Reply:** Thank you for your tip. The sentence was rewritten as "Next, the BASEMENT model was used to simulate the generation and propagation of the avalanche-induced displacement waves in the lake. With the model, the overtopping flows and erosion on the moraine dam and the subsequent downstream floods were also simulated.".

**Line 18:** Change "demonstrate" to "indicate" and change "produce" to "may cause"

**Reply:** Thank you for your tip. We made the change accordingly. The sentence was modified as "The results indicate that the ice avalanche scenario may cause the largest mass volume entering the lake, resulting in a displacement wave up to 25.2 m in amplitude (Scenario A3) near the moraine dam.".

**Line 19:** Add "entering" before "into", change the sentence "resulting in displacement wave amplitudes of up to 25.2 m" to "resulting in a displacement wave up to 25.2 m in height".

**Reply:** Thank you for your tip. We made the change accordingly. But here we want to express the wave amplitude (the parameter "a" in Figure 2), not the wave height (the parameter "H" in Figure 2). Therefore, the sentence is correct as "resulting in a displacement wave up to 25.2 m in amplitude".

[Figure]

**Figure 2.** Principal wave parameters presented on an idealized sine wave.

**Line 20:** Delete "near the moraine dam".

**Reply:** Thank you for your tip. Here we cannot delete "near the moraine dam". The reason is that the wave amplitude of 25.2 m is indeed the highest value near the moraine dam, and the highest value in the entire lake is 66.4 m.

**Line 20:** Change "Smaller volumes of landslides" to "Smaller landslides volumes", delete "only". Delete "in the lake, such as that Scenario C1 has a wave amplitude below 1 m near the moraine dam.". please remove this part of the sentence. Since the Scenarios were not properly introduced, there is little value in telling what the wave amplitude is -- it is an arbitrary number.

**Reply:** Thank you for your tip. We made the change accordingly. This sentence was modified as "Landslide scenarios with smaller volumes entering the lake result in smaller displacement waves.".

**Line 22:** Add "volumes" behind "water".

**Reply:** Thank you for your tip. We made the change accordingly. This sentence was modified as "Scenarios A1, A2, and A3 result in released water volumes from the lake of $24.1 \times 10^6$ m³, $25.3 \times 10^6$ m³, and $26.4 \times 10^6$ m³, respectively.".

**Line 22:** Change ", and" to ". Corresponding". Add "are" behind "dam".

**Reply:** Thank you for your tip. We made the change accordingly.

**Line 23:** There is only one (singular) dam, not several, are there? Change "dams" to "dam".

**Reply:** Thank you for your tip. There is only one dam. We corrected the mistake.

**Line 24:** Here and everywhere else in the document: please check the number of significant digits -- it is not credible to me that the modeling is able to predict the breach dept to an accuracy of 10 cm. This, however, is what the provided number of significant digits implies...

**Reply:** This is written based on statistics. However, this accuracy may be impractical in simulation. The sentence is corrected as "These high discharges cause erosion of the moraine dam, resulting in breach widths of 295 m, 339 m, and 368 m, respectively, with the generally similar breach depth of approximately 19 m." Thank you again.

**Line 24:** Delete "However" and change "in" to "In".

**Reply:** Thank you for your tip. We made the change accordingly.

**Line 25:** Change "caused" to "cause" and delete "moderate".

**Reply:** Thank you for your tip. We made the change accordingly.

**Line 26:** Add "According to our simulations," before "GLOFs".

**Reply:** Thank you for your tip. We made the change accordingly.

**Line 27:** Delete "can" and change "and well threaten" to "thereby threatening".

**Reply:** Thank you for your tip. We made the change accordingly. But here, we remove the "thereby" because we consider that the "threatening" has the meaning of indicating a result.

**Line 28:** Change "produced floods" to "produce GLOFs".

**Reply:** Thank you for your tip. We made the change accordingly.

**Line 28:** I don't understand this addition: is it meant to be an indication of time? If so, (1) this should be clarified, and (2) the same information should be provided for the preceding sentence for reasons of consistency.

**Reply:** Thank you for your tip. This is an indication of time. We provided the time indication for the preceding sentence.

**Line 28:** here and elsewhere: please check the tenses. Some sentences ago the used tense switched from present to past, and it is unclear why. Please ensure consistency (here, you need "have")

**Reply:** Thank you for your tip. It should be "have". We are sorry for the inconsonant tenses. In this revision, we checked the tense of the whole text.

**Line 29:** A comparison is always between something and something else. So what are the compared elements in this case?

**Reply:** Thank you for your tip. We changed the sentence as "Comparisons of the area, depth, and volume of glacial lakes for which the bathymetry data are available show that...".

**Line 29:** A comparison is always between something and something else. So what are the compared elements in this case? why "relative"? What is the meaning of this word here? A lake is either the deepest or not -- there is no relative way of casting that...

**Reply:** Thank you for your tip. The compared elements in this case are the area, depth, and volume of glacial lakes for which the bathymetry data are available. The sentence is revised as "Comparisons of the area, depth, and volume of glacial lakes for which the bathymetry data are available show that Bienong Co is the known deepest glacial lake with a same surface area on the Tibetan Plateau.".

**Line 30:** Please keep separate messages in separate sentences, i.e. start a new sentence here. There is no relation between Bienong Co being the deepest known lake and the fact that the study provided new insights, is there? I mean: new insights would have been provided also if the lake happened to be the third deepest, isn't it?

**Reply:** Thank you for your tip. We made the change accordingly. The sentence was modified as "Comparisons of the area, depth, and volume of glacial lakes for which the bathymetry data are available show that Bienong Co is the known deepest glacial lake in the same surface area on the Tibetan Plateau. This study could provide a new insight into moraine-dammed glacial lakes in the SETP and a valuable reference for GLOF disaster prevention for the local government.".

**Introduction**

**Line 41-43:** Once the dam is damaged, the water can be suddenly and catastrophically released to form glacial lake outburst floods (GLOFs), which may cause severe social and geomorphic impacts several dozens of kilometers and more downstream.

**Reply:** In this sentence, we removed the "and catastrophically" and re-edited the sentence after "which". This sentence was modified as "Once the dam is damaged, the water can be suddenly released to form glacial lake outburst floods (GLOFs), which could have a significant downstream social and geomorphic impact across several dozens of kilometers.".

**Line 44:** Moraine-dammed glacial lakes are of particular concern due to their large volume (Fujita et al., 2013; Veh et al., 2019), weak dam composition, and exposure to various triggers, such as ice and/or rock avalanche, heavy precipitation, and intense glacier melting (Emmer and Cochachin, 2013; Nie et al., 2018), which are the most common sources of GLOFs (Watanbe and Rothacher, 1996; Westoby et al., 2014).

**Reply:** In this sentence, the "ice and/or rock avalanches" was changed to "ice or rock avalanche", the "melting" was changed to "meltwater", and "which are the most common sources of GLOFs (Watanbe and Rothacher, 1996; Westoby et al., 2014)" was deleted. This sentence was modified as "Moraine-dammed glacial lakes are of particular concern due to their vast volumes (Fujita et al., 2013; Veh et al., 2019), weak dam composition, and exposure to various triggers, such as ice or rock avalanches, heavy precipitation, and intense glacier meltwater (Emmer and Cochachin, 2013; Nie et al., 2018).".

**Line 48:** The Himalayas and the Southeastern Tibetan Plateau (SETP) are regions of the frequent occurrence of GLOFs caused by moraine-dammed glacial lakes (Wang, 2016).

**Reply:** In this sentence, "the frequent occurrence of GLOFs" was changed to "frequent GLOFs". This sentence was modified as "The Himalayas and the Southeastern Tibetan Plateau (SETP) are regions of frequent GLOFs caused by moraine-dammed glacial lakes (Wang, 2016).".

**Line 49:** Research shows that the Himalayas, especially the southern region, are likely to experience more GLOFs in the coming decades (Veh et al., 2019).

**Reply:** This sentence was modified as "Veh et al. (2019) deemed that the Himalayas, particularly the southern region, would likely experience more GLOFs in the coming decades.".

**Line 52:** The SETP is a broad mountainous area covering the central and eastern Nyainqêntanglha Ranges, eastern Himalayas and western Hengduan Mountains, and has highly complicated terrains (Ke et al., 2014).

**Reply:** This sentence was modified as "The SETP is a broad mountainous region with highly complicated terrains, including the central and eastern Nyainqêntanglha Ranges, the eastern Himalayas, and the western Hengduan Mountains (Ke et al., 2014)".

**Line 52:** Controlled by warm and humid Indian monsoons, a large number of temperate glaciers have developed here (Yang et al., 2008), featured as adequate recharge, strong ablation, low snowline distribution, high temperature, fast movement, and strong geological, as well as geomorphological, effect (Li et al., 1986; Qin et al., 2012; Liu et al., 2014), which have been observed with markedly negative mass balances during

the past decades (Kääb et al., 2012; Neckel et al., 2014; Kääb et al., 2015; Brun et al., 2017; Dehecq et al., 2019).

**Reply:** This sentence was modified as "A substantial number of temperate glaciers have developed here under the influence of the warm and humid Indian monsoons (Yang et al., 2008). These glaciers have characteristics of adequate recharge, intense ablation, low snowline distribution, high temperature, fast movements, and strong geomorphological effects (Li et al., 1986; Qin et al., 2012; Liu et al., 2014). In the past decades, glaciers in this region have undergone significant negative mass balances (Kääb et al., 2012; Neckel et al., 2014; Kääb et al., 2015; Brun et al., 2017; Dehecq et al., 2019).".

**Line 61:** Studies of glacial lakes in the SETP have mainly focused on the regional-scale assessments of glacial lake changes.

**Reply:** This sentence was modified as "Studies of glacial lakes in the SETP have mainly focused on the regional-scale assessments of glacial lake changes, …".

**Line 69:** Zheng et al. (2021) analyzed and reconstructed a GLOF process chain of Jinwu Co…

**Reply:** In this sentence, the "analyzed and" was deleted. This sentence was modified as "Zheng et al. (2021) reconstructed the GLOF process chain of Jinwu Co…".

**Line 69:** As a key factor related to the peak discharge and outburst volume of a GLOF event (Evans, 1987; Huggel et al., 2002), lake volume is difficult to directly obtain by means of satellite remote sensing approach.

**Reply:** This sentence was modified as "Lake volume is a key factor related to the peak discharge and drainable volume of a GLOF event (Evans, 1987; Huggel et al., 2002), but it is difficult to obtain using a remote sensing approach directly.".

**Line 72:** Currently, due to the easy availability of area information from remote sensing images, the volume of glacial lakes is generally estimated using the developed empirical formulas to connect glacial lake area and volume based on bathymetric data for a small number of glacial lakes (O'Connor et al., 2001; Huggel et al., 2002; Yao et al., 2014).

**Reply:** This sentence was modified as "Currently, the volume of a glacial lake is generally estimated using empirical formulas and its area information from remote sensing images. Empirical formulas linking the area and volume of glacial lakes were generally developed based on the bathymetry data of a small number of glacial lakes (O'Connor et al., 2001; Huggel et al., 2002; Yao et al., 2014).".

**Line 78:** Although the SETP region is an area with a high incidence of GLOFs (Sun et al., 2014; Zheng et al., 2021; Zhang et al., 2023), there have been few publicly available bathymetric data of glacial lakes and related research works.

**Reply:** This sentence was modified as "Despite the high incidence of GLOFs in the SETP region (Sun et al., 2014; Zheng et al., 2021; Zhang et al., 2023), there are few available bathymetry data of glacial lakes.".

**Line 80:** Previous bathymetric works…

**Reply:** This wording was modified as "The previous bathymetry works…".

**Line 84:** This is unfavorable to…

**Reply:** This wording was modified as "It is unfavorable to…".

**Line 86:** …in certain scenarios…

**Reply:** This wording was modified as "…in specific scenarios…".

**Line 87:** …moraine surveys…

**Reply:** This wording was modified as "…marine surveys…".

**Line 88:** Glacial lakes are mostly located at high altitudes and in harsh environments (Zhang et al., 2020), and USVs make the measurement of the underwater topography of glacial lakes safer, more convenient, and more accurate (Li et al., 2021).

**Reply:** This sentence was modified as "Glacial lakes are mainly located at high altitudes and in harsh environments (Zhang et al., 2020). The application of USVs makes the underwater topography measurement of glacial lakes more accurate, efficient, and safer (Li et al., 2021).".

**Line 91:** an end moraine-dammed glacial lake

**Reply:** This wording was modified as "a moraine-dammed glacial lake".

**Line 93:** First, the lake basin morphology of Bienong Co is modelled.

**Reply:** This sentence was modified as "First, we determined the lake basin morphology of Bienong Co.".

**Line 93:** Then, multiple components of the GLOF process chain, including initial mass movement from the mother glacier and lateral moraine slope, displacement wave generation and propagation in the lake, overtopping flow and erosion on the moraine dam, and subsequent downstream flooding were simulated.

**Reply:** This sentence was modified as "Then, we simulated the GLOF process chain starting with the initial mass movements from the mother glacier and the lateral moraines, the generation and propagation of displacement waves in the lake, the overtopping flows and erosion on the moraine dam, and the subsequent downstream floods.".

**Line 96:** This study will assist the local government to understand the potential hazards of Bienong Co and serve as a reference for other scholars studying glacial lakes and GLOFs in the SETP region.

**Reply:** In this sentence, the wording "…to understand…" was modified as "…in understanding…". This sentence was modified as "This study will assist the local government in understanding the potential hazards of Bienong Co and serve as a reference for studying glacial lakes and GLOFs in the SETP region.".

**Study area**

**Line 102:** The terrain is high in the west and low in the east with high mountains and valleys.

**Reply:** This sentence was modified as "With tall mountains and deep valleys, the terrain is high in the west and low in the east.".

**Line 103:** The climate is warm and humid, featuring a mean annual precipitation of 958 mm and a mean annual temperature of 8.8 °C (Ke et al., 2013, 2014).

**Reply:** This sentence was modified as "The region experiences a warm and humid climate with a mean annual precipitation of 958 mm and a mean annual temperature of 8.8 °C (Ke et al., 2013, 2014).".

**Line 105:** There was 1,907.76 km$^2$ glacier coverage, and 105 moraine-dammed glacial lakes with a total area

of 16.87 km$^2$, in 2016 (Duan et al., 2020).

**Reply:** This sentence was modified as "In 2016, there were 105 moraine-dammed glacial lakes with a total area of 16.87 km$^2$ in the Yi'ong Zangbo basin (Duan et al., 2020).".

**Line 107:** Seven glacial lakes in the watershed, including Bienong Co, were considered to have high GLOFs potential (Duan et al., 2020), of which Jinwu Co collapsed on June 26, 2020 (Zheng et al., 2021).

**Reply:** In this sentence, "GLOFs potential" was changed to "GLOF potential", and "collapsed" was changed to "failed". This sentence was modified as "Seven glacial lakes in the basin, including Bienong Co, were considered to have high GLOF potential (Duan et al., 2020), of which Jinwu Co failed on June 26, 2020 (Zheng et al., 2021).".

**Line 108:** As of 2021, there have been three recorded large GLOF events in the basin, all of which caused very significant damage to infrastructures in the downstream region (Sun et al., 2014; Yao et al., 2014; Zheng et al., 2021) (Fig. 1b).

**Reply:** This sentence was modified as "Three catastrophic GLOFs have occurred in the Yi'ong Zangbo basin as of 2021, all of which significantly damaged infrastructures in the downstream (Sun et al., 2014; Yao et al., 2014; Zheng et al., 2021) (Fig. 1b).".

**Line 111:** an end moraine-dammed glacial lake

**Reply:** This wording was modified as "a moraine-dammed glacial lake".

**Line 112:** a massive unconsolidated terminal moraine dam on the northwest

**Reply:** The wording "terminal" was removed.

**Line 112:** The elevation of the water surface in 2021 was 4,745 m covering an area of 1.15 ± 0.05 km$^2$ that has experienced less significant changes.

**Reply:** This sentence was modified as "In 2021, it had an area of 1.15 ± 0.05 km$^2$ and a surface elevation of 4,745 m.".

**Line 113:** The Mulang Glacier had an area of 8.29 ± 0.22 km$^2$ and a mean surface slope of ~18.28°, which has also remained a largely unchanged area over the last 45 years.

**Reply:** This sentence was modified as "The Mulang Glacier, which has remained stable in area over the past 45 years, had an area of 8.29 ± 0.22 km$^2$ with an average surface slope of ~18.28° in 2021.".

**Line 115:** However, the glacier ablation zone experienced a thinning process of 6.5 m/a.

**Reply:** This sentence was modified as "The glacier ablation zone, however, underwent an intense thinning at a speed of -6.5 m/a.".

**Line 116:** The flow of Bienong Co converges into Xiong Qu ("Qu" means "river" in Tibetan), which is one of the two main tributaries of the upper Yi'ong Zangbo (Fig. 1b).

**Reply:** This sentence was modified as "One of the major tributaries of the upper Yi'ong Zangbo, Xiong Qu ("Qu" means "river" in Tibetan), receives the outflow from Bienong Co (Fig. 1b).".

**Line 118:** The flow channel from Bienong Co to the confluence of Xiong Qu and Song Qu (another main tributary of the upper Yi'ong Zangbo) stretches ~53 km, with a river longitudinal drop ratio of 14.48‰.

**Reply:** This sentence was modified as "The flow channel from Bienong Co to the confluence of Xiong Qu

and Song Qu (the other major tributary of the upper Yi'ong Zangbo) is ~53 km long with a longitudinal drop ratio of 14.48‰.".

**Line 120:** There are 18 settlements and 13 bridges densely distributed along the flow channel, as well as a large amount of agricultural land.

**Reply:** This sentence was modified as "There are 18 settlements, 13 bridges, and a substantial area of cultivated land along the flow channel.".

**Line 120:** In addition, the Jiazhong Highway extends closely along the river (Fig. 1d).

**Reply:** This sentence was modified as "Additionally, the Jiazhong Highway extends adjacent to the river (Fig. 1d).".

**Line 129:** The area of Bienong Co hasremained basically stable in the past 40 years, but its area of $1.15 \pm 0.05$ km$^2$ in 2021 was almost twice the size of the two nearby outburst moraine-dammed glacial lakes, one of which is Jinwu Co (Zheng et al., 2021) and the other is Ranzeria Co (Sun et al., 2014; Zhang et al., 2022), which were located just 24 km and 9 km southeast of Bienong Co, respectively.

**Reply:** This sentence was modified as "The area of Bienong Co has changed minimally over the past 40 years. However, the area of $1.15 \pm 0.05$ km$^2$ in 2021 was nearly twice as large as the two nearby outburst moraine-dammed glacial lakes, Jinwu Co (Zheng et al., 2021) and Ranzeria Co (Sun et al., 2014; Zhang et al., 2022), which are only 24 km and 9 km to the southeast of Bienong Co, respectively.".

**Line 133:** The moraine dam of Bienong Co has an average height of 72 m, enclosing a lake volume of $65.2\times10^6$ m$^3$, accounting for 64% of the total (Fig. 2e).

**Reply:** This sentence was modified as "With an average height of 72 m, Bienong Co's moraine dam encloses a water volume of $65.2\times10^6$ m$^3$, accounting for 64% of the total lake volume (Fig. 2e).".

**Line 133:** Overall, the greater is the volume of water retained in the lake, the greater is the volume of water available for potential flooding (Westoby et al., 2014), and the greater is the hazard caused by GLOFs.

**Reply:** This sentence was modified as "Overall, the greater the volume of water retained in the lake, the greater the volume of water available for a potential flood (Westoby et al., 2014), and the greater the hazard caused by a GLOF.".

**Line 136:** GLOFs are highly complex phenomena, each of which constitutes a distinctly unique event with characteristics determined by the triggering mechanism, lake hypsometry, geometry, composition, and structural integrity of the moraine dam, as well as the topography and geology of the flood path (Westoby et al., 2014).

**Reply:** This sentence was modified as "GLOFs are highly complicated phenomena, each being a unique individual event influenced by triggers, lake geometries, the composition and structural integrity of the moraine dam, and the topography and geology of the flow path (Westoby et al., 2014).".

**Line 139:** According to historical GLOF studies, the most common cause of glacial lakes' failure in the Himalayas is mass movement (snow, ice, or rock) entering lakes (Richardson and Reynolds, 2000; Wang et al., 2012a; Emmer and Cochachin, 2013; Worni et al., 2014), and subsequently overtopping and eroding of the moraine dam (Risio et al., 2011).

**Reply:** This sentence was modified as "According to historical GLOF studies, the most common cause of glacial lakes' failure in the Himalayas is overtopping and erosion of moraine dams (Risio et al., 2011) triggered by mass movements (snow, ice, rock, or mixtures) entering lakes (Richardson and Reynolds, 2000; Wang et al., 2012a; Emmer and Cochachin, 2013; Worni et al., 2014).".

**Line 143:** Bienong Co is directly connected to the Mulang Glacier, whose ablation zone is defined as the mother glacier tongue in this study, and has an average slope of 20° with well-developed ice crevasses (Fig. 2a and b).

**Reply:** This sentence was modified as "The Mulang Glacier directly connects Bienong Co; its tongue (here is the ablation zone) contains well-developed ice crevasses with an average slope of 20° (Fig. 2a and b).".

**Line 146:** Lv et al., (1999) proposed that a slope of mother glacier tongue greater than 8° is conducive to the occurrence of ice avalanche. In the context of global warming, glacial meltwater can lubricate the glacier itself, increasing the likelihood of overhanging ice sliding into the lake (Wang et al., 2015).

**Reply:** This sentence was modified as "According to Lv et al. (1999), ice avalanches are more likely to occur when the slope of the mother glacier tongue is greater than 8°. Moreover, climate warming-induced glacial meltwater can lubricate glaciers, increasing the possibility of overhanging ice sliding into lakes (Wang et al., 2015).".

**Line 149:** Therefore, ice disintegration from the Mulang Glacier could be a potential trigger for GLOFs of Bienong Co.

**Reply:** This sentence was modified as "Therefore, the Mulang Glacier's ice disintegration may be a potential trigger of the GLOF of Bienong Co.".

**Line 150:** In addition, the GLOF of Jinwu Co was caused by an initial moraine landslide with a slope range of 30° - 45° on the left side (Zheng et al., 2021).

**Reply:** This sentence was modified as "Additionally, the GLOF of Jinwu Co was triggered by an initial moraine landslide with a slope range of 30° - 45° on the left side (Zheng et al., 2021).".

**Line 151:** Bolch et al. (2011) and Rounce et al. (2016) both reported that non-glacierized areas around a lake with a slope > 30° are potential rock fall, landslide, or other solid mass movement regions.

**Reply:** This sentence was modified as "According to Bolch et al. (2011) and Rounce et al. (2016), non-glacierized areas around lakes with a slope greater than 30° are potential locations of rock fall, landslide, or other solid mass movements.".

**Line 153:** There are multiple locations with lateral moraines around Bienong Co that fit into this slope range (Fig. 2c, d, and e). Therefore, lateral moraine landslides could also be a potential trigger for Bienong Co's GLOF.

**Reply:** This sentence was modified as "Around Bienong Co, multiple moraine-covered places fit into the slope range (Fig. 2c, d, and e). Therefore, lateral moraine landslides may also be a potential trigger of Bienong Co's GLOF.".

**Line 156:** Dam characteristics, such as dam geometry (freeboard, width-to-height ratio, distal face slope), dam material properties, and ice-cored moraine conditions, govern the stability of the dam (Huggel et al., 2004;

Prakash and Nagarajan, 2017; Wang et al., 2011a).

**Reply:** This sentence was modified as "The stability of a moraine dam is affected by several parameters, including the geometry features (freeboard, width-to-height ratio, distal face slope), the material property, and the ice-cored condition (Huggel et al., 2004; Wang et al., 2011a; Prakash and Nagarajan, 2017).".

**Line 158:** Freeboard refers to the vertical distance between the lake level and the lowest point on the dam crest, which reflects the minimum requisite wave amplitude for the occurrence of overtopping, and a higher freeboard is not conducive to the occurrence of overtopping (Emmer and Vilímek, 2014).

**Reply:** This sentence was modified as "Freeboard is the vertical distance between the lake surface and the lowest point on the dam crest, which reflects the minimum wave amplitude necessary for overtopping to occur (Emmer and Vilímek, 2014). A higher freeboard does not favor the occurrence of overtopping.".

**Line 161:** A natural outlet with a width of ~50 m is in the right of the dam (facing downstream) (Fig. 2e and f), indicating that the freeboard of Bienong Co is 0 m, which signals the high potential for overtopping of the lake.

**Reply:** This sentence was modified as "A natural outlet with a width of ~50 m is on the right of the moraine dam (facing downstream) (Fig. 2e and f). As a result, the freeboard of Bienong Co is 0 m, indicating a high potential of overtopping.".

**Line 163:** The moraine dam is ~520 m wide and the height is variable with an average height of ~72 m, and the width-to-height ratio is 7.2 (Fig. 2e).

**Reply:** This sentence was modified as "The moraine dam has a width of ~520 m and an average height of ~72 m, giving the width-to-height ratio of 7.2 (Fig. 2e).".

**Line 168:** However, the freeboard of 0 m and the distal facing slope of 35˚ are conditions that are conducive to GLOFs based on favorable thresholds of smaller than 25 m (Mergili et al., 2011) and larger than 20˚ (Lv et al., 1999).

**Reply:** This sentence was modified as "However, the freeboard of 0 m and the distal facing slope of 35˚ are instability conditions of the moraine dam based on the favorable thresholds of smaller than 25 m (Mergili et al., 2011) and larger than 20˚ (Lv et al., 1999).".

**Line 169:** The moraine dam of Bienong Co is covered with vegetation, the surface layer is a larger particle size of the stone, and below the smaller particle size, the material is loose and poorly cemented, which is susceptible to destruction by water forces (Fig. 2e).

**Reply:** This sentence was modified as "The surface layer of the moraine dam of Bienong Co is made of larger-grained stones covered with vegetation. Tiny granules of loose moraines make up the lower layer, which is easily destroyed by water force (Fig. 2e).".

**Line 172:** The existence of ice core inside of the moraine dam is unknown, but there is no ice core in Jinwu Co's breached dam. The dam crest elevation of Bienong Co is 320 m higher than that of Jinwu Co.

**Reply:** This sentence was modified as "The existence of ice cores within the moraine dam of Bienong Co is unknown. However, there is no one in Jinwu Co's breached dam, which is lower ~320 m than the former.".

**Line 174:** Additionally, McKillop and Clague (2007) argued that moraines with rounded surfaces and minor

superimposed ridges are considered ice-cored, whereas narrow, sharp-crested moraines with angular cross-sections are interpreted as ice-free, and the dam of Bienong Co clearly fits the latter category.

**Reply:** This sentence was modified as "Furthermore, according to McKillop and Clague (2007), moraine dams with rounded surfaces and minor superimposed ridges are considered ice-cored; in contrast, narrow, sharp-crested moraines with angular cross-sections are regarded as ice-free. The moraine dam of Bienong Co fits the latter category.".

**Line 177:** In aggregate, we consider that the potential threats to Bienong Co are mainly ice avalanches from the mother glacier and lateral moraine landslides.

**Reply:** This sentence was deleted in this revision.

**Methodology**

**Line 187:** Lake bathymetric information is one of the most important inputs in the dynamic modeling of GLOFs, and can accurately reflect the topography of the lake basin below the water surface and used to calculate the potential flood volume released in different breach scenarios (Westoby et al., 2014).

**Reply:** This sentence was modified as "Lake bathymetry data can be used to depict the underwater topography and calculate the lake volume, which is one of the most critical parameters for the simulation of GLOF process chain (Westoby et al., 2014)." in this revision.

**Line 190:** In this study, the depth data were obtained by a USV (APACHE 3) system, …

**Reply:** In this sentence, "the depth data" was changed to "the bathymetry data". This sentence was modified as "In this study, the bathymetry data of Bienong Co was obtained using a USV (APACHE 3) system, …"

**Line 195:** The D230 Single-Frequency Depth Sounder mounted on the USV is designed to measure a range of …

**Reply:** This sentence was modified as "The D230 Single-Frequency Depth Sounder mounted on the USV has a measurement range of…"

**Line 198:** Field measurements were carried out on August 27, 2020.

**Reply:** This sentence was modified as "Field investigation was carried out on August 27, 2020."

**Line 201:** …and our survey was conducted at a speed of 2 m/s for a total route of 22.58 km in Bienong Co.

**Reply:** This sentence was modified as "…and our survey was carried out at a speed of 2 m/s over a total route of 22.58 km in Bienong Co.".

**Line 202:** Due to absence of any obstructions on the lake, such as ice or small islands, the high performance of the USV and the real-time monitoring, the survey was accurately completed along the designed route.

**Reply:** This sentence was modified as "With no obstruction on the lake, the USV's strong performance and the real-time monitoring allowed the survey to be accurately completed along the designed route.".

**Line 203:** A total of 16,020 valid sounding points, essentially covering the entire glacial lake, were measured, which well fulfilled the data density requirement to model the lake basin topography (Fig.3c).

**Reply:** This sentence was modified as "A total of 16,020 valid sounding points were measured, essentially

covering the entire glacial lake, which fulfilled the data density requirement to model the lake basin topography of Bienong Co (Fig.3c).".

**Line 207:** The bathymetric map…

**Reply:** This wording was modified as "The DEM of the lake basin…".

**Line 209:** Lake capacity can be understood as the volume of water storage below a certain water level, which is the volume between a certain spatial curved surface and a certain horizontal surface (Shi et al., 1991).

**Reply:** This sentence was modified as "Lake volume can be understood as the volume between a certain spatial curved surface and a certain horizontal surface (Shi et al., 1991).".

**Line 211:** In this study, the volume of Bienong Co was obtained by multiplying the depth data and map resolution (5 m) as follows:

**Reply:** This sentence was modified as "In this study, the volume of Bienong Co was obtained by multiplying the depth value and the DEM resolution (5 m) as follows:".

**Line 215:** …the bathymetric map…

**Reply:** This wording was modified as "the DEM of the lake basin…".

**Line 221:** …the assessment of glacial lake hazards should be carried out based on a systematic and scientific analysis of…

**Reply:** In this sentence, we deleted the "carried out".

**Line 223:** The methodology used in this study refers to the GLOF process chain proposed by Worni et al. (2014), which has been utilized by Somos-Valenzuela et al. (2016) and Lala et al. (2018) to study Imja Tsho in Nepal and Palcacocha and Huaraz lakes in Peru, respectively.

**Reply:** This sentence was modified as "The methodology of this study is based on the GLOF process chain proposed by Worni et al. (2014), which was used by Somos-Valenzuela et al. (2016) and Lala et al. (2018) to research Imja Tsho in Nepal and Palcacocha and Huaraz lakes in Peru, respectively.".

**Line 225:** In this study, we aim to depict potential GLOFs induced by ice avalanches originating from the Mulang Glacier (Fig. 2a and b) and landslides from the lateral moraine (Fig. 2c and d), and assess potential inundation in the downstream region.

**Reply:** This sentence was modified as "In this study, we intend to depict the potential GLOFs triggered by ice avalanches from the mother glacier (Fig. 2a and b) and landslides from the lateral moraines (Fig. 2c and d).".

**Line 228:** The wave resultant from material entering Bienong Co might overtop the moraine dam and initiate an erosive breaching process, …

**Reply:** This sentence was deleted.

**Line 230:** Three models were used to simulate the GLOFs process chain: the RAMMS model was used for simulation of potential mass movement (Christen et al., 2010); the BASEMENT model was used to simulate the displacement wave in the lake; and the Heller-Hager model was used as a calibration for BASEMENT's results.

**Reply:** This sentence was modified as "Three models were used to simulate the GLOF process chain: the Rapid Mass Movement Simulation (RAMMS) model (RAMMS, 2017) was used to simulate the mass

movements; the Basic Simulation Environment for Computation of Environmental Flow and Natural Hazard Simulation (BASEMENT) model (Vetsch et al., 2022) was used to simulate the displacement waves in the lake; and the Heller-Hager model was used to calibrate the BASEMENT modeling results.".

**Line 235:** The BASEMENT model was also adopted to simulate the dynamic breaching process of the moraine dam, the propagation of the flood wave, and the inundation downstream.

**Reply:** This sentence was modified as "The BASEMENT model was also adopted to simulate the dynamic breaching process of the moraine dam and the downstream inundation.".

**Line 236:** In the next sections, simulation methods for ice avalanches and landslides, displacement waves in lakes, overtopping flow and erosion on the moraine dam, and downstream inundation are described.

**Reply:** This sentence was modified as "In next sections, simulating methods are described in detail.".

**Line 239:** Triggers determination and simulation

**Reply:** This wording was modified as "Trigger determination and simulation".

**Line 239:** Mass movements into lakes generate impulse waves that may produce overtopping flow scouring and erosion of moraine dams, or disrupt the hydrostatic pressure-bearing capacity of moraine dams.

**Reply:** This sentence was modified as "Displacement waves caused by mass movements entering a lake might result in overtopping flows that destroy the moraine dam or disrupt the hydrostatic pressure-bearing capacity of the moraine dam (Somos-Valenzuela et al., 2016).".

**Line 242:** Based on a survey of the environment surrounding Bienong Co, ice avalanches from the Mulang Glacier and lateral moraine landslides at two locations were selected as potential triggers for GLOFs.

**Reply:** This sentence was modified as "Based on the investigation of Bienong Co, ice avalanches from the mother glacier and landslides from the lateral moraines at two locations were selected as potential triggers of GLOFs.".

**Line 244:** Wang et al. (2012) defined the volume of a dangerous glacier as the volume from the location of the abrupt changing slope to the glacier termini or the volume of the glacier tongue where ice cracks are well developed.

**Reply:** This sentence was modified as "Wang et al. (2012) defined the dangerous ice body of a glacier as either the volume from the location of the abrupt change in slope to the glacier termini or the volume of the glacier tongue where ice cracks are well developed.".

**Line 246:** We adopted the latter, i.e., the crevasse-developed ice body of the Mulang Glacier shown in the MapWorld image with a surface area of $0.19 \text{ km}^2$ was selected as the potential ice avalanche source of Bienong Co (Fig. 2a).

**Reply:** This sentence was modified as "In this study, the crevasse-developed ice body of the Mulang Glacier with a surface area of $0.19 \text{ km}^2$, as shown in the MapWorld image (Fig. 2a), was selected as the potential ice avalanche source.".

**Line 248:** For convenience of subsequent description, we name it Scenario A.

**Reply:** This sentence was modified as "For convenience of the subsequent description, the ice avalanche-triggered situation was named after Scenario A.".

**Line 250:** We divided the dangerous ice body into three parts according to elevation range to simulate subsequence processes from ice avalanches of different magnitudes (ice avalanche 1, 2, and 3 in Fig. 2b).

**Reply:** This sentence was modified as "We divided the dangerous ice body into three parts according to elevation ranges as ice avalanche scenarios of different magnitudes (ice avalanche 1, 2, and 3 in Fig. 2b).".

**Line 252:** Scenario A1 was defined as a low-magnitude trigger, and the ice body at an elevation below 4,844 m yields a release area of 0.05 km$^2$ with the maximum and average elevation differences of 99 m and 75.8 m from the lake surface, respectively.

**Reply:** This sentence was modified as "Scenario A1 was defined as a low-magnitude trigger; the ice body with an elevation below 4,844 m yields a release area of 0.05 km$^2$ with the maximum and the average elevation differences of 99 m and 75.8 m from the lake surface, respectively.".

**Line 254:** Scenario A2 was defined as a moderate-magnitude trigger, ice body at elevation below 4,889 m yields a release area of 0.11 km$^2$ with the maximum and average elevation differences of 144 m and 102.7 m from the lake surface, respectively.

**Reply:** This sentence was modified as "Scenario A2 was defined as a moderate-magnitude trigger; the ice body below 4,889 m yields a release area of 0.11 km$^2$ with the maximum and the average elevation differences of 144 m and 102.7 m from the lake surface, respectively.".

**Line 256:** Scenario A3 was defined as an extreme-magnitude trigger; the total ice body of crevasse with an area of 0.19 km$^2$ was set as a release area, with the average elevation difference between glacier surface and lake surface of 131 m.

**Reply:** This sentence was modified as "Scenario A3 was defined as an extreme-magnitude trigger; the total ice body of crevasse with an area of 0.19 km$^2$ was set as a release area, with the average elevation difference between the glacier surface and the lake surface of 131 m.".

**Line 258:** In the above three cases, we assumed that the release depths of ice avalanches are the average elevation differences from the glacier surface to the lake surface, i.e., the glacier is supported by flat bedrock located at the height of the lake water table.

**Reply:** This sentence was modified as "In the above three scenarios, we assumed that the release depths of ice avalanches are the average elevation differences from the glacier surface to the lake surface, i.e., the glacier is supported by a flat bedrock located at the height of the lake surface.".

**Line 261:** Lateral moraine landslides as a GLOFs trigger are not common on the Tibetan Plateau, but the GLOF of Jinwu Co in 2020 was caused by a lateral moraine landslide (Liu et al., 2021; Zheng et al., 2021), and thus was taken as a trigger of the potential GLOF for Bienong Co.

**Reply:** This sentence was modified as "Lateral moraine landslide as the GLOF trigger is not common on the Tibetan Plateau, but the GLOF of Jinwu Co in 2020 was caused by a lateral moraine landslide (Liu et al., 2021; Zheng et al., 2021). Therefore, it was taken as a trigger of the potential GLOF of Bienong Co.".

**Line 263:** Two areas of lateral moraine within the slope range of 30˚ - 45˚ were selected as potential landslide sites, one of which is located on the left bank (in this study, the left and right sides are defined in a downstream-oriented manner) of Bienong Co, near the moraine dam with an area of 0.015 km$^2$, and we named it Scenario

B (Fig. 2c).

**Reply:** This sentence was modified as "Two locations of lateral moraines within the slope range of 30° - 45° were selected as potential landslide sites. One is located on the left bank of Bienong Co, near the moraine dam with an area of 0.015 km$^2$, and we named it Scenario B (Fig. 2c). The other is located on the right bank, near the mother glacier with an area of 0.024 km$^2$, we named it Scenario C (Fig. 2d).".

**Line 269:** …and nine different magnitudes of materials were designed to enter the lake as potential triggers for GLOFs in this study.

**Reply:** This wording was modified as "and nine different magnitudes of mass movements entering the lake were designed as the potential triggers for the GLOF of Bienong Co.".

**Line 271:** The above design fully considers the impact of triggers on Bienong Co under different magnitudes, and the results are used as the input for the subsequent disaster chain simulation.

**Reply:** This sentence was modified as "The above design took account of the impact of triggers of different magnitudes on Bienong Co, and the results were used as the input for the subsequent simulation.".

**Line 273:** In this study, ice avalanches and lateral moraine landslides of Bienong Co were modeled using the Avalanche module of the Rapid Mass Movement Simulation RAMMS model (Bartelt et al., 2013), which has been successfully used for simulating triggers of GLOFs (Somos-Valenzuela et al., 2016; Lala et al., 2018; Sattar et al., 2021).

**Reply:** In this sentence, "were modeled" was modified as "were simulated" and "Rapid Mass Movement Simulation" was deleted because it has been mentioned before. This sentence was modified as "In this study, ice avalanches and lateral moraine landslides in Bienong Co were simulated using the Avalanche module of the RAMMS model (Somos-Valenzuela et al., 2016; Lala et al., 2018; Sattar et al., 2021).".

**Line 275:** RAMMS adopts…

**Reply:** This wording was modified as "It adopts…".

**Line 277:** Based on the basic inputs of DEM, the initial release area and depth, the calculation domain, the friction parameters $\mu$ (the velocity-independent dry Coulomb) and $\xi$ (velocity-dependent turbulent friction terms), the outputs of runout distances, flow height, and flow velocity can be calculated.

**Reply:** This sentence was modified as "Based on the inputs of DEM data, the initial release area and depth, calculation domain, friction parameters $\mu$ (the velocity-independent dry Coulomb term) and $\xi$ (the velocity-dependent turbulent term), the RAMMS model can calculate the runout distance, flow height, and flow velocity of mass movements.".

**Line 279:** In addition, the time series of material entering the glacial lake can serve as the input condition for subsequent simulations.

**Reply:** This sentence was modified as "Then, the time series of the volume of mass entering the lake was taken as the input for subsequent simulation.".

**Line 280:** For this study case, the initial release area was determined by combining the MapWorld image with a spatial resolution of 0.5 m (https://www.tianditu.gov.cn/) and ALOS PALSAR DEM with a spatial resolution of 12.5 m (https://asf.alaska.edu/data-sets/derived-data-sets/alos-palsar-rtc/alos-palsar-radiometric-terraincorrection/).

**Reply:** This sentence was modified as "In this study, the initial release area was determined by combining the MapWorld image (0.5 m) (https://www.tianditu.gov.cn/) and ALOS PALSAR DEM (12.5 m) (https://asf.alaska.edu/data-sets/derived-data-sets/alos-palsar-rtc/alos-palsar-radiometric-terrain-correction/).".

**Line 285:** …which agree with values used in previous GLOF-producing avalanche models (Schneider et al., 2014; Somos-Valenzuela et al., 2016).

**Reply:** In this sentence, "GLOF-producing avalanche models" was modified as "avalanche-induced GLOFs studies". This sentence was changed to "…which agree with values used in previous avalanche-induced GLOF studies (Schneider et al., 2014; Somos-Valenzuela et al., 2016).".

**Line 293:** The simulation of hydrodynamic waves in the lake is performed using the 2D modeling of BASEMENT based on unstructured grids.

**Reply:** In this sentence, "the 2D modeling of BASEMENT" was modified as "the BASEMENT 2D module".

**Line 294:** The BASEmesh plugin for QGIS (QGIS Development Team, 2016) developed by BASEMENT greatly facilitates the generation of mesh.

**Reply:** This sentence was modified as "The BASEmesh plugin in QGIS software (QGIS Development Team, 2016) greatly facilitates the generation of mesh.".

**Line 295:** The lake bathymetry data were entered into DEM with a spatial resolution of 5 m using ArcGIS Pro software to reflect the lake basin topography.

**Reply:** This sentence was modified as "The bathymetry data was produced as a DEM with a spatial resolution of 5 m using ArcGIS Pro software to reflect the topography of Bienong Co's lake basin.".

**Line 297:** The triangular irregular network (TIN) within the lake area was set to a maximum area of 500 m$^2$ to simulate the generation and propagation of hydrodynamic waves in the lake effectively and accurately.

**Reply:** In this sentence, "within the lake area" was modified as "within the lake" and "hydrodynamic" was modified as "displacement".

**Line 298:** The input boundary conditions are time series of ice avalanches and landslides generated by RAMMS model.

**Reply:** This sentence was modified as "The inflow boundary condition is the time series of ice avalanches or landslides obtained from the RAMMS modeling results.".

**Line 299:** In each time period, the RAMMS calculates the total amount of sediment, and the inflow rate can be determined by calculating the difference of sediment entering the lake at two time points.

**Reply:** This sentence was modified as "The RAMMS model can calculate the sediment volume for each timestep. The time series of ice avalanches or landslides can be determined by counting the sediment volume within the lake boundary at different timesteps.".

**Line 303:** Pure rock landslides have been investigated with densities ranging from 1,950 kg m$^{-3}$ to 2,200 kg m$^{-3}$ (Wang et al., 2017), and most ice-dominated avalanches have densities of approximately 1000 kg m$^{-3}$.

**Reply:** This sentence was modified as "The density of pure rock landslides ranges from 1,950 kg m$^{-3}$ to 2,200

kg m$^{-3}$ (Wang et al., 2017), and most ice-dominated avalanches have a density of approximately 1000 kg m$^{-3}$.".

**Line 304:** In this study, the ice avalanche density was set as 1,000 kg m$^{-3}$, and the landslide density was set as 2,000 kg m$^{-3}$.

**Reply:** This sentence was modified as "In this study, the densities of ice avalanches and landslides were set as 1,000 kg m$^{-3}$ and 2,000 kg m$^{-3}$, respectively.".

**Line 306:** Since BASEMENT only accepts water as an inflow, this difference due to density is considered by expanding the landslide material entry rate by a factor of two (i.e., 1,000 kg m$^{-3}$ of water is equivalent to 1,000 kg m$^{-3}$ of ice avalanche volume, and only 500 kg m$^{-3}$ of landslide material), which is the usual approach used in the simulation process (Byers et al., 2018, 2020).

**Reply:** This sentence was modified as "The BASEMENT model only accepts water as an inflow. Therefore, the ice avalanches and landslides were converted to water with a density scale of 1.0 and 2.0 (i.e., 1,000 kg m$^{-3}$ of water is equivalent to 1,000 kg m$^{-3}$ of ice, and only 500 kg m$^{-3}$ of moraine) to accurately describe the momentum transfer of ice avalanches and landslides entering the lake.".

**Line 312:** It was shown that the 2D SWE used by the BASEMENT model inherently leads to excessive wave attenuation.

**Reply:** This sentence was modified as "It was shown that the 2D SWEs in the BASEMENT model leads to excessive wave attenuation inherently.".

**Line 313:** The Heller-Hager model (Heller et al., 2009) is a combination of analytical and empirical equations used to simulate impulse wave generation, propagation, and run-up from the movement of material entering a lake.

**Reply:** This sentence was modified as "The Heller-Hager model combining analytical and empirical equations is often used to simulate impulse wave generation, propagation, and run-up resulting from mass movements entering a lake (Lala et al., 2018).".

**Line 315:** Although the approach relies on simplifying measurements about lake geometry, it has been used to successfully simulate multiple real events and performs well in characterizing impulse waves within lakes, making it a simple, but useful, calibration measure for more complex hydrodynamic models (Somos-Valenzuela et al., 2016).

**Reply:** This sentence was modified as "Although the model relies on simplified measurements of lakes' geometry, it has been used to simulate multiple actual events successfully. It characterizes impulse waves in lakes well, making it a simple but helpful calibration measure for more complex hydrodynamic models (Somos-Valenzuela et al., 2016).".

**Line 318:** BASEMENT simulated waves are usually considered more accurate when they are of the same order of magnitude as Heller-Hager waves; however, when they are not, the mass entry rate is varied by adjusting the inflow hydrograph and boundary width to match the amplitude of the Heller-Hager empirical model near the dam of the initial wave trajectory (Byers et al., 2018; Lala et al., 2018).

**Reply:** This sentence was modified as "Waves simulated by the BASEMENT model were usually considered

more accurate when they were in the same order of magnitude as those simulated by the Heller-Hager model. However, when they were not, the mass entry rate was changed by adjusting the inflow hydrograph and the boundary width to match the amplitude of the initial wave trajectory near the dam for the Heller-Hager model (Byers et al., 2018; Lala et al., 2018).".

**Line 330:** Abnormally high lake outflow has the potential to destroy the surface protection layer of the outlet streambed and triggers vertical dam erosion. After the initial cut, more lake water will flow out, followed by an increase in sediment transport rate and a gradual widening of the rift.

**Reply:** This sentence was replaced with "Overtopping flows are the most common trigger for moraine-dam breaching (Richardson and Reynolds, 2000). They initiate dam erosion, leading to greater outflow and increasing hydrodynamic force that progressively enlarge the breach (Singh, 1996).".

**Line 334:** In this study, hydro-morphodynamic simulations of potential erosion-driven breach failures of Bienong Co were carried out by the BASEMENT model.

**Reply:** This sentence was replaced with "In this study, the erosion-driven dam failure of Bienong Co was simulated by the BASEMENT model.".

**Line 335:** The model uses the Meyer-Peter and Müller (MPM) equation to characterize sediment transport and estimates suspended and nudged mass fluxes by calculating the shear stress in the flow through the modified Shields parameter (Vetsch et al., 2022).

**Reply:** This sentence was modified as "The model characterizes sediment transports using the Meyer-Peter and Müller (MPM) equation and estimates suspended and nudged mass fluxes by calculating the shear stress in flows through the modified Shields parameter (Vetsch et al., 2022).".

**Line 338:** The overtopping flow leading to erosion of the moraine dam is generated by the wave amplitude of the BASEMENT model calibrated by the Heller-Hager model. In the previous step, we adjusted the wave amplitude near the moraine dam in the BASEMENT model to be close to (difference within 1 m) that calculated by the Heller-Hager model by modifying the width of the upstream boundary.

**Reply:** This sentence was deleted because it was mentioned earlier.

**Line 341:** ALOS PALSAR DEM is the base data for the mesh generation of the moraine dam with the maximum TIN area of 200 m$^2$.

**Reply:** This sentence was modified as "The mesh of the moraine dam was generated from ALOS PALSAR DEM, with a maximum TIN area of 200 m$^2$.".

**Line 342:** We set a cross-section along the crest of the moraine dam (Fig. 2g), where moraine dam deformation, i.e., erosion, overtopping, as well as outflow discharges, were analyzed.

**Reply:** This sentence was modified as "We set a cross-section along the crest of the moraine dam (Fig. 2g), at which the erosion of the moraine dam and outflow discharges were analyzed.".

**Line 346:** The MPM-multi model simulates hiding and armoring processes that may lead to unrealistically low levels of erosion (Vetsch et al., 2022).

**Reply:** This sentence was modified as "The MPM-multi algorithm simulates hiding and armoring processes, potentially leading to unrealistically low-level erosion (Vetsch et al., 2022).".

**Line 348:** The MPM model…

**Reply:** This sentence was modified as "The MPM algorithm…".

**Line 348:** In this study, we applied the MPM-multi model to simulate bed-load transport of the moraine dam, which is composed of materials with different grain sizes.

**Reply:** This sentence was modified as "In this study, we used the MPM-multi algorithm to simulate the bed-load transport of the dam composed of moraines of different grain sizes.".

**Line 352:** The specific grain size distribution was not measured, but was instead taken from an inventory of glacial lakes in the Indian Himalayas (Worni et al., 2013) that had performed well in recreating previous GLOFs in Nepal (Byers et al., 2020).

**Reply:** This sentence was modified as "The specific grain size distribution of Bienong Co's moraine dam was not measured. It was instead taken from an inventory of glacial lakes in the Indian Himalayas (Worni et al., 2013) that had performed well in reconstructing previous GLOFs in Nepal (Byers et al., 2020).".

**Line 354:** Despite uncertainty in the actual grain size distribution, a similar GLOF modeling study in the Barun Valley (Byers et al., 2018) found little difference in simulated moraine erosion between the grain size distributions listed in Worni et al. (2013).

**Reply:** This sentence was not significant enough, so it was deleted.

**Line 356:** The moraine dam of Bienong Co consists of a large grain cover with a thickness of approximately 0.5 m at the top and fine grain underneath, which is clearly visible on the side walls of the channel scoured by water (Fig. 2e).

**Reply:** This sentence was modified as "The moraine dam of Bienong Co consists of large grains with a thickness of approximately 0.5 m from the top and small grains underneath, which is visible on the side wall of the channel scoured by outflows (Fig. 2e).".

**Line 359:** The largest particle size ($d_{50}$ = 180 mm) in the upper layer with a depth of 0.5 m, and the grain size distribution of the lower layer with a depth of 71.5 m (considering the mean height of moraine dam of 72 m),

**Reply:** This sentence was modified as "The upper layer with the large grain ($d_{50}$ = 180 mm) has a depth of 0.5 m; the lower layer with the grain size distribution same as Worni et al. (2013) has a depth of 71.5 m (considering the mean height of moraine dam of 72 m).".

**Line 361:** Finally, a correction factor of 2.0 was used in the model to increase the rate of bed load transport. Values between 0.5 (low transport) and 1.7 (high transport) are generally realistic (Wong and Parker, 2006); whereas a value of 2.0 provides high sediment transport conditions (Somos-Valenzuela et al., 2016) to compensate for the lower erosion of the MPM-multi model.

**Reply:** This sentence was modified as "In addition, a correction factor of 2.0 was used to increase the bed load transport rate. Values between 0.5 (low transport) and 1.7 (high transport) are generally realistic (Wong and Parker, 2006); whereas a value of 2.0 provides a high sediment transport condition (Somos-Valenzuela et al., 2016) to compensate for the lower erosion of the MPM-multi algorithm.". This sentence was moved to the end of the sentence "In this study, we used the MPM-multi algorithm to simulate the bed-load transport of the dam composed of moraines of different grain sizes.".

**Line 367:** In this study, the BASEMENT model was used to simulate the hydrodynamic behavior of potential GLOFs along the flow channel, and the hazard of floods was assessed by analyzing the inundation area, flow velocity, and flood arrival time at these settlements.

**Reply:** This sentence was modified as "This study used the BASEMENT model to simulate the hydrodynamic behavior of potential GLOFs along the flow channel. We assessed the hazard of GLOFs by analyzing the inundation extent and depth, flow velocity, and flood arrival time at these settlements.".

**Line 369:** The 2D model for an unsteady hydraulic simulation requires input of terrain data and boundary conditions.

**Reply:** This sentence was modified as "The BASEMENT 2D module for an unsteady hydraulic simulation requires inputs of terrain data and boundary conditions.".

**Line 370:** The terrain data were represented by a 2D mesh covering the entire flow area, which was obtained from ALOS PALSAR DEM.

**Reply:** This sentence was modified as "The terrain was represented by a 2D mesh covering the entire flow channel, obtained from ALOS PALSAR DEM.".

**Line 372:** The mesh was also generated by the BASEmesh plugin of QGIS software, and the individual cell area for the main flow channel and other regions were set to 500 m$^2$ and 5,000 m$^2$, respectively, considering the accuracy requirements of the simulation and the computational efficiency of the model.

**Reply:** This sentence was modified as "The mesh was generated by the BASEmesh plugin of QGIS software, and the maximum TIN areas of the main flow channel and other regions were set as 500 m$^2$ and 5,000 m$^2$, respectively, for the accuracy requirement of the simulation and the computation efficiency of the model.".

**Line 374:** Friction of the river channel to a given flow was determined by the Manning's roughness coefficient (Coon, 1998), which is dependent on the land use and land cover of the modeling river channel in the study area.

**Reply:** This sentence was modified as "The friction of a flow channel was determined by Manning's roughness coefficient (Coon, 1998), which is dependent on land use and cover.".

**Line 378:** The upstream boundary is the outflow hydrograph from the moraine dam simulated by the BASEMENT model.

**Reply:** This sentence was modified as "The inflow boundary is the outflow hydrograph from the lake.".

**Line 379:** The downstream boundary is the water level-discharge relationship of the cross-section in the downstream boundary of the simulation area, and was estimated by the critical depth method (Byers et al., 2018).

**Reply:** This sentence was modified as "The outflow boundary is the water level-discharge relationship of a cross-section downstream, estimated by the critical depth method (Byers et al., 2018).".

**Result**

**Line 383:** The basin morphology of Bienong Co was modeled based on the TIN grid created by the field depth

data (Fig. 3).

**Reply:** This sentence was modified as "The basin morphology of Bienong Co was determined based on the bathymetry data (Fig. 3).".

**Line 384:** Apparently, this lake has a relatively flat basin bottom, and both flanks are deep (Fig. 4).

**Reply:** This sentence was modified as "As shown in Fig. 4, the lake has a relatively flat bottom and steep flanks.".

**Line 385:** Similar to most glacial lakes (Yao et al., 2012; Zhou et al., 2020), the slope of the lake shores near the glacier is steeper than that near the moraine dam.

**Reply:** This sentence was modified as "The slope of the lake shore near the glacier is steeper than that near the moraine dam, which is similar to most moraine-dammed glacial lakes (Yao et al., 2012; Zhou et al., 2020).".

**Line 392:** The volume of Bienong Co, calculated using the surface elevation and the lake bed derived from the TIN grid, was approximately $102.3 \times 10^6$ m$^3$ in 2020, which is a generally accurate estimate of the magnitude of this moraine-dammed lake.

**Reply:** This sentence was simplified as "The volume of Bienong Co was approximately $102.3 \times 10^6$ m$^3$ in 2020.". The methods of lake volume calculation are described in detail in the section 3.1 Bathymetry and modeling. Therefore, we deleted it in this sentence.

**Line 397:** 4.2 Potential GLOFs modeling

**Reply:** This "GLOFs" was modified as "GLOF".

**Line 401:** Most of the materials enter the lake within approximately 120 s. Based on the area of ~1.15 km$^2$ in 2021, the above three scenarios could result in a rise of approximately 3.3 m, 4.2 m, and 5.1 m in the lake surface, respectively.

**Reply:** In this sentence, "in the lake surface" was modified as "in the lake level".

**Line 403:** Material volumes entering the lake by both landslides are much smaller than those of the ice avalanches (Fig. 5b and c). Scenarios B1, B2, and B3 and C1, C2, and C3 dump $0.03 \times 10^6$ m$^3$, $0.09 \times 10^6$ m$^3$, and $0.17 \times 10^6$ m$^3$, and $0.06 \times 10^6$ m$^3$, $0.15 \times 10^6$ m$^3$, and $0.30 \times 10^6$ m$^3$ of mass volume into the lake, respectively.

**Reply:** This sentence was modified as "The volumes of landslides entering the lake are much smaller than those of ice avalanches (Fig. 5b and c). In Scenarios B1, B2, and B3, the volumes of moraines entering the lake are $0.03 \times 10^6$ m$^3$, $0.09 \times 10^6$ m$^3$, and $0.17 \times 10^6$ m$^3$, respectively. In Scenarios C1, C2, and C3, the volumes of moraines entering the lake are $0.06 \times 10^6$ m$^3$, $0.15 \times 10^6$ m$^3$, and $0.30 \times 10^6$ m$^3$, respectively.".

**Line 406:** The time for materials entering the lake is less than in Scenario A, with Scenario B being completed in approximately 10 s and Scenario C in approximately 15 s.

**Reply:** This sentence was modified as "It takes less time for the moraines to enter the lake, with Scenarios B1-3 being completed in approximately 10 s and Scenarios C1-3 in approximately 15 s.".

**Line 411:** The impact area caused by material entering the lake differs for different scenarios.

**Reply:** This sentence was modified as "The impact zones caused by mass entering the lake differ for different scenarios.".

**Line 411:** The impact zones caused by Scenarios A1, A2 and A3 are 0.27 km², 0.31 km², and 0.38 km², with horizontal distances of 549 m, 629 m, and 752 m from the upper boundary, respectively (Fig. 6).

**Reply:** This sentence was modified as "They are 0.27 km², 0.31 km², and 0.38 km² for Scenarios A1, A2, and A3, with horizontal distances of 549 m, 629 m, and 752 m from the inflow boundary, respectively (Fig. 6).".

**Line 413:** In contrast, each of these three scenarios for Scenarios B and C result in a relatively small impact zone, with Scenario C3 being the largest at 0.14 km² and Scenario B1 being the smallest at 0.04 km².

**Reply:** This sentence was modified as "In contrast, each scenario of the Scenarios B1-3 and C1-3 results in a relatively small impact zone, with Scenario C3 being the largest of 0.14 km² and Scenario B1 being the smallest of 0.04 km².".

**Line 415:** Scenarios A1, A2, and A3 produce maximum flow heights of 39.5 m, 46.2 m, and 53.5 m, and average flow heights of 12.2 m, 14.6 m, and 12.3 m in the impact area, respectively.

**Reply:** In this sentence, we add "the" before "maximum flow heights" and "average flow heights", and "impact area" was modified as "impact zones". This sentence was modified as "Scenarios A1, A2, and A3 produce the maximum flow heights of 39.5 m, 46.2 m, and 53.5 m, and the average flow heights of 12.2 m, 14.6 m, and 12.3 m in the impact zones, respectively.".

**Line 416:** The maximum flow height range for Scenarios B1, B2, and B3 is 6.8 - 14.6 m, and the average flow height range is 1.8 - 3.5 m.

**Reply:** This sentence was modified as "The maximum and the average flow heights of Scenarios B1-3 range from 6.8 m to 14.6 m and from 1.8 m to 3.5 m, respectively.".

**Line 418:** The maximum and average flow height ranges for Scenario C1, C2, and C3 are 5.7 - 29.2 m and 2.0 - 4.7 m, respectively (Fig. 6).

**Reply:** This sentence was modified as "The ranges of the maximum and the average flow heights of Scenarios C1-3 are 5.7 - 29.2 m and 2.0 - 4.7 m, respectively (Fig. 6).".

**Line 419:** The maximum flow velocities for Scenarios A1, A2, and A3 ….

**Reply:** In this sentence, "for" was modified as "of".

**Line 421:** The maximum and average flow velocities for Scenarios B1, B2,

**Reply:** In this sentence, "for" was modified as "of".

**Line 426:** By counting material volumes of ice avalanches and landslides entering Bienong Co at different time periods based on the RAMMS model, we derived the time series of the material-entering rate, as shown in Fig. 7.

**Reply:** This sentence was modified as "By counting the volumes of ice avalanches and landslides entering Bienong Co in different timesteps obtained from the RAMMS model, we derived the time series of the mass-entering rate, as shown in Fig. 7.".

**Line 427:** Compared to Scenarios A2 and A3, Scenario A1 has the highest peak flow rate of 439,952.57 m³/s, but it decreases rapidly after reaching the peak within 2 s of the ice avalanche, i.e., the ice avalanche occurs in a moment.

**Reply:** This sentence was modified as "In Scenarios A1-3, Scenario A1 has the highest peak flow rate of

439,952.57 m$^3$/s; however, it decreases rapidly after reaching the peak within 2 s of the ice avalanche, i.e., the ice avalanche occurs in a brief moment.".

**Line 429:** Scenarios A2 and A3 exhibit obvious fluctuations in the process of ice avalanches into the lake, with sub-peaks in both scenarios that are comparable to the first peak, after which the flow rates still possess strong fluctuations.

**Reply:** This sentence was modified as "Scenarios A2 and A3 exhibit apparent fluctuations in the process of ice avalanches entering the lake, with the sub-peaks comparable to the peaks.".

**Line434:** This is because the ice avalanches of Scenarios A2 and A3 are further away from the lake than Scenario A1, and total volumes of ice avalanches are larger than Scenario A1, and thus they undergo a more complex process when they enter the lake.

**Reply:** This sentence was modified as "Scenarios A2 and A3 have the larger volumes of ice avalanches and the further transfer distances than Scenario A1, leading to the more complex processes of ice bodies entering the lake.".

**Line 437:** The peak flows increase sequentially from Scenarios B1 and C1, B2 and C2, to B3 and C3, with peak values of 50,849 m$^3$/s and 92,529 m$^3$/s for Scenarios B3 and C3, respectively. The peak flow for Scenario B3 is approximately 3.8 times that of Scenario B1, and that of Scenario C3 is 6.5 times that of Scenario C1, respectively.

**Reply:** This sentence was modified as "The peak flows increase sequentially from Scenarios B1 and C1 to B2 and C2, and then to B3 and C3. The values of Scenarios B3 and C3 are 50,849 m$^3$/s and 92,529 m$^3$/s, respectively, which are approximately 3.8 times and 6.5 times those of Scenarios B1 and C1.".

**Line 439:** The peak flows for the six scenarios of Scenarios B and C occur in the range of 6 - 8 s seconds (Fig. 7).

**Reply:** This sentence was modified as "The peak flows of Scenarios B1-3 and C1-3 occur in 6 - 8 s (Fig. 7).".

**Line 444:** The time-volume relationships of materials entering a lake have important effects on the generation and propagation of displacement waves in the lake.

**Reply:** This sentence was modified as "The time-volume relationship of mass entering a lake has an important effect on the generation and propagation of displacement waves in the lake.".

**Line 445:** Ice avalanche scenarios (A1, A2, and A3) have a much greater impact on Bienong Co than the two landslide scenarios (B1, B2, B3, C1, C2, and C3) because the assumed release volume of ice avalanches is much greater than that of landslides.

**Reply:** In this sentence, "because the assumed release volume of ice avalanches is much greater than that of landslides" was modified as "because the release volumes of ice avalanches are much greater than those of landslides".

**Line 447:** The wave height near the moraine dam is the result of the BASEMENT model calibrated by the Heller-Hager model. By adjusting the inflow boundary's width, we made the BASEMENT model produce wave amplitudes near the dam with a difference smaller than 1 m of those calculated by the Heller-Hager model, which is important for subsequent simulations, although the maximum wave amplitude in the lake is

exaggerated.

**Reply:** This sentence was deleted because the content has been mentioned in the Methodology section.

**Line 450:** In Scenario A, displacement waves propagate straight from the glacier to the moraine dam, and arrive at the vicinity of the moraine dam at approximately 70 s.

**Reply:** In this sentence, "and arrive at" was modified as "arriving to".

**Line 452:** Scenarios A1, A2, and A3 produce the highest wave amplitudes in the lake of 35.2 m, 39.0 m, and 66.4 m, respectively, and wave amplitudes near the moraine dam of 17.1 m (72 s), 20.2 m (74 s), and 25.2 m (72 s), respectively (Fig. 8).

**Reply:** In this sentence, "and wave amplitudes near the moraine dam of 17.1 m (72 s), 20.2 m (74 s), and 25.2 m (72 s), respectively (Fig. 8)." was modified as "and the wave amplitudes near the moraine dam are 17.1 m, 20.2 m, and 25.2 m, respectively (Fig. 8).".

**Line 454:** In Scenario B, a landslide occurs at the left shore of Bienong Co near the moraine dam (Fig. 2c).

**Reply:** This sentence was modified as "In Scenarios B1-3, the landslides occur at the left shore of Bienong Co near the moraine dam (Fig. 2c).".

**Line 459:** The landslide in Scenario C occurs on the right bank of Bienong Co near the glacier, in a same manner as that in Scenario B, in which waves propagate to the opposite bank first after materials enter the lake, with the maximum wave amplitudes of 4.8 m, 9.6 m, and 24.7 m for Scenarios C1, C2, and C3.

**Reply:** This sentence was modified as "The landslides in Scenarios C1-3 occur at the right shore of Bienong Co near the mother glacier. The displacement waves propagate in the same way as in Scenarios B1-3, in which waves propagate to the opposite shore first after moraines enter the lake, with the maximum wave amplitudes of 4.8 m, 9.6 m, and 24.7 m for Scenarios C1, C2, and C3, respectively.".

**Line 462:** Unlike Scenario B, displacement waves in Scenario C cross the entire lake, reaching the moraine dam with wave amplitudes of 0.6 m, 2.2 m, and 4.9 m near the moraine dam, respectively (Fig. 8).

**Reply:** This sentence was modified as "Unlike Scenarios B1-3, the displacement waves in Scenarios C1-3 cross the entire lake, with wave amplitudes of 0.6 m, 2.2 m, and 4.9 m near the moraine dam, respectively (Fig. 8).".

**Line 463:** Therefore, although the landslide volume of Scenario C is larger than that of Scenario B, wave amplitudes near the moraine dam are smaller than those of Scenario B.

**Reply:** This sentence was modified as "Therefore, although the volumes of landslides in Scenarios C1-3 are larger than those in Scenarios B1-3, the wave amplitudes near the moraine dam are smaller than those in Scenarios B1-3.".

**Line 469:** Since the freeboard of the moraine dam is 0 m, the occurrence of overtopping flow is inevitable in all scenarios, but there are differences in magnitude.

**Reply:** This sentence was modified as "Since the freeboard of the moraine dam is 0 m, the occurrence of overtopping is inevitable in all scenarios, but there are differences in the magnitude of overtopping.".

**Line 470:** In Scenarios A1, A2, and A3, peak discharges at breaches of the moraine dam are 4,996 m³/s, 7,817 m³/s, and 13,078 m³/s, corresponding to a total released volume of $24.1×10^6$ m³, $25.3×10^6$ m³, and $26.4×10^6$

m$^3$, respectively (Fig. 9).

**Reply:** This sentence was modified as "In Scenarios A1-3, the moraine dam is eroded by enormous discharges, forming a breach. The peak discharges at the breach are 4,996 m$^3$/s, 7,817 m$^3$/s, and 13,078 m$^3$/s, corresponding to the total released volumes of 24.1×10$^6$ m$^3$, 25.3×10$^6$ m$^3$, and 26.4×10$^6$ m$^3$, respectively (Fig. 9).".

**Line 475:** The moraine dam is eroded by enormous discharge output in Scenarios A1, A2, and A3, forming breaches.

**Reply:** We repositioned this sentence and placed it in the second sentence.

**Line 475:** Due to the similar volume of released water, the depth of the breach is slightly different for Scenarios A1, A2, and A3, which are 19.0 m, 19.1 m, and 19.3 m, respectively (Fig. 10).

**Reply:** This sentence was modified as "Due to the similar volume of water released in Scenarios A1-3, the depth of the breach is similar at approximately 19.0 m (Fig. 10).".

**Line 477:** Moreover, the peak discharge is quite different for the three scenarios, resulting in different breach widths of 295.0 m, 339.4 m, and 368.5 m, respectively.

**Reply:** This sentence was modified as "However, the peak discharges differ among the three scenarios, resulting in different breach widths of 295.0 m, 339.4 m, and 368.5 m, respectively.".

**Line 479:** Scenarios B1, B2, and B3 and C1, C2 and C3 resulted in overtopping flows of 0.6 × 10$^6$ m$^3$, 1 × 10$^6$ m$^3$, and 2.6 × 10$^6$ m$^3$, and 0.1 × 10$^6$ m$^3$, 0.9 × 10$^6$ m$^3$, and 3.4 × 10$^6$ m$^3$, respectively, in which only Scenarios B3 and C3 cause erosion of the moraine dam and form a breach.

**Reply:** This sentence was modified as "Scenarios B1, B2, and B3 and C1, C2, and C3 result in overtopping flows of 0.6 × 10$^6$ m$^3$, 1 × 10$^6$ m$^3$, and 2.6 × 10$^6$ m$^3$, and 0.1 × 10$^6$ m$^3$, 0.9 × 10$^6$ m$^3$, and 3.4 × 10$^6$ m$^3$, respectively, with only Scenarios B3 and C3 causing erosion of the moraine dam and forming a breach.".

**Line 481:** Discharges at the breach become stable beginning at 18,000 s following landslide material entry into the lake in Scenarios B3 and C3, with breach depths of 6.5 m and 7.9 m, and breach widths are 153 m and 169 m with peak discharges of 504 m$^3$/s and 733 m$^3$/s, respectively.

**Reply:** This sentence was modified as "In Scenarios B3 and C3, the discharges at the breach become stable after 18,000 s of moraines enter the lake. The peak discharges are 504 m$^3$/s and 733 m$^3$/s, with breach depths of 6.5 m and 7.9 m and breach widths of 153 m and 169 m, respectively.".

**Line 489:** 4.2.4 GLOFs impact in the downstream region

**Reply:** In this sentence, "GLOFs" was modified as "GLOF".

**Line 490:** The hydraulic behavior of GLOFs in the flow channel immediately downstream of moraine dam to the convergence with Song Qu with the distance of ~53 km was simulated using the BASEMENT model.

**Reply:** This sentence was modified as "The hydraulic behavior of GLOFs in the flow channel from the moraine dam to the convergence with Song Qu (~53 km) was simulated using the BASEMENT model.".

**Line 492:** Due to the lack of reliable data on small baseflows in the flow channel, they were neglected in the simulation.

**Reply:** This sentence was modified as "The small baseflows in the flow channel were neglected due to the

lack of reliable data.".

**Line 493:** Considering the propagation of floods in different scenarios, we assessed the propagation of GLOF in the downstream channel within 20 h from ice avalanche and landslide materials entry into the lake.

**Reply:** This sentence was modified as "Considering the differences between scenarios, we evaluated the propagation of GLOFs in the downstream channel within 20 h.".

**Line 495:** Peak discharges at the breach outlet of Scenarios A1, A2, and A3 all occur approximately 600 s after the ice avalanche material enters the lake, and they are 4,996 $m^3/s$, 7,817 $m^3/s$, and 13,078 $m^3/s$, respectively, based on which floods all pass through 18 settlements in the downstream river in 20 h with the areas of 7.6 $km^2$, 8.0 $km^2$, and 8.5 $km^2$, as well as average water depths of 8.4 m, 9.1 m, and 10.0 m, respectively (Fig. 11).

**Reply:** This sentence was modified as "The peak discharges at the breach of Scenarios A1, A2, and A3 are 4,996 $m^3/s$, 7,817 $m^3/s$, and 13,078 $m^3/s$, respectively, all occurring approximately 600 s after ice bodies enter the lake (abbreviated as after 600 s). GLOFs of the three scenarios all pass through 18 settlements in the downstream flow channel in 20 h. The inundation areas of Scenarios A1, A2, and A3 are 7.6 $km^2$, 8.0 $km^2$, and 8.5 $km^2$, respectively, with the average water depths of 8.4 m, 9.1 m, and 10.0 m, respectively (Fig. 11).".

**Line 500:** Scenarios B1 and C1 only have a small volume of overtopping flow from the lake (peak discharges of 106 $m^3/s$ (after 40 s) and 12 $m^3/s$ (after 50 s)), and fail to generate runoff downstream of the dam.

**Reply:** This sentence was modified as "Scenarios B1 and C1 result in small overtopping flows, with peak discharges of 106 $m^3/s$ (after 40 s) and 12 $m^3/s$ (after 50 s), respectively.".

**Line 502:** Scenarios B2 and C2 produce very limited overtopping flow with peak discharges of 177 $m^3/s$ (after 240 s) and 186 $m^3/s$ (after 480 s), respectively, and outflows remain only within approximately 1 km downstream of the dam. Peak discharges at the breach outlet of Scenarios B3 and C3 are 504 $m^3/s$ (after 240 s) and 733 $m^3/s$ (after 480 s), yielding inundation areas of 1.7 $km^2$ and 2.2 $km^2$ with average water depths of 1.9 m and 2.4 m in the downstream region, respectively.

**Reply:** This sentence was modified as "Scenarios B2 and C2 produce limited overtopping flows with peak discharges of 177 $m^3/s$ (after 240 s) and 186 $m^3/s$ (after 480 s), respectively. Outflows remain only within approximately 1 km downstream of the moraine dam. The peak discharges at the breach of Scenarios B3 and C3 are 504 $m^3/s$ (after 240 s) and 733 $m^3/s$ (after 480 s), yielding the inundation areas of 1.7 $km^2$ and 2.2 $km^2$ with the average water depths of 1.9 m and 2.4 m in the downstream region, respectively.".

**Line 507:** Among the nine scenarios that we considered, only Scenarios A1, A2, A3, B3, and C3 caused GLOFs propagation in the downstream region with a far distance, in which Scenario A3 had the largest flood magnitude, and Scenario B3 had the smallest flood magnitude.

**Reply:** In this sentence, "had" was modified as "has". This sentence was modified as "In the nine scenarios, only Scenarios A1-3, B3, and C3 result in GLOFs propagating in the downstream flow channel over a considerable distance; Scenario A3 has the largest flood magnitude, whereas Scenario B3 has the smallest flood magnitude.".

**Line 514:** GLOFs of different magnitudes pose different potential hazards to each settlement along the flow

channel.

**Reply:** This sentence was modified as "GLOFs of different magnitudes pose different hazards to settlements along the flow channel.".

**Line 514:** Scenario A3 produces the most severe threat of GLOF to the downstream region.

**Reply:** In this sentence, "of GLOF" was deleted.

**Line 515:** Six settlements, including Ang'na, Wa'na, Haqing, Kemaluo, Xinka, and Dading, will be completely submerged by flooding.

**Reply:** In this sentence, "Ang'na" was modified as "Ang'na". This sentence was modified as "Six settlements are entirely submerged by the flood, including Ang'na, Wa'na, Haqing, Kemaluo, Xinka, and Dading villages.".

**Line 516:** Kemaluo village, located 37.9 km downstream of Bienong Co, will experience the relatively largest maximum flow depth of 19.8 m, and the village of Ang'na, having a distance of 6.0 km from Bienong Co, will experience the relatively smallest maximum flow depth of 6.1 m.

**Reply:** This sentence was modified as "Kemaluo village, located 37.9 km downstream of Bienong Co, experiences the largest maximum flow depth of 19.8 m, and the Ang'na village, located 6.0 km from Bienong Co, experiences the smallest maximum flow depth of 6.1 m.".

**Line 519:** Wa'na village is the most affected of all of the villages by GLOF due to the most flooded houses.

**Reply:** This sentence was modified as "Wa'na is the village most affected by the GLOF due to the largest flooded area.".

**Line 521:** The maximum flow depth of 11.0 m in Bada village is the largest, and that of 7.2 m in both Dama and Zhibu villages is the smallest.

**Reply:** This sentence was modified as "The maximum flow depth in Bada village is the largest (11.0 m); that in Dama village and Zhibu village is the smallest (7.2 m).".

**Line 523:** Four settlements, Yilongduo, Jiawu, Zongri, and Dazi are spared from floods, in which Yilongduo may be slightly affected, while Dazi is the safest village due to its far distance from the river.

**Reply:** This sentence was modified as "Four settlements, Yilongduo, Jiawu, Zongri, and Dazi villages, are spared from flooding. Among them, Yilongduo village may be slightly affected; Dazi is the safest village due to its far distance from the river.".

**Line 524:** Flooding in Scenario A2 has a relatively small impact on downstream villages compared to that of Scenario A3, showing the reduced extent of inundation and flow depth.

**Reply:** This sentence was modified as "The GLOF of Scenario A2 has a minor impact on downstream villages than that of Scenario A3, with the reduced inundation extent and flow depth.".

**Line 526:** The flooding extents in Ang'na and Haqing villages have been reduced. However, villages Wa'na, Kemaluo, Xinka, and Dading will still be completely flooded, but the maximum flow depth is reduced from 16.5 m, 19.8 m, 12.5 m, and 17.5 m, to 13.6 m, 19.3 m, 12.0 m, and 16.6 m, respectively.

**Reply:** This sentence was modified as "The flooding extent in Ang'na and Haqing villages has a reduction. The Wa'na, Kemaluo, Xinka, and Dading villages are still entirely flooded, however, the maximum flow

depths are decreased from 16.5 m, 19.8 m, 12.5 m, and 17.5 m to 13.6 m, 19.3 m, 12.0 m, and 16.6 m, respectively.".

**Line 529:** For the nine villages that are partially affected by Scenario A2, they are still affected by flooding of Scenario A2 except for Dama village, but the impact of flooding is diminished.

**Reply:** This sentence was modified as "The nine villages partially affected in Scenario A3 are still affected by the GLOF of Scenario A2 except for Dama village, but the impact magnitude is diminished.".

**Line 531:** Scenario A1 differs from Scenario A3 in that Dama and Xingkamaluo villages were spared from flooding, while other villages experienced significant reductions in flood extent and inundation depth.

**Reply:** This sentence was modified as "Scenario A1 differs from Scenario A3 in that the Dama and Xingkamaluo villages are spared from the GLOF, and other villages experience greatly reduced flooding extents and inundation depths.".

**Line 533:** The GLOFs of Scenarios B3 and C3 have significantly minor impact in the downstream than Scenarios A1, A2, and A3.

**Reply:** This sentence was modified as "The GLOFs of Scenarios B3 and C3 have a significantly minor impact on the downstream flow channel than Scenarios A1-3.".

**Line 534:** Only Wa'na, Qiangjiuci, and Bula villages will be partially affected, with the maximum flow depth of 3.1 m, 1.9 m, and 2.0 m in Scenario B3 and 4.0 m, 2.4 m, and 2.7 m in Scenario C3, respectively (Fig. 12).

**Reply:** In this sentence, "Wa'na" was modified as "Wa'na", "will be" was modified as "are". The sentence was changed to "Only Wa'na, Qiangjiuci, and Bula villages are partially affected, with the maximum flow depths of 3.1 m, 1.9 m, and 2.0 m in Scenario B3 and 4.0 m, 2.4 m, and 2.7 m in Scenario C3, respectively (Fig. 12).".

**Line 536:** The total areas of houses, courtyards, and farmlands (around settlement) affected by Scenarios A1, A2, A3, B3, and C3 were estimated to be 23,984 m$^2$, 32,076 m$^2$, 41,038 m$^2$, 3,820 m$^2$, and 3,918 m$^2$, respectively.

**Reply:** This sentence was modified as "The total areas of houses, courtyards, and farmlands (around settlement) affected by Scenarios A1, A2, A3, B3, and C3 are 23,984 m$^2$, 32,076 m$^2$, 41,038 m$^2$, 3,820 m$^2$, and 3,918 m$^2$, respectively.".

**Line 537:** In Scenarios A1, A2, A3, all 13 bridges and road with a length of approximately 35 km are within the flood zones, and in Scenarios B3 and C3, there are four bridges as well as road with a length of approximately 3.6 km and 6.7 km within the flood zones.

**Reply:** This sentence was modified as "In Scenarios A1-3, all 13 bridges and approximately 35 km of roads (including the Jiazhong Highway) are in the flooding zone. In Scenarios B3 and C3, four bridges and approximately 3.6 km and 6.7 km of roads are in the flooding zone.".

**Line 540:** Here we only assess the potential impact of floods on these man-made structures, but the concrete magnitude of the impact is beyond the scope of this study.

**Reply:** This sentence was modified as "Here we only assess the potential impact of GLOFs on the artificial structures, but the concrete impact magnitude is beyond the scope of this study.".

**Line 543:** The peak discharge and velocity of GLOFs in these villages experience a gradually decreasing process, while the arrival time of peak discharges is prolonged, favoring the evacuation of residents in the downstream area.

**Reply:** This sentence was modified as "Peak discharges and velocities of GLOFs gradually decrease from upstream to downstream. Meanwhile, the arrival times of peak discharges are prolonged, favoring the evacuation of residents in the downstream region.".

**Line 545:** Peak discharges in S1, Yilongduo, Jiawu, and Ang'na villages are similar for each scenario, ~4,000 $m^3$/s of Scenario A1, ~6,000 $m^3$/s of Scenario A2, and ~10,000 $m^3$/s of Scenario A3. Wa'na, Qiangjiuci, and Bula villages have similar peak discharges, which are ~3,800 $m^3$/s, ~5,000 $m^3$/s, and ~8,000 $m^3$/s for Scenarios A1, A2, and A3, respectively.

**Reply:** This sentence was modified as "The peak discharges at S1, Yilongduo, Jiawu, and Ang'na villages are similar, they are ~4,000 $m^3$/s in Scenario A1, ~6,000 $m^3$/s in Scenario A2, and ~10,000 $m^3$/s in Scenario A3.".

**Line 548:** Beginning with Bula village, peak discharges of each scenario decrease significantly towards the downstream.

**Reply:** This sentence was modified as "From Bula village, the peak discharges decrease significantly towards the downstream.".

**Line 549:** Taking Scenario A3 as an example, at Bula village, peak discharge is 7,512 $m^3$/s, at Haqing village, it becomes smaller than 6,000 $m^3$/s, at Dama village, it drops below 4,000 $m^3$/s, at Qiwuxing village it drops below 3,000 $m^3$/s, and at Xinka village it decreases to below 1,000 $m^3$/s (Fig. 13).

**Reply:** This sentence was modified as "Taking Scenario A3 as an example, the peak discharge is 7,512 $m^3$/s at Bula village; then it is gradually less than 6,000 $m^3$/s, 4,000 $m^3$/s, 3,000 $m^3$/s, and 1,000 $m^3$/s from Haqing village, Dama village, Qiwuxing village to Xinka village (Fig. 13).".

**Line 553:** The flood flow velocity varies dramatically, with Scenarios A1, A2, and A3 corresponding to maximum velocities of 8.9 m/s, 12.2 m/s, and 14.9 m/s, respectively, at village S1. At Dading village, the maximum flow velocity of GLOFs is approximately 2m/s.

**Reply:** This sentence was modified as "The velocities of GLOFs vary dramatically from upstream to downstream. At village S1, the maximum velocities of Scenarios A1, A2, and A3 are 8.9 m/s, 12.2 m/s, and 14.9 m/s, respectively. At Dading village, the maximum velocities in Scenarios A1, A2, and A3 are all approximately 2 m/s.".

**Discussion**

**Line 568:** Based on the accurate bathymetric results of the USV, we found that the maximum and average depth of Bienong Co were 181 m and 85.4 m, respectively, with the lake volume of 102.3 × $10^6$ $m^3$ in August 2020.

**Reply:** This sentence was modified as "Based on the accurate bathymetry data from the USV, we obtained the maximum and the average depths of Bienong Co as 181 m and 85.4 m, respectively, in August 2020, with a

lake volume of $102.3 \times 10^6$ m$^3$.”.

**Line 570:** Considering the rarity of bathymetric data but the frequent occurrence of GLOFs in the region, we attempted to obtain more information about glacial lakes in the region by using bathymetry and lake volume of Bienong Co.

**Reply:** This sentence was modified as “Considering the rarity of bathymetry data but the frequent GLOFs in the Yi'ong Zangbo basin, we attempted to explore more information about glacial lakes in the region from the bathymetry data of Bienong Co.”.

**Line 572:** First, relationships with significant correlations for area-volume, area-depth, and depth-volume of Bienong Co were established (Fig. 15), and it is hoped that this information could provide a valuable data reference for future studies of Bienong Co and other glacial lakes in the region.

**Reply:** This sentence was modified as “First, relationships for area-volume, area-depth, and depth-volume of Bienong Co were established (Fig. 15). It is hoped that this information could provide a valuable reference for future studies of Bienong Co and other glacial lakes in the region.”.

**Line 574:** Then, we compared the depth and lake volume information of Bienong Co with other glacial lakes that have been measured.

**Reply:** This sentence was modified as “Then, we compared the area, depth, and volume of Bienong Co with other glacial lakes for which the bathymetry data are available.”.

**Line 575:** At present, there are few glacial lakes with measured bathymetry on the Tibetan Plateau, and they are mainly concentrated in the Himalayas of Nepal and the Poiqu Basin of Tibet, China.

**Reply:** This sentence was modified as “Bathymetry data are currently available for a small number of glacial lakes on the Tibetan Plateau, which are mainly concentrated in the Himalayas of Nepal and the Poiqu basin of Tibet, China.”.

**Line 577:** Longbasaba is an end moraine-dammed glacial lake located at the northern slope of the Himalayas, which had an area of $1.22 \pm 0.02$ km$^2$ in 2009, with average and maximum depths of $48 \pm 2$ m and $102 \pm 2$ m, respectively, and a volume of $64 \times 10^6$ m$^3$ (Yao et al., 2012).

**Reply:** This sentence was modified as “Longbasaba is a moraine-dammed glacial lake located at the northern slope of the Himalayas, which had an area of $1.22 \pm 0.02$ km$^2$ in 2009, with the average and the maximum depths of $48 \pm 2$ m and $102 \pm 2$ m, respectively, and the volume of $64 \times 10^6$ m$^3$ (Yao et al., 2012).”.

**Line 580:** Although the area of Longbasaba is approximately 6% larger than that of Bienong Co, the lake volume is only 60% of that of Bienong Co.

**Reply:** In this sentence, “the lake volume” was modified as “the volume”.

**Line 581:** This example shows that a glacial lake in the temperate glaciation zone is significantly larger in volume than a similarly-sized glacial lake in the continental glaciation zone. However, due to the lack of measured bathymetric data of glacial lakes in the continental glaciation zone, no more comparisons can be made. We compared the depth and lake volume of Bienong Co with other glacial lakes in the temperate glaciation zone.

**Reply:** This sentence was deleted. Because according to the Reviewer 1's opinion, this statement is not enough

evidenced.

**Line 584:** The area of Lugge glacial lake in Bhutan was approximately 1.17 km$^2$ in 2004, which is slightly larger than that of Bienong Co, but its average depth and maximum depth were 49.8 m and 126 m, respectively, with a lake volume of 58×10$^6$ m$^3$ (Yamada, 2004), which was smaller than the corresponding value of Bienong Co.

**Reply:** This sentence was modified as "Lugge glacial lake in Bhutan had an area of approximately 1.17 km$^2$ in 2004, with the maximum and the average depths of 49.8 m and 126 m, respectively, and the volume of 58×10$^6$ m$^3$ (Yamada, 2004). The area of Lugge glacial lake is slightly larger than that of Bienong Co, but the average and the maximum depths are smaller than the corresponding values of Bienong Co.".

**Line 591:** Areas of Raphsthren glacial lake in Buhtan and Tsho Rolpa glacial lake in Nepal were 1.4 km$^2$ and 1.5 km$^2$ when bathymetries were carried out, which are 22% and 30% larger than that of Bienong Co, but their lake volume is 65% and 84% of that of Bienong Co, respectively (Geological Survey of India, 1995; ICIMOD, 2011).

**Reply:** This sentence was modified as "Areas of Raphsthren glacial lake in Bhutan and Tsho Rolpa glacial lake in Nepal were 1.4 km$^2$ and 1.5 km$^2$ when bathymetries were carried out (Geological Survey of India, 1995; ICIMOD, 2011), which are 22% and 30% larger than these of Bienong Co. However, their volumes are 65% and 84% of these of Bienong Co, respectively.".

**Line 593:** The area of Lower Barun glacial lake in Nepal was 1.8 km$^2$, which is 57% larger than that of Bienong Co, but the lake volume was 112×10$^6$ m$^3$, which is only 9% larger than that of Bienong Co (Haritashya et al., 2018), showing that Bienong Co is deeper and has larger volume.

**Reply:** This sentence was modified as "The area of Lower Barun glacial lake in Nepal was 1.8 km$^2$ when it was measured (Haritashya et al., 2018), which is 57% larger than that of Bienong Co. However, the volume (112×10$^6$ m$^3$) is only 9% larger than that of Bienong Co.".

**Line 597:** Additionally, due to the scarcity of glacial lake bathymetry data and their importance for GLOF hazards, scholars have proposed relationships to estimate volumes of glacial lakes through area, width, and length (O'Connor et al., 2001; Huggel et al., 2002; Sakai, 2012; Wang et al., 2012a; Yao er al., 2012, Cook and Quincey, 2015; Qi et al., 2022).

**Reply:** This sentence was modified as "Additionally, due to the scarcity of glacial lake bathymetry data and its significance for glacial lake hazard studies, scholars have proposed relationships to estimate glacial lake volume by area, width, and length (O'Connor et al., 2001; Huggel et al., 2002; Sakai, 2012; Wang et al., 2012a; Yao et al., 2012, Cook and Quincey, 2015; Qi et al., 2022).".

**Line 600:** We estimate the lake volume of Bienong Co using published equations based on glacial lakes on the Tibetan Plateau, and the results show that the eight published volume-area/width-length relationships all underestimate the volume of Bienong Co to varying degrees.

**Reply:** This sentence was modified as "We estimated the volume of Bienong Co using published relationships based on glacial lakes on the Tibetan Plateau. The results show that the eight published volume-area/width-length relationships all underestimate the volume of Bienong Co to varying degrees.".

**Line 602:** It can be inferred that Bienong Co is the relative deepest glacial lake among these on the Tibetan Plateau that currently have been measured.

**Reply:** This sentence was modified as "It can be inferred that Bienong Co is the deepest glacial lake with a same surface area among the glacial lakes with bathymetry data on the Tibetan Plateau.".

**Line 604:** Whether this is unique to Bienong Co or a common feature of glacial lakes in the region is not yet known, as few glacial lakes in this region have field bathymetry. Future bathymetry is necessary for more typical glacial lakes in the region.

**Reply:** This sentence was deleted because it makes little sense.

**Line 613:** Triggers are the beginning of the simulated GLOFs process chain in this study, and we only consider ice avalanche and landslide scenarios, instead of other factors, such as increased glacial meltwater and heavy precipitation.

**Reply:** This sentence was modified as "The GLOF process chain simulated in this study starts with triggers, including ice avalanches from the mother glacier and landslides from the lateral moraines. Other triggers are not considered, such as increased glacial meltwater or heavy precipitation.".

**Line 616:** The magnitude, location, and probability of ice avalanches and landslides constitute the largest sources of uncertainty in this study.

**Reply:** This sentence was modified as "The magnitude, location, and probability of ice avalanches and landslides are the most significant uncertainty sources of this study.".

**Line 618:** Ice avalanches are the trigger for over 70% of GLOFs on the Tibetan Plateau, but there is no reliable reference for the magnitude, including the release area and depth, of previous ice avalanche events.

**Reply:** This sentence was modified as "Ice avalanches triggered more than 70% of GLOFs on the Tibetan Plateau, but reliable data on the release area and depth from previous ice avalanche events are scarce.".

**Line 620:** Ice avalanches in this study come from the mother glacier tongue, where the slope is relatively steep and the fissures are well-developed.

**Reply:** In this sentence, "relatively" was deleted.

**Line 623:** The RAMMS model can estimate the possible release volume based on the input DEM data and the release area, as well as the release depth, with the estimated released volume of $5 \times 10^6$ m$^3$, $13.1 \times 10^6$ m$^3$, and $41.3 \times 10^6$ m$^3$ for Scenarios A1, A2, and A3, respectively.

**Reply:** This sentence was modified as "The RAMMS model can estimate the possible release volume based on the input of DEM and the release area and depth. The estimated release volumes of Scenarios A1, A2, and A3 are $5 \times 10^6$ m$^3$, $13.1 \times 10^6$ m$^3$, and $41.3 \times 10^6$ m$^3$, respectively.".

**Line 627:** The simulation duration is set to 600 s to ensure integrity of the ice avalanche process, and most of the ice avalanche has already entered the lake within 100 s.

**Reply:** This sentence was modified as "The simulation duration was set to 600 s to ensure the integrity of ice avalanche processes, with most of the ice body entering the lake within the first 100 s.".

**Line 629:** The difference between the volume of the ice avalanche entering the lake and the estimated release volume is mainly determined by the slope between the ice body and the lake and the distance from the lake.

**Reply:** This sentence was modified as "The difference between the volume of the ice avalanche entering the lake and the estimated release volume is mainly determined by the slope and distance between the release source and the lake.".

**Line 632:** The gentler slope and far distance from the lake of the ice body in Scenarios A2 and A3 result in fewer ice avalanches entering the glacial lake, and affect the process of ice avalanche material entering the lake, e.g., Scenarios A2 and A3 have stronger fluctuations in the ice avalanche process than Scenario A1.

**Reply:** This sentence was modified as "In Scenarios A2 and A3, the ice bodies entering the lake account for a small part of the estimated release volumes due to a gentle slope and far distance between the release sources and the lake. The ice avalanche processes are also affected, for example, Scenarios A2 and A3 fluctuate more dramatically than Scenario A1.".

**Line 635:** In addition, we also consider landslides as a trigger given the failure of Jinwu Co in 2020.

**Reply:** This sentence was modified as "In addition, the landslide is determined as a trigger, given the failure of Jinwu Co in 2020.".

**Line 636:** Two release areas were selected by referring to the slope and location of Jinwu Co's landslide.

**Reply:** This sentence was modified as "We selected two release areas by referring to the slope and location of Jinwu Co's landslide.".

**Line 637:** However, the release depth has no quantified reference data and we assumed three release depths of 2 m, 5 m, and 10 m for each release area to simulate the consequences resulting from as many scenarios as possible.

**Reply:** This sentence was modified as "However, the release depth has no specifically quantified reference. We assumed three release depths of 2 m, 5 m, and 10 m for each release area to simulate multiple scenarios.".

**Line 640:** A reliable simulation of potential disaster events such as ice avalanche or landslide in the future using scientific methods can assist people understand the possible risks and raise prevention awareness.

**Reply:** This sentence was modified as "In this study, a reliable simulation of potential future disaster events, such as ice avalanches or landslides, can assist people in understanding the possible risk and raising prevention awareness.".

**Line 642:** However, accurate prediction is generally difficult due to limited geological information and details of triggering mechanisms such as extreme rainfall event.

**Reply:** This sentence was modified as "However, an accurate prediction is generally difficult due to the limited geological information and triggering mechanism details.".

**Line 643:** In this study, the RAMMS model was used to estimate the consequences of potential ice avalanche and landslide events.

**Reply:** This sentence was modified as "In this study, the RAMMS model was used to simulate the movements of potential ice avalanches and landslides.".

**Line 644:** The accuracy and resolution of DEM data is crucial for this model calculations, which can greatly affect the topography and the avalanche and landslide pathways, and is concerned with the accurate reflection of the process of an ice avalanche event (Schneider et al., 2010).

**Reply:** This sentence was modified as "The accuracy of DEM data is crucial for the calculation of the model, which significantly affect the paths, precision, and processes of mass movements (Schneider et al., 2010).".

**Line 647:** The issue of DEM accuracy in glacial environments is mainly due to the topographic changes between the times of DEM acquisition and the avalanche event.

**Reply:** This sentence was modified as "The accuracy of DEM is mainly related to two aspects. One is the topographic change of glacial environment that occurs in the period between the DEM data acquisition and the avalanche event.".

**Line 649:** The DEM with lower resolution can lead to an overestimation of sediment area and an underestimation of sediment thickness (Cesca and D'Agostino, 2008).

**Reply:** This sentence was modified as "The other is the spatial resolution of DEM, a lower one can lead to an overestimation of sediment area and an underestimation of sediment thickness (Cesca and D'Agostino, 2008).".

**Line 651:** ALOS PALSAR DEM product used in this study was released globally in October 2014, with a spatial resolution of 12.5 m, which can generally reflect the intensity and pathway of the ice avalanche and landslide events simulated in this study.

**Reply:** This sentence was modified as "ALOS PALSAR DEM data used in this study was released globally in October 2014, with a spatial resolution of 12.5 m, which can reflect the intensity and path of ice avalanches and landslides simulated in this study.".

**Line 653:** The volumes of future ice avalanches or landslides will almost certainly be smaller than the volumes formulated in this study due to glacier melting caused by climate warming.

**Reply:** This sentence was deleted.

**Line 654:** Moreover, the dry-Coulomb type friction $\mu$ and the viscous-turbulent friction $\xi$ can also influence on the modelling results (Bezak et al., 2019).

**Reply:** This sentence was modified as "Moreover, the dry-Coulomb type friction $\mu$ and the viscous-turbulent friction $\xi$ also influence the modeling results (Bezak et al., 2019).".

**Line 655:** Mikoš and Bezak (2021) suggested that the ice avalanche or landslide magnitude slightly decreases and increases with increasing friction parameters $\mu$ and $\xi$, while the correlation is not strong.

**Reply:** This sentence was modified as "Mikoš and Bezak (2021) found that the magnitude of a mass movement slightly decreases and increases with increasing friction parameters $\mu$ and $\xi$, respectively, while the correlation is not strong.".

**Line 656:** Schneider et al. (2014) found that higher $\mu$ and $\xi$ combinations lead to higher peak velocities, faster stopping mechanisms, and hence shorter process durations.

**Reply:** This sentence was modified as "Schneider et al. (2014) found that a higher $\mu$ and $\xi$ combination leads to a higher peak velocity, faster stopping mechanism, and shorter process duration.".

**Line 659:** The values of $\mu$ and $\xi$ cover wide ranges in the past applications (Mikoš and Bezak, 2021), and they were usually calibrated by actual events. We chose 0.12 and 1,000 m s$^{-2}$ as the values of $\mu$ and $\xi$ in this study, which were used in the avalanche simulation of Lake 513 (Schneider et al., 2014).

**Reply:** This sentence was modified as "The values of $\mu$ and $\xi$ cover wide ranges in past applications (Mikoš and Bezak, 2021), and they were usually calibrated by actual events. In this study, we chose the values of 0.12 and 1,000 m s$^{-2}$ for $\mu$ and $\xi$, which were used in the avalanche simulation of Lake 513 (Schneider et al., 2014).".

**Line 662:** The BASEMENT model was applied to simulate the subsequent chains of GLOF process following ice avalanches and landslides.

**Reply:** This sentence was modified as "The BASEMENT model was applied to simulate the GLOF process chain following the ice avalanche and landslide.".

**Line 664:** Numerical simulations have been limited to simplified 2D SWE simulations (Ghozlani et al., 2013), however, the 2D SWEs are not adequate to simulate waves generated by avalanches because of the large energy dissipation due to significant vertical accelerations (Lala et al., 2018).

**Reply:** This sentence was modified as "Numerical simulations have been limited to simplified 2D SWEs (Ghozlani et al., 2013). However, the 2D SWEs does not perform very well in avalanche-generated wave simulations because of the large energy dissipation caused by significant vertical accelerations (Lala et al., 2018).".

**Line 667:** Therefore, empirical models are often used to calibrate the numerical simulations, such as the Heller-Hager model was employed to calibrate the simulation of the BASEMENT model in past studies (Byers et al., 2018; Lala et al., 2018).

**Reply:** This sentence was modified as "Therefore, empirical models are often used to calibrate the numerical simulations. For example, the Heller-Hager modeling result was employed to calibrate the simulation of the BASEMENT model in previous studies (Byers et al., 2018; Lala et al., 2018).".

**Line 669:** Nevertheless, the numerous parameters required by the Heller-Hager model are still subject to potential uncertainties owing to its simple quantifications about the geometry of the lake.

**Reply:** This sentence was modified as "Nevertheless, the numerous parameters required by the Heller-Hager model are still subject to potential uncertainties because most of them are simple quantifications of a lake's geometry.".

**Line 672:** It is also mentioning that the BASEMENT model can only accept inflow in the form of water volume, but not the mass itself, such as the moraine, ice, snow, and their combinations (Kafle et al., 2016).

**Reply:** This sentence was modified as "Additionally, the BASEMENT model can only accept inflow in the form of water, not actual mass such as moraines, ice, snow, or mixtures (Kafle et al., 2016).".

**Line 675:** Generally, simulations are calibrated by controlling the height and depth of the release area and converting the density between the material and water to influence the flow height and flow velocity in the model as avalanches enter a lake. However, the energy dissipation between the water and the true avalanche mixture is different, and the model can't present the real situation of the avalanche fluid mixing with a lake (Somos-Valenzuela et al., 2016).

**Reply:** This sentence was simplified as "Therefore, the actual situation of avalanches mixing with lake water cannot be simulated because the energy dissipation of water and avalanches entering the lake is different (Somos-Valenzuela et al., 2016).".

**Line 679:** Furthermore, the maximum area of TIN representing a lake also has influence on the wave amplitude, smaller areas lead to finer values, which have little influence on the absolute value of wave amplitude.

**Reply:** This sentence was simplified as "Furthermore, the maximum area of TIN slightly influences the wave amplitude in a lake.".

**Line 680:** The grain size distribution of Bienong Co's dam is an important parameter in simulating the moraine dam's erosions, but it is not obtained in this study.

**Reply:** This sentence was modified as "The grain size distribution of a moraine dam is critical to the dam erosion simulation.".

**Line 682:** The simulations were performed by referencing an inventory of glacial lakes in the Indian Himalayas that have been validated as generally reliable (Worni et al., 2013), but errors in Bienong Co are inevitable.

**Reply:** This sentence was modified as "In this study, simulations were performed by referencing an inventory of the grain size distribution of glacial lakes in the Indian Himalayas. The inventory has been validated as generally reliable (Worni et al., 2013), however, the error in Bienong Co is inevitable.".

**Line 686:** ALOS PALSAR DEM with a spatial resolution of 12.5 m has been widely used in studies on cryospheric changes and disasters.

**Reply:** This sentence was modified as "ALOS PALSAR DEM has been widely used in studies on cryospheric changes and disasters.".

**Line 687:** It was pre-processed to fill sinks in this study, but there is still the phenomenon of flood water accumulating in deep puddles, especially in the relatively narrow valley.

**Reply:** This sentence was modified as "In this study, the preprocessing of sink filling for the DEM was performed, but the phenomenon of flood water accumulating in deep puddles still exists, especially in relatively narrow valleys.".

**Line 690:** However, there are still some smaller bumps that converge the flow to a section of the flow channel, mainly in the downstream area.

**Reply:** This sentence was modified as "Nevertheless, there are still some smaller bumps that converge the flood to some areas of the flow channel, mainly in the downstream region.".

**Line 691:** Therefore, the flooding situation in Qiwuxing, Kemaluo, Xingka and Dading villages might be overestimated, especially in the former two villages because they are relatively far away from the river.

**Reply:** This sentence was modified as "The GLOF impact in Qiwuxing, Kemaluo, Xingka, and Dading villages might be overestimated, especially in the former two villages because they are relatively far away from the river.".

**Line 692:** The latter two villages are still very likely to be threatened by flooding due to their proximity to the river. The resolution of the ALOS PALSAR DEM is clearly insufficient for accurate simulation of flooding in the river-channel.

**Reply:** This sentence was modified as "The two latter villages are still very likely to be threatened by GLOFs

due to the proximity to the river. More accurate simulation of GLOFs in the flow channel is limited by the resolution of ALOS PALSAR DEM.".

**Line 695:** Therefore, more precise topographic information, such as DEM data generated from panchromatic stereo images with spatial resolutions better than 0.8 m obtained by the Gaofen-7 satellite and UAV-derived DEM products with low-cost are prospective.

**Reply:** This sentence was modified as "Therefore, more precise topographic data, such as DEM generated from panchromatic stereo images with spatial resolutions better than 0.8 m obtained by the Gaofen-7 satellite and UAV-derived DEM products with low-cost are prospective.".

**Line 701:** In this study, we do not consider the effects of sediment on flow rheology, as well as the erosion and deposition of sediment along the flow path.

**Reply:** In this sentence, "as well as" was modified as "and".

**Conclusion**

**Line 714:** As a moraine-dammed glacial lake located in a temperate glaciation region, Bienong Co has been highly regarded by local government due to its larger area and high potential for GLOF hazards.

**Reply:** This sentence was modified as "As a moraine-dammed glacial lake located in the SETP, Bienong Co has received significant attention from local government due to its larger area and the high potential for GLOF hazard.".

**Line 715:** Based on bathymetric data, remote sensing images and DEM data, combined with the multiple models of RAMMS, BASEMENT, and Heller-Hager, we completed a comprehensive investigation of the potential GLOF process chain of Bienong Co, including the initial mass movement from the mother glacier and the lateral moraine slope, displacement wave generation and propagation in the lake, overtopping flow and erosion on the moraine dam, and subsequent downstream flooding.

**Reply:** In this sentence, "bathymetric data" was modified as "bathymetry data", "the multiple models" was modified as "multiple models", "completed" was modified as "conducted", "the initial mass movement" was modified as "initial mass movements", "the lateral moraine slope" was modified as "the lateral moraines", "displacement wave" was modified as "displacement waves'", "subsequent downstream flooding" was modified as "subsequent downstream floods". This sentence was modified as "Based on bathymetry data, remote sensing images and DEM data, combined with multiple models of RAMMS, BASEMENT, and Heller-Hager, we conducted a comprehensive investigation of the potential GLOF process chain of Bienong Co, including initial mass movements from the mother glacier and the lateral moraines, displacement waves' generation and propagation in the lake, overtopping flows and erosion on the moraine dam, and subsequent downstream floods.".

**Line 721:** According to the field bathymetric data, the lake basin morphology of Bienong Co features a relatively flat basin bottom and steep flanks, with the slope near the glacier (16.5˚) being steeper than that near the moraine dam (11.3˚).

**Reply:** This sentence was modified as "According to the bathymetry data, the morphology of Bienong Co's lake basin features a relatively flat bottom and steep flanks, with the slope near the glacier (16.5°) being steeper than that near the moraine dam (11.3°).".

**Line 723:** The water storage of Bienong Co was ~102.3 × 10⁶ m³ in August 2020, with a maximum depth of ~181 m. The enormous water storage combined with the fissure-developed mother glacier tongue, steep lateral moraine slope, steep distal facing slope of the moraine dam, and low freeboard make it possess high GLOF potential.

**Reply:** This sentence was modified as "The volume of Bienong Co was ~102.3 × 10⁶ m³ in August 2020, with a maximum depth of ~181 m. Bienong Co is highly prone to a potential GLOF due to the huge volume, the fissure-developed mother glacier tongue, steep slopes of lateral moraines, the steep distal facing slope of the moraine dam, and the low freeboard.".

**Line 731:** As a result, the impact zone, maximum flow height, and maximum flow velocity in the lake also show that Scenarios A1, A2, and A3 are significantly larger than the other six scenarios, in which Scenario B1 is the smallest and Scenario A3 is the largest.

**Reply:** This sentence was modified as "As a result, the impact zones, maximum flow heights, and maximum flow velocities in the lake of Scenarios A1-3 are significantly larger than those of Scenarios B1-3 and C1-3.".

**Line 734:** The overtopping flow of all three scenarios causes erosion of the dam, with little difference in breach depth (19.0 m, 19.1 m, and 19.3 m), but large difference in breach width (295.0 m, 339.4 m, and 368.5 m).

**Reply:** This sentence was modified as "Overtopping flows of all three scenarios cause dam erosion, with little difference in breach depth (19.0 m, 19.1 m, and 19.3 m), but the larger difference in breach width (295.0 m, 339.4 m, and 368.5 m).".

**Line 736:** The volumes of water lost in the lake of the three scenarios are 24.1 × 10⁶ m³, 25.3 × 10⁶ m³, and 26.4 × 10⁶ m³, and the flood peak flows are 4,996 m³/s, 7,817 m³/s, and 13,078 m³/s, respectively.

**Reply:** This sentence was modified as "The volumes of water lost in the lake of Scenarios A1, A2, and A3 are 24.1 × 10⁶ m³, 25.3 × 10⁶ m³, and 26.4 × 10⁶ m³ with peak discharges of 4,996 m³/s, 7,817 m³/s, and 13,078 m³/s, respectively.".

**Line 738:** Among the other six scenarios, only Scenarios B3 and C3 with larger magnitudes formed breaches on the moraine dam, with breaches of 6.5 m and 7.9 m in depth and 153 m and 169 m in width, respectively.

**Reply:** This sentence was modified as "Among Scenarios B1-3 and C1-3, only Scenarios B3 and C3 with larger magnitudes cause dam erosion, with breach depths of 6.5 m and 7.9 m and breach widths of 153 m and 169 m, respectively.".

**Line 741:** Floods all pass through 18 settlements in the downstream river in 20 h, with inundation areas of 7.6 km², 8.0 km², and 8.5 km², as well as average water depths of 8.4 m, 9.1 m, and 10.0 m, respectively.

**Reply:** This sentence was modified as "According to the bathymetry data, the morphology of Bienong Co's lake basin features a relatively flat bottom and steep flanks, with the slope near the glacier (16.5°) being steeper than that near the moraine dam (11.3°).".

**Line 742:** The GLOFs threatened more than half of the villages in the downstream region. Scenarios B1 and B2 and C1 and C2 produce very limited overtopping flow that cannot pose a threat to the downstream region.

**Reply:** This sentence was modified as "Floods threaten more than half of the villages in the downstream flow channel. Scenarios B1, B2, C1, and C2 produce limited overtopping flows that cannot pose a threat to the downstream settlements.".

**Line 744:** Both Scenarios B3 and C3 produced floods that flow through eight downstream settlements within 20 h and had a relatively small impact on them.

**Reply:** This sentence was modified as "Both Scenarios B3 and C3 produce GLOFs that flow through first eight downstream settlements in 20 h, but the impact is relatively small.".

**Line 747:** Bienong Co is the relative deepest glacial lake among those on the Tibetan Plateau that have currently been measured and is markedly different from glacial lakes on the south slope of the Himalayas. Furthermore, it is also important to use high-precision topographic data for disaster simulation of GLOF lakes.

**Reply:** This sentence was modified as "Bienong Co is the known deepest glacial lake with the same surface area on the Tibetan Plateau. There are uncertainties in the GLOF process chain simulation. However, this study is significant for understanding the potential hazard of Bienong Co.".